# Simple physics-based adjustments reconcile the results of Eulerian and Lagrangian techniques for moisture tracking in atmospheric rivers

Alfredo Crespo-Otero[1], Damián Insua-Costa[2], Emilio Hernández-García[3], Cristóbal López[3] and Gonzalo Míguez-Macho[1]

[1]CRETUS, Non-linear Physics Group, Universidade de Santiago de Compostela, Galicia, Spain
[2]Hydro-Climate Extremes Lab (H-Cel), Ghent University, Ghent, 9000, Belgium
[3]Instituto de Física Interdisciplinar y Sistemas Complejos (IFISC), CSIC-UIB, Campus Universitat de les Illes Balears, 07122, Palma de Mallorca, Spain

*Correspondence to*: Alfredo Crespo-Otero (alfredocrespo.otero@usc.es)

**Abstract.** The increase in the number and quality of numerical moisture tracking tools has greatly improved our understanding of the hydrological cycle in recent years. However, the lack of observations has prevented a direct validation of these tools, and it is common to find large discrepancies among the results produced by them, especially between Eulerian and Lagrangian methodologies. Here, we evaluate two diagnostic tools for moisture tracking, the Sodemann et al., (2008) and the Dirmeyer and Brubaker, (1999) methodologies, using simulations from the Lagrangian model FLEXPART. We assess their performance against the Weather Research and Forecasting (WRF) model with Eulerian Water Vapor Tracers (WRF-WVTs). Assuming WRF-WVTs results as a proxy for reality, we explore the discrepancies between the Eulerian and Lagrangian approaches for five precipitation events associated with atmospheric rivers and assess some physics-based adjustments to the Lagrangian tools. As in previous studies, we find a negative bias in the contribution of remote sources, such as tropical ones, and an overestimation of local contributions. Quantitatively, the mean absolute error skill score (MAESS) with respect to WRF-WVTs for contributions from selected source regions is 0.74 for the Sodemann et al., (2008) methodology and 0.77 for the Dirmeyer and Brubaker, (1999) diagnostic tool. The implementation of simple and logical corrections leads to a significant improvement in both methodologies, as the skill score improves to 0.84 and 0.87, respectively. Although these modifications may need to be adjusted for other types of precipitation events, our results demonstrate that Lagrangian techniques are a viable and compatible alternative to Eulerian water vapor tracers, and that the main discrepancies between the different methodologies can be derived from the obviation of basic physical considerations that may be easily straightened out.

## 1 Introduction

Water is one of Earth's most important resources, and its availability and distribution are crucial to the future of the different ecosystems, including humans. Given its importance and scarcity, it is vital to understand how water is transported between

different regions of our planet (Oki and Kanae, 2006). Water can be transported within a catchment through rivers and groundwater flow. However, the transport of water between basins, or from ocean to land, is mostly done via the atmosphere, through what is known as the atmospheric branch of the water cycle. To investigate the latter, researchers have developed different moisture tracking methods that make it possible to analyse where moisture contributing to precipitation

has previously evaporated (see Gimeno et al., 2012, for a review). Apart from analytical approaches (Trenberth, 1999; Dominguez et al., 2006; Rios-Entenza and Miguez-Macho, 2014), the most used models to this end are numerical or computational routines. Within this group, two main classes can be distinguished: Eulerian water vapor tracers, e.g., (Koster et al., 1986; Yoshimura et al., 2004; Sodemann et al., 2009; Insua-Costa and Miguez-Macho, 2018), and Lagrangian moisture source diagnostics, (Dirmeyer and Brubaker, 1999; Stohl and James, 2004; Sodemann et al., 2008). The

classification can be based on alternative criteria, e.g. whether the moisture tracking is performed simultaneously with the computation of meteorological fields, such as wind or specific humidity (online), or not (offline). Additionally, the tracking can be either forward or backward, depending on whether the moisture is tracked forward or backward in time. Despite the diversity of methodologies, most academics often use a single model, and the few works in which multiple methods have been tested show that results may not be in agreement (van der Ent et al., 2013; Winschall et al., 2014; Cloux et al., 2021).


The aforementioned techniques have been particularly used to identify moisture sources in precipitation events associated with atmospheric rivers (ARs). ARs are structures of enhanced moisture and intense water vapor transport in the atmosphere, typically located in the pre-cold frontal region of an extratropical cyclone (Ralph et al., 2005), and are responsible for the majority of the poleward water vapor flux across mid-latitudes (Zhu and Newell, 1998). Their large-scale nature, together

with their connection to extreme precipitation events (Ralph et al., 2006; Lavers and Villarini, 2013), has made them the focus of several studies aimed at determining the origin of the moisture they carry—specifically, whether it is primarily transported from remote regions or local sources. This issue has been addressed using Eulerian water vapor tracers (Sodemann and Stohl, 2013; Eiras-Barca et al., 2017; Hu and Dominguez, 2019), Lagrangian techniques (Liberato et al., 2013; Ramos et al., 2016) or both (Bonne et al., 2015). However, those studies in which they quantify the relative

importance of different moisture sources focus on individual cases, so the debate on the origin of moisture in ARs is not yet completely closed. This is reflected in the definition of AR given in the Glossary of Meteorology, where it is indicated that the sources of moisture can be tropical and/or extratropical (Ralph et al., 2018; American Meteorological Society, 2025).

In this context, the goal of this paper is to assess the differences and reduce the discrepancies between two Lagrangian

methodologies for the computation of moisture sources for precipitation (or precipitation sources) focusing on AR-related rainfall events. The strategy we adopt is to run the Lagrangian models on atmospheric data from simulations of the Weather Research and Forecasting (WRF) model with Water Vapor Tracers (WRF-WVTs; Insua-Costa and Miguez-Macho, 2018) and introduce physically based modifications so that the results are aligned with those provided by the latter tool. The rationale for this approach is simple. Online Eulerian water vapor tracers, being coupled to a meteorological model, account

for all the physical processes affecting atmospheric moisture that are resolved or parameterized by the model. In the case of WRF-WVTs, they are internally consistent, showing an almost exact performance within the "model world" (Insua-Costa and Miguez-Macho, 2018), i.e. they constitute synthetic observations generated from the model simulation. Furthermore, in the absence of direct observations, results provided by WRF-WVTs are particularly suitable to be considered as reference when comparing with other methods, as long as the simulated atmosphere behaves like the real one and follows it closely.

Their disadvantage, however, is that they are computationally expensive, and therefore their application over long time periods or in many case studies is often unfeasible. Additionally, the amount of information they offer is limited, as the moisture source to be tagged needs to be predefined. In contrast, Lagrangian methods are much more computationally efficient and provide gridded information, but they are sensitive to a range of hypotheses and parameter choices, which significantly increases their uncertainty. Achieving a Lagrangian moisture source diagnostic that mimics WRF-WVTs results

would therefore imply having a very accurate and at the same time flexible tool that can be applied to a large number of ARs, our goal for the future, but probably also to other types of weather or climate phenomena.

The strategy of using water vapor tracers as ground truth versus Lagrangian diagnostics has been previously used in several studies. For example, in van der Ent et al., (2013) the outcomes of a tagging tool implemented in the MM5 model are taken

as ground truth to analyse two other offline methods. Winschall et al., (2014) employed a moisture tagging technique integrated into the COSMO weather prediction model and compared the results with those of the WaterSip moisture source diagnostic (Sodemann et al., 2008), which used air particle trajectories from the LAGRANTO model (Sprenger and Wernli, 2015). More recently, Cloux et al., (2021) used this same Lagrangian diagnostic tool to compute precipitation sources, but with trajectories generated with FLEXPART-WRF (Brioude et al., 2013), and compared the results with those of WRF-

WVTs. While Winschall et al., (2014) show the complementarity of the results provided by the Eulerian and Lagrangian approaches, in Cloux et al., (2021) they specifically highlight the large discrepancies between the results provided by Lagrangian and Eulerian tools, although they did not provide improvements to reconcile the different methodologies.

In our case, the FLEXPART-WRF model is employed to generate back trajectories of air parcels contributing to

precipitation in five AR events, and our own implementations of two widely used Lagrangian diagnostic tools for estimating moisture sources are assessed: the Sodemann et al., (2008) and the Dirmeyer and Brubaker, (1999) methodologies. Our focus is on understanding the origins of discrepancies between the outcomes of these methodologies and those derived from WRF-WVTs, with the aim of introducing physics-based adjustments to them that minimize these differences. Our framework is particularly well suited for this validation, as both the moisture tracking with WRF-WVTs and the calculation of air particle

trajectories with FLEXPART-WRF are driven by the same WRF-simulated atmospheric fields. Additionally, we validate the proposed modifications under a different scenario where trajectories are computed using the FLEXPART model (Pisso et al., 2019) forced with data from the ERA5 reanalysis (Hersbach et al., 2020), instead of WRF.

In what follows, we first present the AR cases studied (Section 2.1) and the Eulerian and Lagrangian methodologies used to calculate precipitation sources (Sect. 2.2 and Sect. 2.3). Section 3 includes the main results of our study, where we first focus on the comparison of the results produced by the Lagrangian methodologies with those provided by the WRF-WVTs model, and then assess how the Lagrangian and Eulerian approaches can be brought into closer agreement. Finally, Sect. 4 provides a summary and the conclusions of this work.

## 2 Methods

### 2.1 Selected AR cases

In this section we introduce the five precipitation events selected, all of them caused by the landfall of an AR and well documented in the literature. We chose cases from all over the world, not just from a specific region. In Fig. 1 we show both the integrated water vapor and accumulated precipitation fields from WRF simulations (see Sect. 2.2) for these cases. Black boxes in this figure highlight the areas most affected by rainfall (Table 1). A more detailed description of these episodes can be found in Sect. S1 in the Supplement.

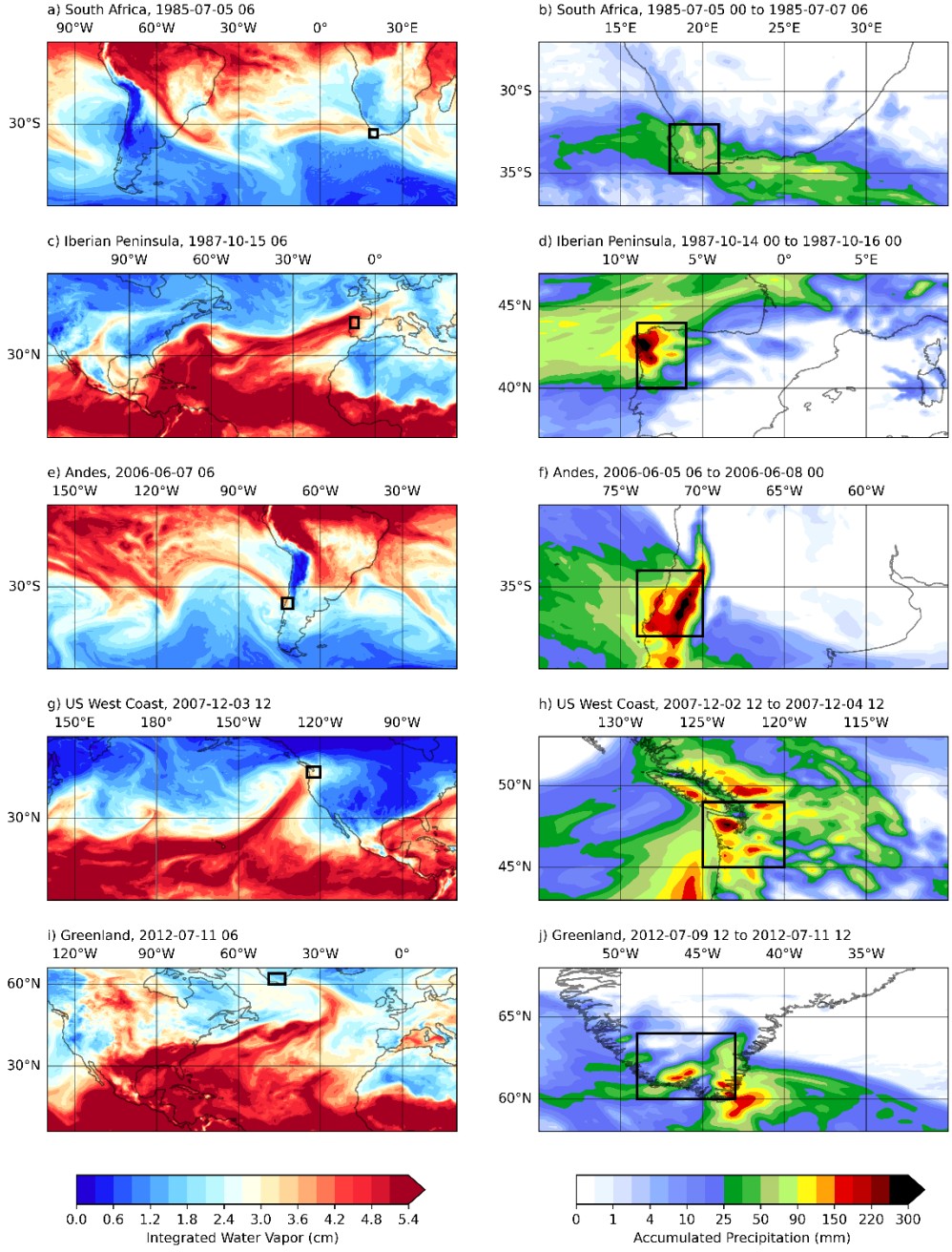

Figure 1: Integrated water vapor (left) and accumulated precipitation (right) during the South Africa (a and b), Iberian Peninsula (c and d), Andes (e and f), US West Coast (g and h), and Greenland (i and j) AR-related precipitation events. The black boxes are the regions in which precipitation will be tracked.

| case | $t_i$ | $\Delta t$ (h) | $\lambda_1$ (°) | $\lambda_2$ (°) | $\phi_1$ (°) | $\phi_2$ (°) | $P_{WRF}$ (mm) | $P_{ERA5}$ (mm) |
|---|---|---|---|---|---|---|---|---|
| South Africa | 1985-07-05 00 | 55 | -35.0 | -32.0 | 18.0 | 21.0 | 45.9 | 46.4 |
| Iberian Peninsula | 1987-10-14 00 | 49 | 40.0 | 44.0 | -9.0 | -6.0 | 91.9 | 89.4 |
| Andes | 2006-06-05 06 | 67 | -38.0 | -34.0 | -74.0 | -70.0 | 127.3 | 120.8 |
| US West Coast | 2007-12-02 12 | 49 | 45.0 | 49.0 | -125.0 | -120.0 | 88.3 | 100.3 |
| Greenland | 2012-07-09 12 | 49 | 60.0 | 64.0 | -49.0 | -43.0 | 36.4 | 31.6 |

Table 1: Starting date and time ($t_i$) and duration ($\Delta t$) of the rainfall events associated to the five ARs studied, together with the coordinates defining the region where precipitation will be tracked (black boxes in Fig. 1): $\lambda_1$, $\lambda_2$ (latitudes) and $\phi_1$, $\phi_2$ (longitudes). The last two columns show the average precipitation in the region simulated by WRF ($P_{WRF}$) and in the reanalysis ($P_{ERA5}$).

The first rainfall event considered affected South Africa in July 1985. Figure 1a and 1b clearly show that the event was linked to an AR, as already indicated by other authors (Blamey et al., 2018). The second AR affected the northwest region of the Iberian Peninsula (Fig. 1c and 1d) and was associated with the infamous extratropical cyclone coined as the "Great Storm" for the catastrophic impacts it caused in the United Kingdom. Moisture sources for this AR-related precipitation event were previously analysed using the WRF-WVTs tool in Eiras-Barca et al., (2017). The third case selected corresponds to an AR that impacted central Chile and the Andes (Fig. 1e and 1f), resulting in floods and damage in the region, and was analysed in Viale et al., (2013). The fourth AR considered (Fig. 1g and 1h) was associated with the well-known Great Coastal Gale of 2007, affecting the US West Coast. The moisture sources for this event were also investigated in Eiras-Barca et al., (2017) using the WRF-WVTs model. Finally, the last AR studied hit Greenland (Fig. 1i and 1j) leading to a severe ice and snow melting episode, (Mattingly et al., 2018).

## 2.2 WRF-WVTs

As previously mentioned, the moisture tracking model used as a proxy for reality in this study is WRF-WVTs, (Insua-Costa and Miguez-Macho, 2018), a moisture tagging tool implemented in the WRF model version 4.3.3, (Skamarock et al., 2021). Here WRF is run at a spatial resolution of 20 km and 38 vertical levels in two different semi-hemispherical domains (Fig. 2), depending on whether the AR of study occurs in the northern or southern hemisphere. Initial and boundary conditions come from the ERA5 reanalysis. We use a spectral nudging technique (Miguez-Macho et al., 2004) to prevent large-scale atmospheric fields (waves longer than around 1000 km) from deviating significantly from the reanalysis. In our case, only winds, temperature and geopotential height are nudged. Spectral nudging ensures an accurate representation of the atmosphere throughout the simulation period, even several days after the simulation has started. This aspect is particularly important in our tracking experiments, since we start our simulations 30 days before the beginning of the rainfall episode

(Table 1) to allow enough time for the moisture to evaporate. The underlying reason for this long spin-up time is that
probability density functions for atmospheric residence time of water vapor are positively skewed (van der Ent and
Tuinenburg, 2017). Finally, the main parameterizations used were the Yonsei University (YSU) for the boundary layer
(Hong et al., 2006), the WRF single-moment-6-class (WSM6) for microphysics (Hong and Lim, 2004), and the Kain-Fritsch
for convection (Kain, 2004), which are required to use the moisture tagging capability.

The WRF-WVTs tool is an Eulerian, online and forward moisture tracking technique, as the water vapor tracers are coupled
to the meteorological model, and the latter needs to be run forward in time. As clarified in Insua-Costa et al., (2022), to track
moisture coming from a source region $S$ in the WRF-WVTs framework it is necessary to modify the source term in the WRF
prognostic equation for moisture ($QFX$):

$$TRQFX = QFX \cdot M, \tag{1}$$

where $M$ is a binary array designating the region $S$ with values of 1 and the rest with 0. In our case $QFX$ does not come from
the evaporative flux simulated by WRF land surface scheme, but we ingest it from the ERA5 reanalysis. Specifically, if $E$
represents the ingested evaporation interpolated onto the model grid, $\rho_{\mathrm{w}}$ denotes the water density and $\Delta T$ is the time interval
in the reanalysis, then the moisture flux from the surface $QFX$ can be expressed as:

$$QFX = -\frac{\rho_{\mathrm{w}} E}{\Delta T}. \tag{2}$$

The negative sign accounts for the different criteria for positive surface fluxes between WRF and ERA5. If $\Delta T$ were large, a
time interpolation would also be needed. Finally, the tool tracks moisture until it precipitates, so a new variable representing
tracer precipitation, i.e. originating from the source region $S$ is defined ($TP_S$). Consequently, the fraction of rainfall in a
specific region $R$ coming from $S$ can be determined as

$$F_S = \frac{\sum_{(i,j) \in \mathrm{R}} TP_S(i,j)}{\sum_{(i,j) \in \mathrm{R}} P(i,j)}, \tag{3}$$

where $P$ is total precipitation and regions $R$ for the different ARs are defined in Table 1 and plotted in Fig. 1 as black boxes.

WRF-WVTs can track not only moisture evaporated from $S$, but also moisture advected from $S$, by changing the evaporative
flux $QFX$ by the specific humidity $q$ in Eq. (1). In this case the source is three dimensional (3-D) and in the former, two-
dimensional (2-D). We consider 11 source regions in each domain (Fig. 2), selected to maximize the contribution from the 2-
D sources (nine in total). We only use two 3-D sources at the model domain boundaries, in order to track all moisture
originating from outside the model domain (red lines in Fig. 2). For additional information on the WRF-WVTs simulations,
we refer to Sect. S2 in the Supplement.

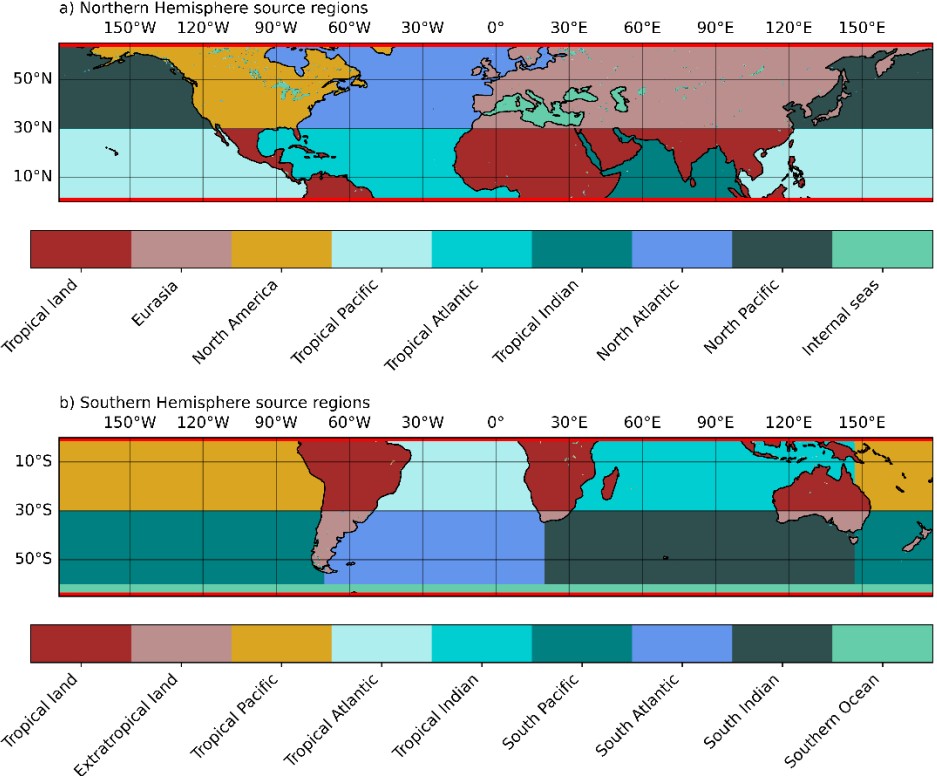

Figure 2: The two simulation domains and the nine 2-D moisture sources analysed for each of them, along with the two 3-D sources at the domain boundaries (red lines). Northern Hemisphere cases use the configuration in a), while Southern Hemisphere ones use that in b).

## 2.3 Lagrangian moisture source diagnostics

The two Lagrangian moisture source diagnostics we use, as previously commented, operate as post-processing routines for the Lagrangian particle dispersion model FLEXPART (Pisso et al., 2019), widely utilized in studies dedicated to understanding the origin and transport of atmospheric humidity (Sodemann and Stohl, 2009; Drumond et al., 2014; Ramos et al., 2016). This model is prepared to read input data from the European Centre for Medium-Range Weather Forecasts (ECMWF) Integrated Forecast System (IFS) and the United States National Center of Environmental Prediction (NCEP) Global Forecast System (GFS). Additionally, an adapted version of FLEXPART, known as FLEXPART-WRF (Brioude et al., 2013), is enabled to process output data from the WRF model. We will start using FLEXPART-WRF to generate the air parcel trajectories and then extend our comparison with WRF-WVTs to FLEXPART constrained with ERA5 (from now on, FLEXPART-ERA5). Both FLEXPART-ERA5 and FLEXPART-WRF provide hourly information about the 3-D position, specific humidity, pressure, density and temperature of the parcel, together with the atmospheric boundary layer (ABL) height. Except for the position of the parcel and the ABL height, the other variables are obtained by interpolating ERA5 or

WRF to the position of the parcels. While FLEXPART-WRF input data were described in the previous section (they are exactly the WRF 3-hourly output data), in our case FLEXPART-ERA5 reads hourly data from the ERA5 reanalysis on a $0.5° \times 0.5°$ grid and across the 70 vertical model levels closest to the surface, with the pressure at the highest level we use being around 140 hPa. In both cases parcels are released using the domain filling option over the black boxes in Fig. 1, such that they are vertically distributed following the density profile. Additional details about FLEXPART-ERA5 and FLEXPART-WRF simulations are given in Sect. S2 in the Supplement.

The first Lagrangian moisture source diagnostic we employ is based on the methodology introduced in Sodemann et al., (2008), currently known as WaterSip (Sodemann, 2025) and also implemented in other moisture tracking frameworks (Fernández-Alvarez et al., 2022; Keune et al., 2022). In our case we use our own implementation of the diagnostic, which we cannot assure is identical to WaterSip, and we will refer to it as SOD08. This method assumes that the atmospheric column over the region where precipitation occurs is filled with air parcels, and that their trajectories contain information about their location and specific humidity at 6-hourly intervals for the previous days, in our case 30 days. Using this information, SOD08 begins by identifying where each parcel uptakes water, by computing the difference in specific humidity between consecutive time steps, $\Delta q_{\mathrm{parcel},t} = q(\vec{r}_{\mathrm{parcel},t}, t) - q(\vec{r}_{\mathrm{parcel},t-1}, t-1)$. A positive difference is interpreted as evaporation, while a decrease in humidity is linked to precipitation. In Sodemann et al., (2008), apart from the specific humidity increase, additional criteria are imposed to determine whether a moisture increment is actually linked to surface evaporation or not. These are (1) requiring that the specific humidity increase exceeds a minimum threshold, that we set here to $\Delta q = 0.2$ g kg$^{-1}$ in a 6-hourly interval, and (2) it occurs within the ABL. This allows to identify the uptake points for each parcel by attributing the selected moisture increments to the mid-point parcel position $(\vec{r}_{\mathrm{parcel},t} + \vec{r}_{\mathrm{parcel},t-1})/2$. SOD08 then proceeds forward in time, linearly discounting each moisture uptake $\Delta q_{\mathrm{parcel},t}$ every time a subsequent specific humidity decrease is observed. Once the most recent time step is reached (i.e., at the precipitation event), a spatial distribution for the moisture origin of each parcel is obtained. The final stage of the methodology involves selecting only those parcels that contribute to precipitation and weighting the spatial distributions by the final humidity loss, thus obtaining the moisture sources for precipitation. For the selection of these parcels, a threshold of 80% is applied to the relative humidity at arrival, ensuring the exclusion of unsaturated parcels that could hardly have contributed to the precipitation. Obviously, parcels with a final humidity increase are also discarded. For a detailed mathematical description of the method, see Sect. S3.1 in the Supplement.

This diagnostic has been used in other studies with some modifications with respect to the original methodology introduced in Sodemann et al., (2008). Some subsequent works (Fremme and Sodemann, 2019) have ignored the ABL filter for identifying moisture uptakes arguing that parcels above the ABL can still be indirectly influenced by surface evaporation through convection. As this is the configuration mostly used nowadays, this will be for us the basic SOD08 configuration. A

less common modification is to filter specific humidity decreases, such that previous contributions are only discounted if a specific humidity decrease occurs and the relative humidity of the parcel is higher than 80 % (Dütsch et al., 2018; Cheng and Lu, 2023). This should not be confused with the relative humidity filter applied at the most recent time step used to select parcels contributing to the precipitation event, as in the case of Dütsch et al., (2018) and Cheng and Lu, (2023) the criterion is applied en-route and used to filter humidity decreases, not parcels. Finally, the Sodemann et al., (2008) methodology has also been shown to be sensitive to the choice of the minimum specific humidity increment. Here, we initially use the recommended and most common setup: $\Delta q = 0.2$ g kg$^{-1}$ for 6-hourly trajectories. Note that the time resolution of the trajectories is degraded from 1 to 6 hours, as using a very high temporal resolution can introduce noise into the diagnostic, leading to systematic biases (see Sect. 3.2.1 for further details).

The second type of Lagrangian moisture source diagnostic we employ was originally introduced by Dirmeyer and Brubaker, (1999), and is currently widely used in the framework of the UTrack-atmospheric-moisture model (Tuinenburg and Staal, 2020). The same approach is also used by other studies, such as Holgate et al., (2020), so we will refer to it as the Dirmeyer and Brubaker, (1999) methodology (hereafter, DB99). Unlike SOD08, the latter does not attribute moisture sources based on the specific humidity of air parcels, but using evaporation and precipitable water fields. When evaporation occurs at a parcel's location, a fraction of its moisture (equal to the ratio of evaporation to precipitable water) is attributed to that location, and the parcel's moisture content is updated accordingly. This process continues backward in time until 99 % of the parcel's moisture has been allocated, with a maximum duration of 30 days in our case. At the end of the calculation, a spatial distribution for the moisture origin of each parcel is obtained, similar to SOD08. When aggregating results from all parcels, as all of them account for total rainfall amount, the precipitation sources are obtained. An important difference with S0D08 is that DB99 is supposed to calculate the parcel trajectories itself. When doing that, parcels are initially released over the region where precipitation occurs at a random, humidity-weighted vertical level, so that the contribution of each parcel is weighted by the humidity profile, instead of the water lost in the last time step, as in SOD08. However, in our case we use FLEXPART-ERA5 and FLEXPART-WRF trajectories at hourly resolution and implement only the diagnostic tool to compute the moisture sources for precipitation. Thus, since in our simulations parcels are vertically released following the density profile, we weight the contribution of each parcel by its specific humidity to match the DB99 methodology. For a detailed explanation of this method, we again refer to Sect. S3.1 in the Supplement.

Finally, in order to compare both Lagrangian moisture source diagnostics with WRF-WVTs, their output fields - representing the amount of evaporated water resulting in precipitation - must be aggregated to the selected source regions and then divided by total precipitation, in order to calculate the rainfall fractions $F_S$, as in Eq. (2). This allows us to assess each Lagrangian method by using the Root Mean Square Error with respect to WRF-WVTs,

$$\text{RMSE}_m = \sqrt{\frac{1}{N}\sum_{i=1}^{N}\left(F_{S_i}^{\text{WVTs}} - F_{S_i}^{m}\right)^2},$$

(6)

where $N$ is the number of sources and $F_{S_i}^{\mathrm{WVTs}}$, $F_{S_i}^m$ the precipitation fractions for WRF-WVTs and for the evaluated Lagrangian method, respectively. Given that this metric is very sensitive to outliers, the Mean Absolute Error (MAE) and its associated score, the Mean Absolute Error Skill Score (MAESS), are also used to obtain an average rating:

$$\mathrm{MAE}_m = \frac{1}{N}\sum_{i=1}^{N}\left|F_{S_i}^{\mathrm{WVTs}} - F_{S_i}^m\right|, \quad \mathrm{MAESS} = 1 - \frac{\mathrm{MAE}_m}{\mathrm{MAE}_r}, \tag{7}$$

where $\mathrm{MAE}_r$ is the MAE if all rainfall fractions were equal to 1/$N$. As usual with a skill score, the closer to 1 means that the results of the Lagrangian diagnostic are closer to those of WRF-WVTs. Note that we use the MAESS and not the coefficient of determination as skill score because the latter leads to negative values in our analysis, and this can be problematic when averaging over all AR cases.

## 3. Results

In this section the main results of this study are presented. In Sect. 3.1 the most basic configurations of the SOD08 and DB99 methodologies are assessed by comparing their outputs with those provided by the WRF-WVTs tool. Next, in Sect. 3.2 some physics-based adjustments are introduced in the Lagrangian methodologies with the intention of minimising the discrepancies with the WRF-WVTs results. Finally, in Sect. 3.3 we test the proposed modifications when the trajectories are generated by FLEXPART-ERA5, with input data from the ERA5 reanalysis. In this case, the additional fields required by

the diagnostic tools (e.g. evaporation and precipitable water in the case of DB99) are also taken from ERA5, rather than from WRF simulations.

### 3.1 Raw comparison of WRF-WVTs vs SOD08 and DB99

Figure 3 illustrates the rainfall fractions from WRF-WVTs for the five precipitation events introduced before, and for the eleven source regions considered. The results are categorized into Northern Hemisphere and Southern Hemisphere cases, as

the selected source regions are identical for ARs occurring in the same hemisphere. For the Iberian Peninsula, US West Coast, and South Africa events, the most important contributions are from the extratropical oceanic areas where the ARs developed. Conversely, in the Andes case the primary contribution is from the Tropical Pacific. In these four cases, more than 75 % of the precipitation originates from oceanic sources. However, a different pattern is observed in the Greenland case, where there is a remarkable continental contribution from North America, reducing the oceanic precipitation fraction to

below 50 %. This is consistent with previous studies showing that ARs in polar regions can exhibit unique features (Guan and Waliser, 2019) and that moisture sources in AR-related precipitation events can be highly variable.

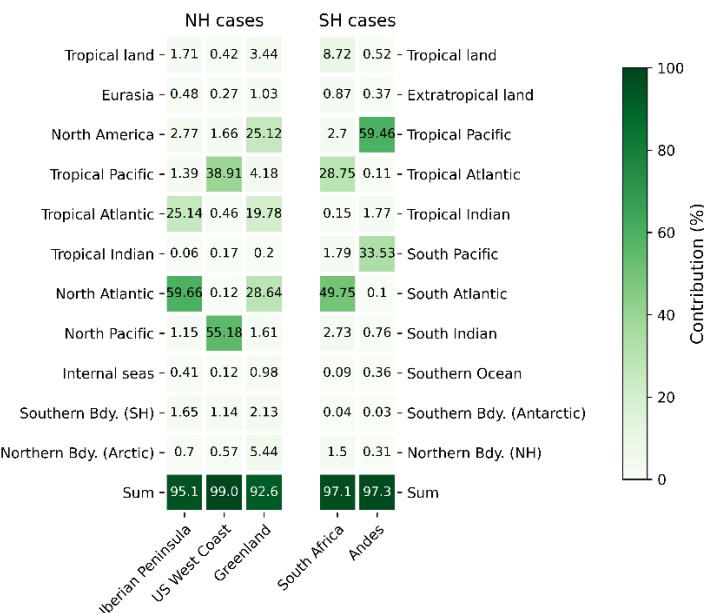

Figure 3: Precipitation fractions (%) in the different rainfall events originating from the selected sources, computed using the WRF-WVTs model. To the left, Northern Hemisphere (NH) cases. To the right, Southern Hemisphere (SH) cases. The last row shows the sum of all contributions.

We start the comparison with WRF-WVTs by using the most basic configurations of the Lagrangian methodologies, described in Sect. 2.3. Figure 4a presents the results for the SOD08 diagnostic tool. A significant bias is observed in certain contributions, particularly evident in those of the main tropical and extratropical oceanic source regions for each case. There is a clear tendency for SOD08 to underestimate tropical and overestimate extratropical contributions, something that has already been observed in previous studies (Cloux et al., 2021;). For the Iberian Peninsula and South Africa cases, biases are high for the main sources, of around 10 %, with a RMSE (see Fig. 5) of 3.80 % and 4.83 %, respectively. The Greenland case presents larger deviations, as SOD08 overestimates the fraction of precipitation originating from the North Atlantic by almost 40 %, leading to a RMSE of 12.1 %. However, the US West Coast and Andes cases do show results more consistent with WRF-WVTs, with maximum biases of 2.75 % and 2.55 %, respectively. Overall, the average RMSE is 5.20 %, while the average MAESS is 0.74 (see Table S1 in the Supplement).

Fig. 4b displays the results for the DB99 methodology. Comparing to SOD08, the biases are larger for the US West Coast and Andes cases, smaller for the Iberian Peninsula and Greenland cases, and similar for the South Africa case. The average RMSE is smaller, 4.64 %, mainly due to the poor performance of SOD08 in the Greenland case. Again, an underestimation of tropical contributions is observed, particularly evident in the Southern Hemisphere cases. For example, for the South Africa rainfall episode the estimated contribution from the Tropical Atlantic is 12.25 %, far from the "true" value of 28.75 %

shown in Fig. 3. This underestimation of tropical contributions has also been documented in the literature. Specifically, in Staal and Koren, (2023) they compute the rainfall fractions using the UTrack-atmospheric-moisture model for the 2021 European floods and compare their results to the WRF-WVTs fractions calculated in Insua-Costa et al., (2022). Although the results are quite similar, the differences in the North Atlantic and tropical contributions are larger than 10 %, as in Fig. 4b.

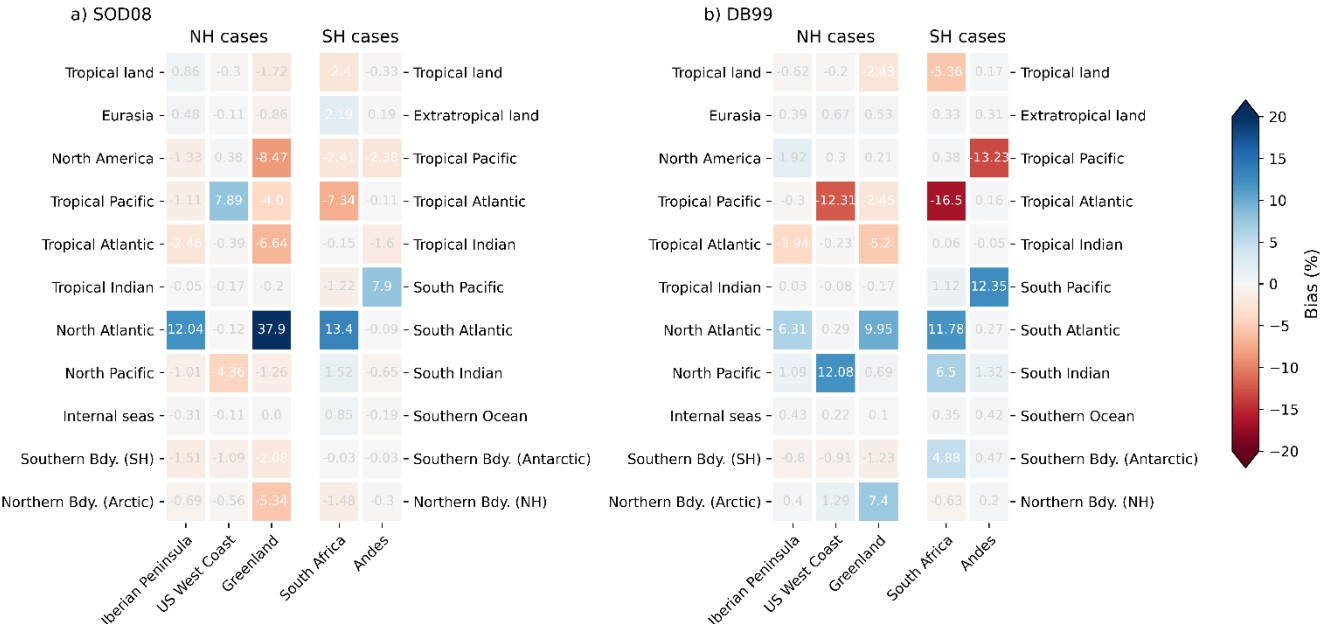

Figure 4: Bias in precipitation fraction (%) obtained using the basic configurations of the SOD08 and DB99 moisture source diagnostics, for trajectories generated with WRF input data (FLEXPART-WRF). Biases are computed subtracting the "true" outcomes of WRF-WVTs from the corresponding values of SOD08 and DB99.

## 3.2 Improvements in the Lagrangian moisture source diagnostics

### 3.2.1 SOD08

The most remarkable conclusion extracted from Fig. 4a is that both the SOD08 and DB99 methodologies present a systematic underestimation of tropical contributions in AR-related precipitation events. While this discrepancy could be attributed to potential systematic errors in trajectory calculations, we will proceed under the assumption that these calculations are correct and instead focus on exploring the inherent capabilities of the SOD08 diagnostic itself to address this issue. Specifically, we explore the hypothesis that humidity fluctuations along the trajectories not associated with surface 320 evaporation or precipitation may account for the observed underestimation of tropical contributions and, more broadly, of remote sources. The problem of these humidity fluctuations in the SOD08 diagnostic was already recognized in the original study of Sodemann et al., (2008), which motivated the introduction of the minimum specific humidity increase $\Delta q$ to filter out uptakes caused by numerical noise, as well as the ABL criterion to exclude humidity increases related to physical

processes other than surface evaporation (e.g., convection, turbulence or evaporating precipitation). To explain how these

fluctuations contribute to the underestimation of remote sources, let us assume an air parcel that at a certain time step increased its specific humidity in 2.0 g kg$^{-1}$, and then experiences a decrease and an increase of 0.05 g kg$^{-1}$ not caused by evaporation or precipitation, such that it returns to its original value. Although these two fluctuations seem to offset each other, the original uptake of 2.0 g kg$^{-1}$ is now reduced to 2.0(1-0.05/2.0)=1.95 g kg$^{-1}$. If another such decrease occurs, this value is updated to 1.95(1-0.05/2.0)=1.90 g kg$^{-1}$. Thus, if these fluctuations continue to occur, we are multiplying the initial

value by a number smaller than 1 many times (potentially, as many as time steps in 30 days), so this original contribution clearly ends up dropping well below its true value. In other words, this shows that negative changes in specific humidity not caused by precipitation penalize much earlier contributions in SOD08, i.e. remote sources, as the error caused by a single fluctuation affects all previous contributions, so that the early moisture uptakes will be affected by many more fluctuations.

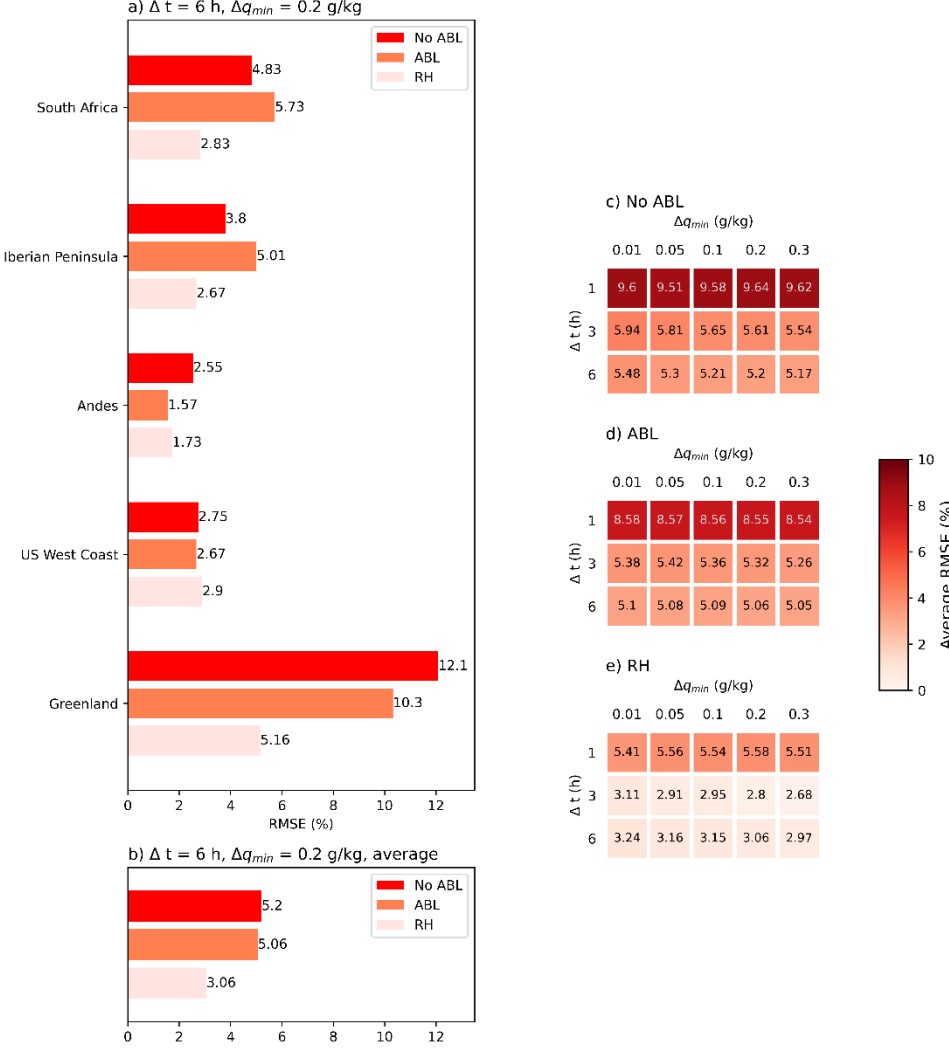

Figure 5: RMSE for the five AR-related precipitation events (panel a) and average of all of them (panel b) in the three tested configurations of SOD08, with the standard values of the specific humidity threshold and time step. On the right, average RMSE for a range of values of these parameters, in the case of the most basic configuration (No ABL, panel c), neglecting increments above the ABL (ABL, panel d), and discarding decreases below a minimum relative humidity (RH, panel e).

In the past, efforts have focused on reducing spurious positive uptakes by imposing a minimum threshold in specific humidity increases and only considering moisture gains below the ABL height. However, as discussed in the methodology, recent studies propose to shift the focus to moisture decreases by requiring a minimum relative humidity of 80 % immediately before a decrease in specific humidity occurs (Dütsch et al., 2018; Cheng and Lu, 2023). If this is not the case, previous contributions are not reduced due to this decrease. This should reduce the decreases in specific humidity not caused

by precipitation, as requiring a minimum relative humidity of 80 % has been a common parameterization of the existence of clouds and precipitation in the past. Thus, according to our hypothesis, the underestimation of tropical contributions observed in Fig. 4a should be reduced.

Figure 5a shows the RMSE of the precipitation fractions computed using the SOD08 diagnostic, using WRF-WVTs results as the true values (Fig. 3), while Fig. 5b shows the average RMSE across the five cases. The values shown by the red bars ("No ABL") have already been discussed above as they correspond to the basic configuration of SOD08. We now present the results also for the configuration in which we discard moisture uptakes above the ABL height ("ABL"; orange) and for the configuration in which we consider the RH criterion to filter specific humidity decreases en-route ("RH"; salmon). In the last two cases, we computed the precipitation fractions dividing by the attributed precipitation, as this is typically much lower ("ABL" configuration) or higher ("RH" configuration) than the precipitation simulated by WRF or in the reanalysis. Consistent with the findings of Cloux et al., (2021), a modest improvement is observed for the "ABL" configuration, as the average RMSE is reduced from 5.20 % to 5.06 %. However, this behavior is not the same for all cases, as for some of them the error increases significantly (South Africa and Iberian Peninsula cases), while in the Andes case the RMSE decreases markedly from 2.55 % to 1.57 %. In contrast, the "RH" configuration results in a more substantial improvement, as reflected by the average RMSE (3.06 % versus 5.06 %). The improvement is especially noticeable in the South Africa, Iberian Peninsula and Greenland cases, where the contribution of the extratropical Atlantic was initially overestimated by 13 %, 12 % and 38 %, respectively. When applying the proposed modification, these biases are almost halved. For the other two cases, the results of the original configuration were already consistent with WRF-WVTs, and remain approximately the same after applying the "RH" modification. In terms of skill score (Table S1 in the Supplement), the "RH" configuration clearly outperforms the original and the "ABL", as the average MAESS is noticeably higher (0.84 versus 0.70 and 0.72). To check if a similar improvement could be achieved by simply changing the minimum specific humidity $\Delta q$ and the time step of the diagnostic tool, Fig. 5c, 5d and 5e present the average RMSE in the "No ABL", "ABL" and "RH" configurations for a range of values of these two parameters. In the case of the specific humidity threshold, the change is minimal, but the modification of the time step can have a considerable effect. Specifically, by reducing it to 3 h the average RMSE remains similar, but by reducing it further to 1 h the results worsen substantially, as evident from the darker colors in the top row of Fig. 5c, 5d and 5e. This aligns with the hypothesis of the effect of specific humidity fluctuations on the underestimation of remote contributions, as increasing the temporal resolution may introduce more noise.

To better illustrate these results, we further examine the moisture sources for two of the selected AR-related precipitation events, specifically, the South Africa and Greenland cases. In Fig. 6 the precipitation sources for these events are depicted using the SOD08 methodology. Fig. 6a and 6c (left) present the results using the basic, "No ABL", configuration, while panels Fig. 6b and 6d (right) correspond to the "RH" experiment. The spatial distributions of these moisture sources reveal a much more pronounced dominance of local sources in the "No ABL" situation, in contrast to the "RH" setup. This is

particularly evident in the Greenland case, where in the basic configuration the moisture source field is essentially over the North Atlantic, as the contribution from this source is overestimated by almost 40 %. Conversely, the situation improves markedly with the "RH" configuration: the moisture source field takes lower values over the North Atlantic and penetrates further into other regions, such as North America. In both cases, the tropical contributions increase and the extratropical ones decrease, coming closer to the results provided by WRF-WVTs (black and red text in Fig. 6). The bias remains after the en-route relative humidity correction, but is much smaller. Analogous results are included in Fig. S9 in the Supplement for the other rainfall events.

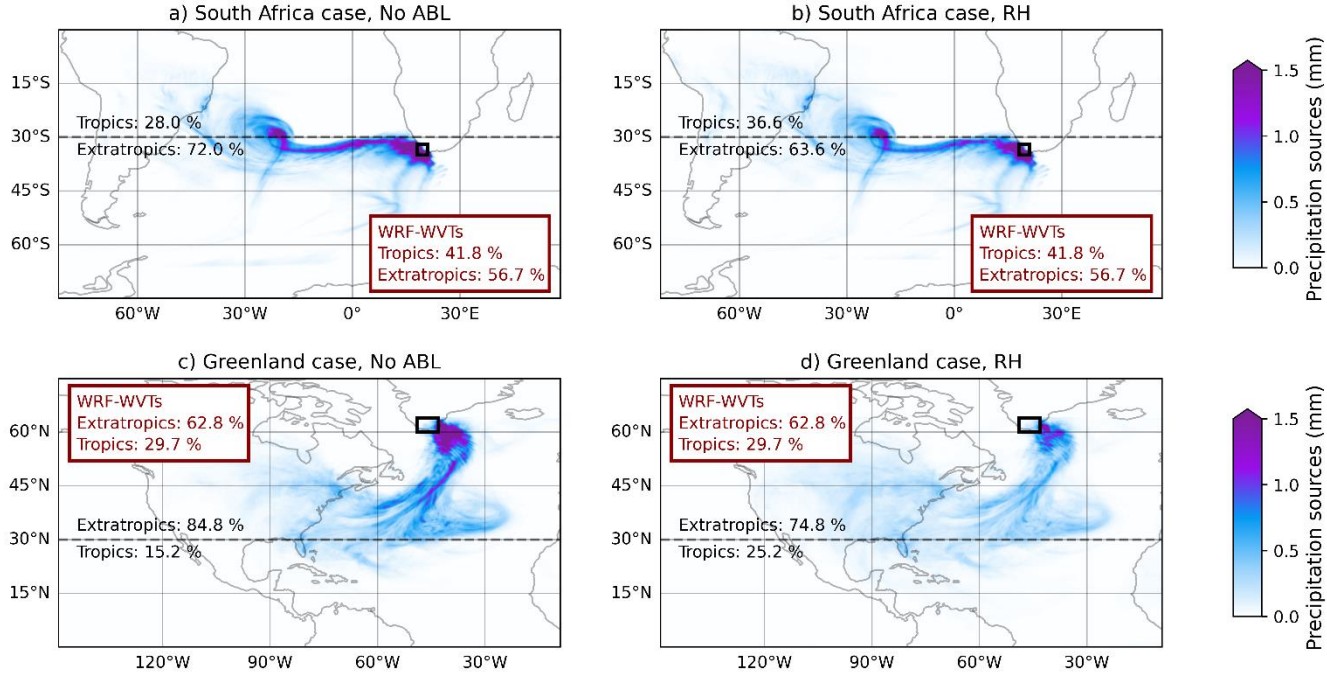

Figure 6: Precipitation sources for the South Africa (a and b), and Greenland events (c and d), computed with the SOD08 moisture source diagnostic. In panels (a) and (c) the most basic configuration is used (No ABL setup), while in panels (b) and (d) we show the results of the "RH" configuration. The fraction of precipitation coming from the tropics and the extratropics is shown in black for each case, and the red box shows these same contributions from WRF-WVTs.

### 3.2.2 DB99

Our analysis indicated that the DB99 methodology, like SOD08, suffers from underestimation of tropical and, in general, remote contributions. In a similar approach to that in the previous section, we take the accuracy of the trajectories generated by FLEXPART-WRF for granted and focus on the capabilities of the Lagrangian tool itself to overcome this limitation. Our hypothesis now is that the way in which the air parcels to be released are selected is behind the biases found. Given that the

initial (that is, at the precipitation event) vertical distribution of particles is proportional to atmospheric humidity, parcels in the lower troposphere are expected to play a more significant role in the calculation of moisture sources for precipitation. However, parcels at these lower atmospheric levels hardly contribute to precipitation since they are generally not oversaturated, i.e. they are outside the cloud level. This factor is crucial, as it is well known that moisture origin can change greatly with altitude (e.g. Hu and Dominguez, 2019). Particles that actually contribute to precipitation could be selected as

in SOD08, taking into account their change in specific humidity. However, the DB99 diagnostic only works with evaporation and precipitable water fields, and to maintain consistency with this, we decided to use another approach, based on finding a threshold height $z_b$, below which it is assumed that Lagrangian particles are not actually contributing to rainfall. The parcels at low levels can obviously rise if an updraft is present and end up contributing to rainfall, but this will be at later time steps, and it is then that they will be considered. Thus, particles are released as usual at the time and location of the

precipitation event, but those below $z_b$ are excluded from the analysis. Moreover, only parcels close to saturation are considered, namely, those with relative humidity higher than 80 % at this initial stage. In short, we conjecture that the basic configuration of this methodology gives too much weight to the lower level air parcels, which usually contain local moisture, and hence the under-estimation of remote sources. For a more technical discussion of this issue, we refer to Sect. 3.2 in the Supplement.


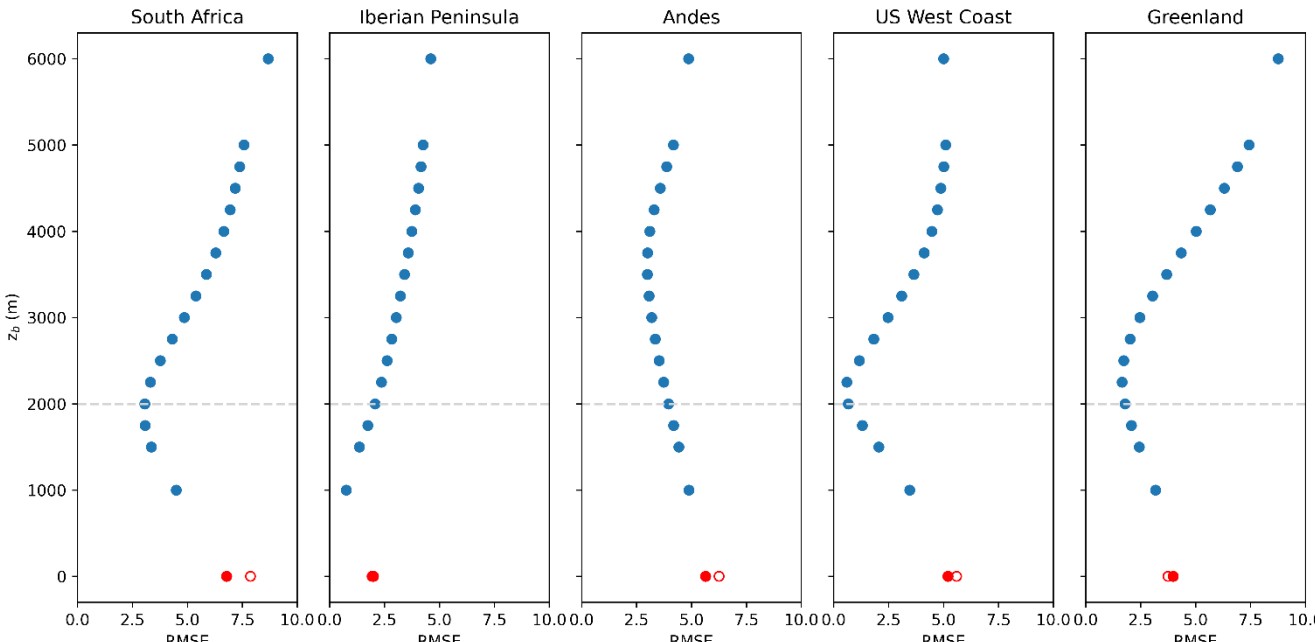

Figure 7: Variation of the RMSE with a threshold height $z_b$ for parcel release in each AR-related rainfall event. True values are from WRF-WVTs, and predicted values are computed with the DB99 methodology, excluding parcels whose initial

height is below $z_b$ and relative humidity below 80 %. In red, the RMSE for the original configuration including all parcels

(empty dots) and applying the relative humidity filter (filled dots). The dashed line indicates the 2 km threshold selected.

In Fig. 7 we show the variation of the RMSE with $z_b$ for the different precipitation events. The original configuration corresponds to the red empty dots, i.e., $z_b = 0$ km, where parcels from the whole atmospheric column are allowed to contribute to the moisture sources calculation. The filled dots at $z_b = 0$ km consider the relative humidity filter to select the

parcels, and this is also applied for all other values of the threshold height. Our findings indicate a initial decrease in RMSE when applying the relative humidity filter, and a continuous decrease as $z_b$ increases, reaching a minimum at a value that is case-dependent. Notably, for the South Africa, US West Coast and Greenland cases, the optimal $z_b$ ranges around 2 to 2.25 km, aligning with the typical lower boundary of mid-level clouds. This altitude, however, could be sensitive to the type of event, meaning that precipitation events not associated with ARs may show a different optimal threshold. The situation for

the Iberian Peninsula and Andes cases is different, as the variation of the RMSE with $z_b$ seems to follow a different pattern. Nevertheless, setting $z_b$ to 2 km results in a decrease in RMSE for all cases, including the latter two. The maximum bias is more than halved in the South Africa, US West Coast and Greenland cases, while for the Iberian Peninsula and Andes cases this maximum reduction is less significant (Fig. S8 in the Supplement). The improvement is further supported by the MAESS (Table S1 in the Supplement), as this metric is higher for all events when the proposed modifications are

introduced. In some cases, such as the US West Coast, the score is exceptionally high, 0.96, indicating a strong alignment with the WRF-WVTs results. On average, the RMSE decreases from 4.64 % to 2.30 %, while the MAESS increases from 0.77 to 0.87. Consequently, we infer that excluding parcels released below 2 km at the rainfall event in the DB99 calculation of precipitation origins is a good approach to rectify the underestimation of remote sources in the case of AR-related precipitation events.


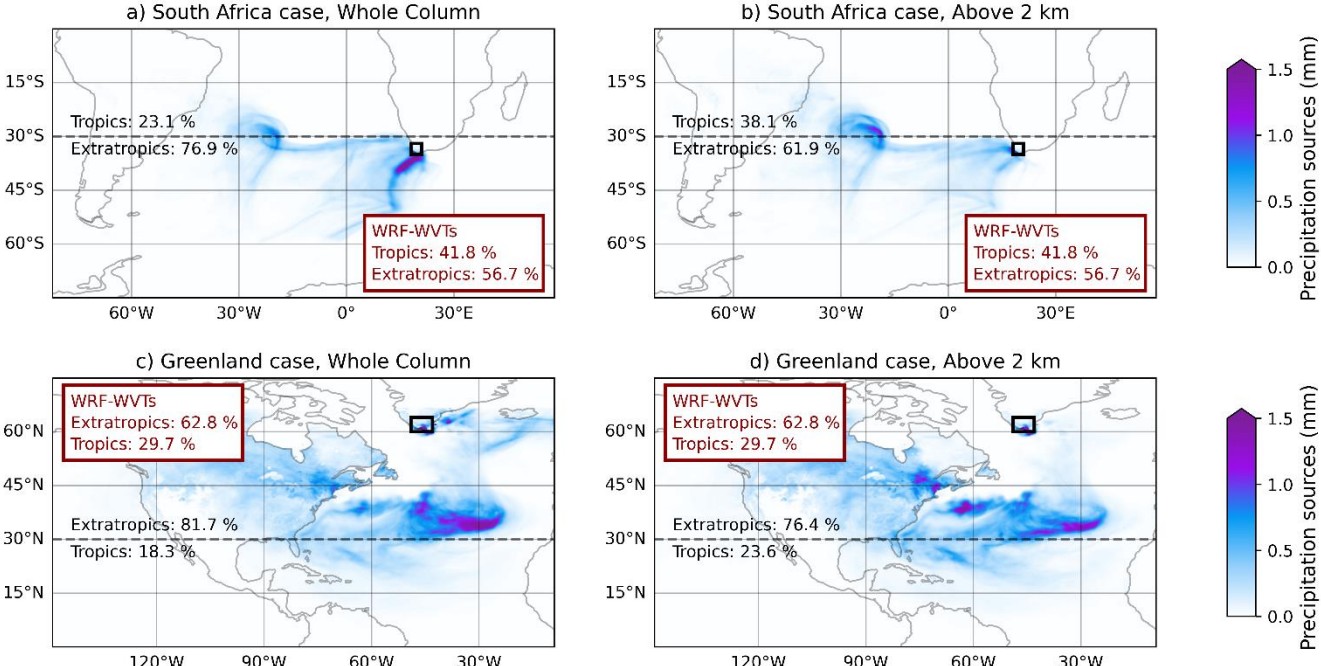

Figure 8: Precipitation sources for the South Africa (a and b), and Greenland events, (c and d), computed with the DB99 methodology. In panels a) and c) the most basic configuration is used, while in panels (b) and (d) parcels below 2 km are not considered. The fraction of precipitation coming from the tropics and the extratropics is shown in black for each case, and in the red box for WRF-WVTs.

As with SOD08, to better illustrate the comparison between the modified and unmodified versions of the DB99 methodology, we examined the spatial distribution of moisture sources for two events, the South Africa and Greenland cases. Figure 8a and c show the results from the basic configuration ("Whole Column"), where all parcels are included in the moisture sources calculation, while Fig. 8b and d represent the scenario where only parcels released above 2 km are considered ("Above 2 km") and the relative humidity filter is applied. We also computed the proportions of precipitation originating from tropical and extratropical regions (black and red text in Fig. 8). In the South Africa case (top), the modified configuration ("Above 2 km") shows less intense moisture uptakes in the oceanic area closest to the target region, indicating a reduced dominance of local sources. The latter is supported by the proportion of rainfall of tropical origin, which increases from 23.2 % to 38.1 %, closely aligning with the "true" value of 43 % provided by WRF-WVTs. In the Greenland case (bottom), we can observe a reinforcement of the contributions from North America and the Tropical Atlantic when excluding parcels below 2 km. Particularly in the case of tropical contributions, there is also a significant improvement, from 18.3 % to 23.6 %, thus approaching the 32.1 % of WRF-WVTs. Obviously, as tropical contributions improve, extratropical

contributions also improve for both cases. It is worth noting that the bias reduction is consistent across different sources for all cases analysed, as explicitly shown in Fig. S8 and S10 in the Supplement.

### 3.3 Extension of the proposed modifications to ERA5 forced simulations

Figure 9 presents the biases in precipitation fraction for both basic and enhanced configurations of the SOD08 and DB99 methodologies, with trajectories generated by FLEXPART-ERA5 using input data from the ERA5 reanalysis. Specifically, Fig. 9a and b display results for the basic configurations, analogous to those in Fig. 4, but with FLEXPART-ERA5 trajectories. The high correlation between both figures (4 and 9) shows that the results are very similar to those obtained with trajectories from FLEXPART-WRF. As in Fig. 4, there is a clear negative bias for tropical sources and a positive bias for extratropical sources. Now these biases are much more evident in the case of DB99 than in SOD08. This is reflected in Tables S1 and S2 of the Supplement, where the average RMSE of SOD08 remains almost unchanged (from 5.20 % to 4.98 %), but that of the DB99 methodology increases from 4.64 % to 5.72 %, mainly due to a worse performance in the Iberian Peninsula and Andes cases. On the other hand, Fig. 9c and d show the biases of the modified versions of both methodologies. The improvements are again evident, as practically all biases are reduced, especially the most important ones. For instance, for SOD08 the biases in the main extratropical sources (North and South Atlantic) are reduced from 15-30 % to below 10 %. In the case of the DB99 moisture source diagnostic the improvements are even more remarkable, as the maximum bias goes from around 20 % to around 5 %. In terms of RMSE (Table S2 in the Supplement) the improvement for SOD08 goes from 4.98 % for the basic configuration to 2.82 % for the modified one, and from 5.72 % to 2.04 % for the DB99 methodology. This improvement is also evident in terms of the MAESS (Table S2 in the Supplement). Overall, the similarity in behavior to that observed with FLEXPART-WRF outcomes suggests that our modifications are also effective when using ERA5 input data.

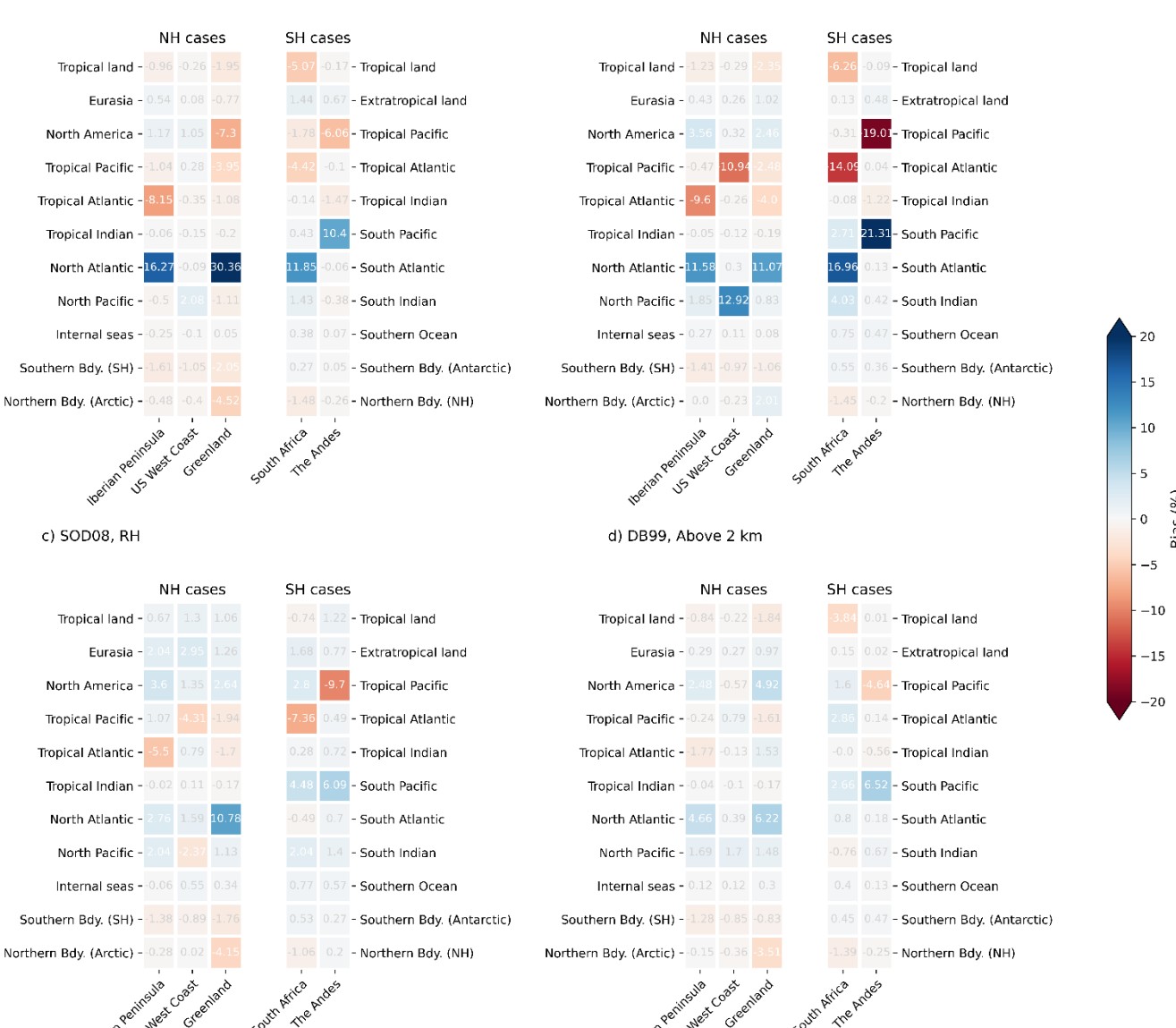

Figure 9: Bias in the precipitation fraction (%) obtained using the basic (a and b) and modified (c and d) configurations of the SOD08 (left) and DB99 (right) diagnostics, for trajectories generated with FLEXPART forced with ERA5 input data (FLEXPART-ERA5). Biases are computed subtracting the "true" outcomes of WRF-WVTs from the corresponding values of the Lagrangian methodologies.

## 4. Summary and conclusions

In this study we have assessed the performance of the SOD08 (Sodemann et al., 2008) and DB99 (Dirmeyer and Brubaker, 1999) methodologies, two of the most used Lagrangian moisture source diagnostics, by comparing their results with the WRF-WVTs tool in the context of AR-related precipitation events. Calculations are performed with the same WRF output data, for which WRF-WVTs results can be considered as reference. The main objective was to obtain a computationally efficient Lagrangian methodology compatible with WRF-WVTs, potentially serving as a substitute for the Eulerian technique in global or climatological applications.

Initially, we evaluated the most basic and commonly used configurations of the SOD08 and DB99 diagnostics. In the case of SOD08, we observed important biases in the estimation of tropical and, more broadly, remote contributions, while there was an overestimation of local sources, especially of the oceanic region adjacent to where the AR makes landfall. These findings are in line with those documented in the literature (e.g. Cloux et al., 2021). Quantitatively, when allowing specific humidity increments above the ABL ("No ABL" configuration), an average RMSE of 5.20 % was obtained, being the average skill score considered in this study (the MAESS) equal to 0.74. When not attributing these increments ("ABL" configuration), we obtained an average RMSE of 5.06 % and average MAESS of 0.71. The similarity in MAESS between these configurations indicates only a minor correction in the "ABL" setup, although for some specific cases, like the Andes case, the improvement is noteworthy. For the DB99 methodology, the initial results showed slightly better agreement with WRF-WVTs, with an average RMSE of 4.64 % and an average MAESS of 0.77. Despite this, there was also a remarkable underestimation of tropical contributions, particularly in certain cases. These findings are also consistent with those reported in previous studies (Insua-Costa et al., 2022; Staal and Koren, 2023).

We then evaluated some physics-based modifications to try to enhance the compatibility of the results produced by both SOD08 and DB99 with those of WRF-WVTs. In the case of the SOD08 diagnostic, we assessed a modification already applied in Dütsch et al., (2018) and Cheng and Lu, (2023): not reducing previous contributions when a specific humidity decrease occurs and the parcel is not close to saturation. Numerically, only decreases in specific humidity that occur when the relative humidity is above a certain threshold are considered for the calculation of moisture sources for precipitation. For the DB99 methodology, as moisture sources are highly dependent on altitude, we proposed excluding from the calculations those parcels released below 2 km at the time and location of the precipitation event, trying to avoid parcels below cloud level, i.e. not contributing to rainfall. In this latter case, we acknowledge that the proposed changes may depend on the type of precipitation event analyzed. Both modifications lead to a notable improvement of the results, as the average RMSE drops to 3.06 % (SOD08) and 2.30 % (DB99), so it is approximately halved. Finally, we also validated our modifications using input data from the ERA5 reanalysis, which is the standard setting for both the SOD08 and DB99 methodologies, and our results show that the proposed modifications also work well in this case.

However, our evaluation framework presents some limitations that could affect the conclusions of our analysis. Apart from its dependence on the type of precipitation event, another drawback is that our selection of source regions for WRF-WVTs may not be fine enough to fully capture the differences between the results provided by this Eulerian tool, which we use as reference, and the Lagrangian estimates of the moisture source field. This is exemplified by the results of the Greenland event in Fig. 6 and 8. In the case of SOD08 (even in the "RH" configuration) there is a significant contribution from the northernmost part of the North Atlantic source (above 45º N), while this contribution is much less relevant for the DB99 methodology. Since the boundary between the "North Atlantic" and "Tropical Atlantic" regions in Fig. 2 is at 30º N, our selection of source regions overlooks this difference. This is a clear limitation of our framework, and a configuration based on smaller source regions delimited by parallels and meridians would be more suitable for comparison purposes. To further validate our results, we have recalculated the RMSEs in Fig. 5 and 7 for all cases with a finer selection of source regions, so that the ocean in which the AR is located for each case is divided into four sections, instead of two. The results, shown in Fig. S12 of the Supplement, demonstrate that the proposed modifications to the Lagrangian tools also provide estimates of the moisture source field that are more consistent with the WRF-WVTs results in this different setup. Although this does not prove that our modifications work even better with WRF-WVTs if we go to a finer scale for the considered sources, it does allow us to conclude that our results are valid for a selection of source regions on a global scale and based on physical criteria (separating the different continents and ocean basins or sub-basins).

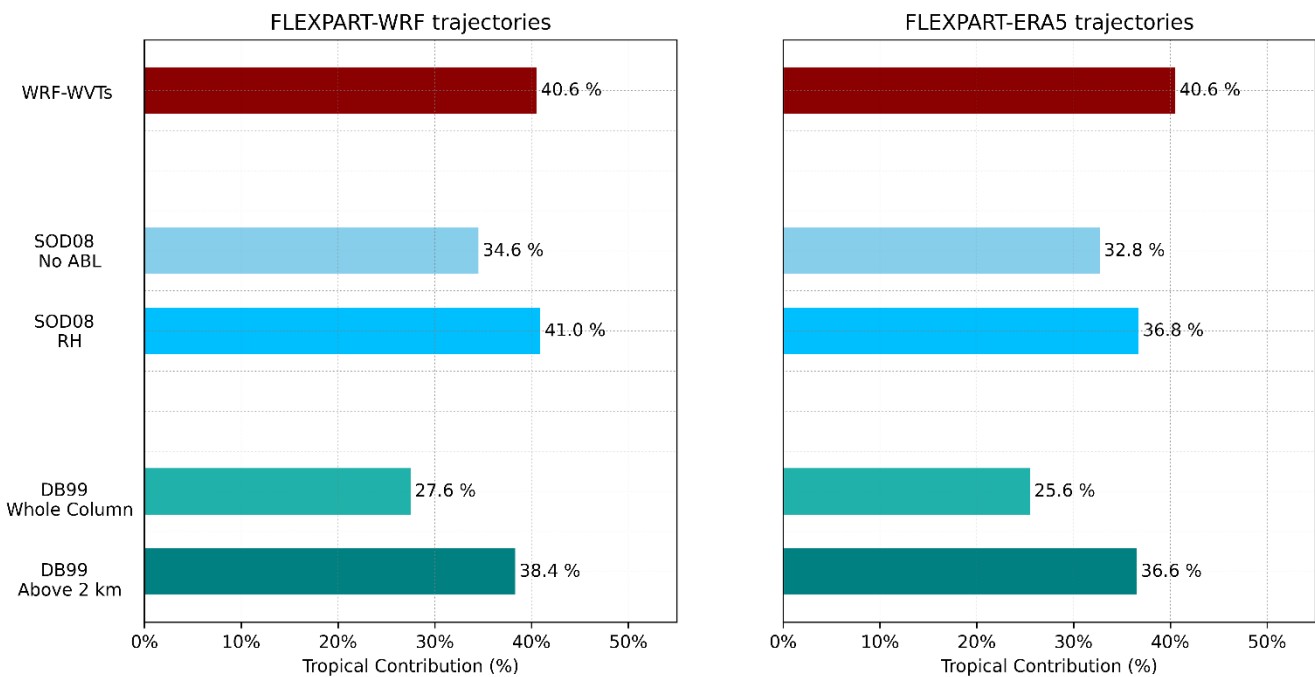

Figure 10: Average tropical contribution (%) obtained using the basic and modified configurations of the SOD08 (bluish colors) and the DB99 (greenish colours) methodologies, for trajectories generated with WRF input data (left) and ERA5 input data (right), in comparison with those obtained with WRF-WVTs (dark red).

Finally, we note that the observed improvements in Lagrangian results correspond to an important rectification of the underestimation of tropical contributions, as highlighted in Fig. 10. The contributions estimated by the SOD08 and DB99 diagnostics, which were initially in the range of 20-30 % on average, now approach 40 %, very close to the WRF-WVTs value of 40.6 %. Therefore, after the modifications introduced, the average results across the set of events lead to similar conclusions for the three methodologies in terms of the partitioning between tropical and extratropical moisture, although there may be substantial differences for specific cases. This is particularly relevant when studying ARs, as the debate remains as to whether or not they are mostly fed by tropical moisture. Moreover, although our analysis focused on specific regions and events, it was not confined to a single geographical area or time period, and the results consistently pointed to the same conclusions. This suggests that our findings are broadly applicable to precipitation events (at least those associated with ARs) anywhere on the globe. In this scenario, we conclude that these Lagrangian moisture source diagnostics can serve as viable alternatives to WRF-WVTs or other similar methods, particularly in global or climatological studies where computational efficiency is a priority. Most interestingly, the results of the different methodologies have converged by introducing only two simple and physically-based (non-artificial) modifications, which suggests that further validation and modifications of the same kind in the future could lead to an extraordinarily high degree of agreement among them.

**Data availability.** ERA5 data on single and pressure levels used as initial and boundary conditions of WRF simulations were downloaded from the Climate Data Store, https://cds.climate.copernicus.eu/datasets/reanalysis-era5-single-levels / and
https://cds.climate.copernicus.eu/datasets/reanalysis-era5-pressure-levels (last access: 30 May 2025). ERA5 data on model levels were downloaded using the flex_extract software (Tipka et al., 2020). The source code for FLEXPART and FLEXPART-WRF can be accessed through https://www.flexpart.eu/ (last access: 30 May 2025), while WRF-WVTs source code can be accessed through https://github.com/damianinsua/WRF-WVTs (last access: 30 May 2025). Finally, analysis scripts to post-process FLEXPART and FLEXPART-WRF simulations are published on GitHub
https://github.com/crespoalfredo/MoistureTracking_Comparison_ESD (last access: 30 May 2025).

**Author contributions.** D.I.C., E.H.G., C.L. and G.M.M. conceptualized the experiment. A.C.O. and D.I.C designed the methodology. A.C.O. performed the simulations, analyzed the data, created the figures, and wrote the first manuscript draft. D.I.C., E.H.G., C.L. and G.M.M. contributed with ideas, interpretation of the results, and manuscript revisions.

**Competing interests**. The authors declare that they have no conflict of interest.

**Acknowledgements.** Funding comes from the Spanish Ministerio de Ciencia e Innovación RIESPIRO (PID2021-128510OB-I00 to G.M.M. and A.C.O.) and LAMARCA (PID2021-123352OB-C32 to C.L. and E.H.G) projects, funded by MICIU/AEI/10.13039/501100011033 and FEDER, "Una manera de hacer Europa". D.I.C. is supported bya BOF postdoctoral fellowship from Ghent University (BOF.PDO.2024.0026.01). E.H.G. and C.L. acknowledge support from the María de Maeztu project CEX2021-001164-M funded by the MICIU/AEI/10.13039/501100011033. A.C.O. acknowledges
Xunta de Galicia for a predoctoral grant (Programa de axudas á etapa predoutoral 2023, ED481A-2023-192). Computation took place at CESGA (Centro de Supercomputación de Galicia), Santiago de Compostela, Galicia, Spain.

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
