# Peer review of "Simple physics-based adjustments reconcile the results of Eulerian and Lagrangian techniques for moisture tracking in atmospheric rivers"

_Earth System Dynamics, 2024_

## Author Comment (AC1)

5    General comments

This study investigates the uncertainty in precipitation source regions estimated by three different modeling approaches. Precipitation sources estimated by the online Eulerian-based WRF-WVT method are taken as the reference, against which estimates from two offline Lagrangian-based methods are compared: the WaterSip and UTrack
10   methods. Both methods are found to exhibit biases in the estimated precipitation sources compared to the reference data set, in particular showing sources to be geographically closer to the precipitation than the more remote sources estimated by the reference. The study then tests a structural modification to each of the WaterSip and UTrack methods and finds bias is reduced and precipitation sources are made
15   geographically closer to those of the WRF-WVT reference. A key conclusion of the study is that the Lagrangian methods can serve as viable alternatives to the more computationally-expensive WRF-WVT method. The study is well-defined, well-written and the conclusions logically follow the results. In particular, the authors are to be commended for detailing the structural differences between the models. The main area
20   of improvement needed is the clarification of the proposed modifications to the Lagrangian models, and their resulting evaluation against the reference dataset.

Thank you very much for your comments, which we think have substantially improved the article.  Please, find below the responses to your comments.

Specifically, the modification of the UTrack model appears to contain two changes: (1)
25   only parcels released from above 2km may be used for tracking, and (2) of those parcels, only those with relative humidity above 90% are subsequently tracked. It is unclear which modification dominates the reported changes to precipitation sources relative to the WRF-WVT sources. Of more minor importance, it is unclear why a higher relative humidity threshold is applied to the UTrack model compared to the WaterSip
30   model; this choice of model modification needs to be clarified.

It is true that the proposed modification of UTrack contains two changes, releasing parcels from above 2 km and retaining only those with relative humidity higher than 80 %, and we agree that we did not test each one separately. Because of this, in the revised version of the manuscript we will make it clear that there are two changes, and
35   we will test them in two steps. First, we will apply only the relative humidity filter and evaluate the improvement. After that, we will repeat our experiment by changing the threshold for the release height. This will be reflected in a modified version of Fig. 7, where we add another red dot resulting from applying only the relative humidity filter. Regarding the choice of the relative humidity threshold, there was a typo in the
40   manuscript, as they should all be 80 %.

The modification of the WaterSip model, requiring parcels to have a minimum relative humidity of 80% immediately before a decrease in specific humidity, needs to be explained more clearly. It needs to be made clearer what the exact problem is with the way WaterSip reduces parcel specific humidity en route, and how applying an 80%
45   threshold of relative humidity helps.

The explanation of the WaterSip model in the methodology will be rewritten. Specifically, we will omit some information that can be found in the literature, particularly the explanation of the basic configuration of WaterSip, well documented in Sodemann et al., (2008). We will focus more on different modifications of this diagnostic tool that have been used in previous studies (Fremme and Sodemann, 2019; Dütsch et al., 2018), as they relate to the problem of specific humidity fluctuations that we investigate later. Moreover, in Sect. 3.2.1 we will clarify why these fluctuations may penalize remote contributions (see our response to your specific comment for a more detailed explanation). Under these assumptions, the application of an 80 % threshold of relative humidity helps to address this problem, as it is a simple approach to filter out non-physical specific humidity decreases (not associated with precipitation), and this will also be included in the second paragraph of that section.

Specific comments:

L47: Which problem is being referred to here?

We refer here to the problem of the origin of moisture in ARs. In the revised manuscript we will make it explicit for clarity, by replacing this sentence by "However, the problem of the origin of moisture in ARs is not yet completely closed, as reflected in the definition of AR given in the Glossary of Meteorology, where it is indicated that the sources of moisture can be tropical and/or extratropical (Ralph et al., 2018)".

L55/60: Here it is asserted that Eulerian approaches are more accurate than Lagrangian approaches. I do not think it is true that, in general, Eulerian tracing approaches are considered to be more reliable than Lagrangian approaches in accurately estimating precipitation sources. Perhaps you mean *online* Eulerian water vapor tracers are considered more accurate? If this is the case, I suggest rephrasing to clarify. Furthermore, if Lagrangian approaches are asserted to contain "more uncertainty", than these uncertainties need to be outlined. Relatedly, I think it is important to be careful about asserting that WRF-WVTs can be "considered as synthetic observations". There needs to be some evidence that WRF-WVTs can in fact accurately represent real observations, for example through comparison with satellite observations of atmospheric moisture. If this or a similar type of evaluation has been done, please refer to it here. Otherwise, I would tone down the language by changing the words "considered as synthetic observations" in L63 (also in L436) to "used as a reference".

We agree with the reviewer that Eulerian approaches are not more accurate than Lagrangian approaches in general. Because of this, we will rewrite this sentence and clarify that it is the online water vapor tracers that we consider to be more accurate. Regarding the uncertainties in Lagrangian models, we agree that we did not go deep enough. In the revised version, we will emphasize that the uncertainty comes from a number of hypotheses and parameters, which are precisely those explored in this study. Finally, we also agree that it is appropriate to soften the language and speak of "reference" instead of synthetic observations when referring to the results of WRF-WVTs, and this will be reflected in the revised manuscript by changing "to be considered as synthetic observations" to "to be considered as reference".

L145: Is the specific humidity assimilated from ERA5 like the evaporation field? Does the WRF model close the water balance if ERA5 evaporation is assimilated?

No, the specific humidity is not assimilated from ERA5 like the evaporation field. The WRF model closes the water balance in this case, as we are only changing the surface moisture flux simulated by WRF by the surface moisture flux in ERA5, and the other moisture fluxes are updated accordingly by the model itself.

L155: While the manuscript makes it clear that parcel trajectories are calculated using WRF data in the first case, and ERA5 data in the second case, it is a little unclear which dataset was used to calculate the moisture contribution for each Lagrangian model. From reading section 2.3, I interpret that in the first case, "FLEXPART-WRF", WaterSip reads the specific humidity field from WRF, and UTrack reads the precipitable water field from WRF but the evaporation field is ERA5 data assimilated into WRF. In the second case, "FLEXPART-ERA5", I interpret that both WaterSip and UTrack read all fields from ERA5. If this is not the correct interpretation, please clarify.

This is exactly the correct interpretation. To make this clearer, we will explain it better in the last sentences of the first paragraph of Sect. 3: "Finally, in Sect. 3.3 we test the introduced modifications when the trajectories are generated by FLEXPART, with input data from the ERA5 reanalysis. In this case the other fields required by the diagnostic tools are also obtained from the same reanalysis, not from WRF simulations."

L172 & L210: The Dirmeyer and Brubaker approach is also used by other studies, whose moisture tracking method is very similar to UTrack, e.g. Holgate, C. M., J. P. Evans, A. I. J. M. van Dijk, A. J. Pitman, and G. D. Virgilio, 2020: Australian Precipitation Recycling and Evaporative Source Regions. Journal of Climate, 33, 8721–8735, https://doi.org/10.1175/JCLI-D-19-0926.1. Similarly, the WaterSip approach is also used by other studies, e.g. Cheng, T. F., and M. Lu, 2023: Global Lagrangian Tracking of Continental Precipitation Recycling, Footprints, and Cascades. Journal of Climate, https://doi.org/10.1175/JCLI-D-22-0185.1. Though these specific methods are not formerly evaluated here, it would be pertinent to acknowledge them.

We were aware of the problem with the nomenclature for the Dirmeyer and Brubaker, (1999) methodology, but did not know how to solve it. In the revised manuscript, we will acknowledge these other studies and refer to the diagnostic tool as "the Dirmeyer and Brubaker, (1999) methodology" instead of "UTrack".

Figures 3 and 4: it would be helpful to the reader if these figures could be placed side by side for easier comparison. Is it possible to combine the two figures into one?

As both figures refer to different subsections, we do not think it is possible to combine them into one. However, we will ask for them to be placed one after the other.

L230: To make it easier for the reader to interpret the error scores, it would be helpful to add a sentence linking each score with a physical meaning, e.g. a higher value of MAESS refers to a more accurate comparison with the reference dataset.

We agree with the reviewer that we did not make clear the interpretation of the Mean Absolute Error Skill Score (MAESS). In the revised version of the manuscript, we will explicitly mention that a closer value of MAESS to 1 means a more accurate aligning with the reference dataset, by adding "as usual with a skill score, the closer to 1 means that the results of the Lagrangian model are closer to those of WRF-WVTs" once the MAESS is introduced.

L303: To make it clearer to the reader, it would be helpful for the accumulation over time to be shown with a simple example. As the manuscript currently reads, it is unclear what the problem with the WaterSip method is.

To better explain how the error accumulates over time, we will include another iteration of our simple example (a couple of non-physical increases and decreases): "If another non-physical decrease occurs, this value is updated to 1.95(1-0.05/2.0)=1,90 g kg$^{-1}$". Moreover, we will make explicit that early contributions are more penalized, as they are affected by many more potential non-physical fluctuations, by adding "as the error caused by a single fluctuation affects all previous contributions, so the early moisture uptakes will be affected by many more non-physical changes".

L378: The original configuration of UTrack appears to release parcels from a random, humidity-weighted vertical level, indicating the starting parcel levels will be in the lower part of the troposphere. Yet here, and in Figure 7, it is indicated that the starting parcel level is 0km. Was the starting parcel height set at 0km in this study, or was a random, humidity-weighted vertical level used as in the original model? Further, did this study use a random, humidity-weighted vertical release height and simply ignore those parcels starting below 2km, or was the release height set at a constant 2km level in the modified case?

We agree with the reviewer that it is not clear how parcels are released vertically, and this should be clarified, as it is a key point of the modification we are proposing. In our case, parcels are vertically released following the density profile, using the domain-filling option of FLEXPART. In the case of the Dirmeyer and Brubaker, (1999) methodology, parcels are released following the humidity profile of the atmosphere. Thus, to match our approach with the original one, we need to weight the moisture origins for each parcel using their humidity. This additional (and important) information will be included in the methodology in the revised version of the manuscript. Specifically, we will add a sentence at the end of the first paragraph of Sect. 2.3 to explain how parcels are vertically released in FLEXPART: "In both cases parcels are released using the domain filling option over the black boxes in Fig. 1, such that they are vertically distributed following the density profile". When explaining the Dirmeyer and Brubaker, (1999) methodology, we will also add another sentence for clarification: "since in our simulations parcels are vertically released following the density profile, we weight the contribution of each parcel by its specific humidity to match the Dirmeyer and Brubaker, (1999) methodology".

Regarding the modification we propose, we simply ignore parcels starting below 2 km, being the rest released as usual. This will also be clearer in Sect. 3.2.2 of the revised manuscript, as we will add "particles are released as usual, but those below $z_b$ are excluded from the analysis" when introducing the modification.

L416: Can you provide some reasoning as to why WaterSip is superior to UTrack when using ERA5 data?

This is probably related to the different number of vertical levels (38 for WRF versus 70 for ERA5) and, in particular, to the extent to which the different methods are sensitive to having more or fewer levels, but this is something we do not know for sure and would require further analysis.

L475: The statement that the Lagrangian methods can serve as viable alternatives for WRF-WVTs is a key conclusion of the study. I would suggest including this conclusion in the abstract.

We thank the reviewer for this suggestion. In the revised version of the manuscript, this will be one of the main points of the abstract. Specifically, the last sentence of the revised abstract will be "Although these modifications may need to be adjusted for

other type of precipitation events, our results demonstrate that Lagrangian techniques are a viable and compatible alternative to Eulerian water vapor tracers, and that the main discrepancies between the different methodologies can be derived from the obviation of certain physical considerations."

190

Technical corrections

Figure 1: it would be helpful if the subplots each had a title describing their geographic location, e.g. "South Africa". These location labels can then be added to Table 1 to make it easier for the reader to associate the numerical description with a real-world
195    location.

This figure will include the corresponding geographic labels in the revised version.

Figure 2: "Tropical Indic" should perhaps be "Tropical Indian" (same issue applies to later figures). Also some parts of the world are classed as "Tropical land" when they are in fact desert regions (e.g. northern and southern Africa, central Australia, Arabian
200    peninsula). To avoid re-running the model with different regions, I suggest touching on the implications of this classification in your results.

We have corrected all figures changing "Indic" by "Indian" where it corresponds. Regarding the classification of desert regions as "Tropical land", we understand that it could lead to confusion if we were to conclude that a certain amount of precipitation
205    comes from these areas, but this is not the case, as we are referring to the source as a whole. In any case, we do not consider it incorrect, since deserts in the tropics are still "tropical lands", so we decided to keep it as it was. Also, note that removing deserts from the sources classified as "tropical lands" would not affect our comparison at all.

L165: Should "Except for the position and the…" be "Except for the position of the
210    parcel and the …"?

The typo has been corrected.

**References**

Dirmeyer, P. A. and Brubaker, K. L.: Contrasting evaporative moisture sources during
215    the drought of 1988 and the flood of 1993, J. Geophys. Res. Atmospheres, 104, 19383–19397, https://doi.org/10.1029/1999JD900222, 1999.

Dütsch, M., Pfahl, S., Meyer, M., and Wernli, H.: Lagrangian process attribution of isotopic variations in near-surface water vapour in a 30-year regional climate simulation over Europe, Atmospheric Chem. Phys., 18, 1653–1669, https://doi.org/10.5194/acp-
220    18-1653-2018, 2018.

Fremme, A. and Sodemann, H.: The role of land and ocean evaporation on the variability of precipitation in the Yangtze River valley, Hydrol. Earth Syst. Sci., 23, 2525–2540, https://doi.org/10.5194/hess-23-2525-2019, 2019.

Sodemann, H., Schwierz, C., and Wernli, H.: Interannual variability of Greenland winter
225    precipitation sources: Lagrangian moisture diagnostic and North Atlantic Oscillation influence, J. Geophys. Res. Atmospheres, 113, D03107, https://doi.org/10.1029/2007JD008503, 2008

---

## Author Comment (AC2)

The authors present a study focused on the comparison between Eulerian and Lagrangian approaches to trace moisture and to identify the evaporation sources of precipitation. Using a regional model simulation with water tagging as a reference, they then evaluate two Lagrangian offline approaches in that framework for a set of Atmospheric River events from different regions. Two tunings are proposed to reduce a general bias towards shorter transport distances in Lagrangian methods. The study is overall interesting, presented clearly, and well written.

Thank you very much for your detailed review. We believe that the modifications you suggest have improved the manuscript and the robustness of our analysis. Please, find below the responses to your comments.

However, the fairly coarse choice of tagging regions, as well as the exclusive selection of AR cases introduces limitations that are currently not well addressed. A more careful and balanced discussion of the results and implications from this study are thus advised. I also see further issues with the proposed tuning and with regards to some parts of the literature detailed below that the authors should address when preparing a revised manuscript.

We are aware of some limitations of our study, particularly regarding the exclusive selection of AR cases. Because of this, we will soften the language in the abstract and conclusions and recognize them in the discussion. Moreover, we conducted additional experiments to address some of the issues you encountered, which will appear in the response to your major and specific comments.

Major comments:

1. Coarse definition of tagging regions. The authors subdivide the hemispheric land and ocean into 9 sectors, separated along 30 N and S. This allows only for a very coarse distinction between ocean basins and continental areals and the boundaries. As the Lagrangian diagnostics are showing, the majority of sources are located at different regions within the same ocean basin. As a consequence, the RMSE computed here only picks up the outermost differences. An example for this is seen for the Greenland AR, where the structures in the North Atlantic region widely differ between the two Lagrangian approaches. The current tagging setup misses these differences entirely, and exclusively focusses on the fringe of the moisture sources. There are probably two ways to approach this deficiency: One is to increase the number of tracer subdivisions depending on every case, adding complexity to the study, but providing more sharpness in the tagging simulation (e.g. using a setup similar to Sodemann and Stohl, 2013). The other way is to openly address this deficiency in the study design, and adjust the discussion to be more nuanced, and formulate the conclusions more carefully.

We agree with the reviewer that our selection of source regions overlooks some differences that may exist between both Lagrangian approaches, this being evident in the Greenland case. This selection is based on the fact that for many applications in the field of moisture sources we are only interested in the basin as a whole and not so much in the contribution of the different sub-areas of the basin. An example is the calculation of continental recycling; in that case we are interested in the contribution of the continents as a whole. Another recurring example with atmospheric rivers is the separation between tropical and non-tropical moisture inputs, for which we only need 50 two sources.

However, for comparison purposes, we acknowledge that this division of sources may substantially affect the results, so we have conducted an additional experiment. Specifically, for each AR case we divided the tropical and extratropical ocean where it is located in two more parts (thus four sources for each basin) and repeated the 55 comparison with WRF-WVTs using only six source regions for each case: the four sources for the corresponding ocean, the next most important source region (continental in all cases except for the Andes) and the rest of the world. This is a slightly coarser configuration than that of Sodemann and Stohl, (2013), but it overlooks much less differences than the original one. For example, in the Greenland case this 60 new setup reveals that the correct distribution of moisture sources - in relation to WRF-WVTs - is that of UTrack, as shown below (UTrack is now referred to as DB99). In particular, WaterSip overestimates the contribution of the northernmost part of the basin, i.e. the part closest to Greenland.

[Figure]

However, what we found is 65 that the main conclusions of our study do not change with this new configuration; when the errors are averaged to obtain a single deviation per event (e.g. Fig. 5), the values obtained are very similar. Thus, this experiment validates our results, and will be included in the Supplement, where we will add the following figure with the results of Fig. 5 and 7 for this new selection of source regions. Additionally, we will briefly refer to 70 this experiment in the main manuscript just after discussing the appropriateness of the selection of source regions.

[Figure]

2. Biased selection of cases. The study includes five AR cases from different parts of the world. All cases are thus potentially related to a large amount of long-range transport. While this selection in itself is no matter of concern, proposing a general tuning of the Lagrangian methods based on a selection of long-range transport cases only is problematic, as it may introduce biases during cases of more local precipitation sources (e.g. convective summertime precipitation, weaker precipitation cases). The focus on AR cases only, and the limitations following along with that, should be more clearly highlighted in the title, abstract, and conclusions.

We agree with the reviewer and therefore propose the following changes:

- New title: "Simple physics-based adjustments reconcile the results of Eulerian and Lagrangian techniques for moisture tracking in atmospheric rivers".
- Abstract. We will rewrite the last sentence of the abstract: "Although these modifications may need to be adjusted for other types of precipitation events, our results demonstrate that Lagrangian techniques are a viable and compatible alternative to Eulerian water vapor tracers, and that the main discrepancies between the different methodologies can be derived from the obviation of certain physical considerations".
- Conclusions. We will again recognize that the proposed modification to UTrack (specifically, the threshold for release height) may depend on the type of precipitation event when discussing the achieved improvements in RMSE.

3. Proposed tuning to the WaterSip method. The authors propose to introduce a relative humidity threshold in the WaterSip method during the identification of precipitation/moisture loss events en route. While such a proposal seems physically plausible at first, there are some downsides as well. Importantly, a moisture loss can be due to one of two reasons, either removal of water vapour from the atmosphere due to condensation and precipitation, or due to the mixing with drier air masses. The second case will be necessarily ignored in unsaturated situations if a relative humidity threshold is introduced as proposed here. Ignoring the lowering of specific humidity due to mixing can then lead to an over-accounting of the moisture sources, i.e. a larger amount of uptakes are assigned to the specific humidity of the air parcel that are contained within. Duetsch et al. 2018 proposed a distinction between mixing events and rainout events. However, both types of situations still need to be part of the accounting method to be physically plausible.

We were not aware that the modification we are proposing had already been introduced in a previous study. Therefore, in the revised version we will acknowledge that it is an existing and used modification to the WaterSip method. This will be recognized both in the methodology and in the conclusions. To be more explicit, in the last case we will do it at the beginning of the third paragraph: "In the case of WaterSip, we assessed a modification already applied in Dütsch et al., (2018) and Cheng and Lu, (2023)". Regarding decreases in specific humidity due to dry air mixing, in our opinion, precisely what the relative humidity filter does is to prevent these decreases from being attributed to precipitation, which would be incorrect.

A more conventional tuning of the WaterSip method is to change the specific humidity thresholds and the time step. While the authors have tested different time steps, the specific humidity threshold has been set to a quite low value compared to literature (a common value is 0.2 g kg-1 6h-1). The specific humidity threshold will have a similar effect as the RH threshold, and is justified by interpolation errors in the offline approach. Can the authors report how sensitive the moisture sources are, and thus the RMSE values to a variety of changes in the specific humidity threshold?

Following the reviewer's suggestion, we have performed several experiments in which we modify the specific humidity threshold and found almost no changes. Regarding the time step, there is a clear dependence on that parameter, as explicitly shown in the Supplement. In the revised version, we will change the standard setup to be that of the literature (0.2 g kg$^{-1}$ in a 6 h time interval) and include in the main manuscript the dependence of the average RMSE on both the specific humidity threshold (setting dq to 0.01, 0.05, 0.1, 0.2 and 0.3 g/kg) and time step (1 h, 3 h and 6 h). This will show that the optimal choice is that of the literature, although the changing dq is not very relevant. All this additional information will be included in a revised version of Figure 5, by adding three more panels reflecting the dependence of the average RMSE on both parameters (one for each of the No ABL, ABL and RH configurations).

Ultimately, I think one also has to acknowledge that offline trajectory methods do have
their inherent limitations, both from the computation of trajectories, and the specifics of the moisture source diagnostic, which are sort of the price for the lower computational expense, and the more detailed spatial information on the source location. Knowing different methods' limitations may be in the end more valuable than tuning methods towards an expected or desired outcome for a specific type of cases. Maybe the
authors could reflect on this perspective in their discussion and conclusions?

We agree that knowing the limitations of the tool being used is always essential in order to make a proper interpretation of the results. However, we believe that, in the long run, merely describing the limitations of tools does not make them any better, which is precisely what we are trying to do here. Finally, it is worth stressing again that
the changes we are proposing have a physical basis, not a mere tuning of methods to match each other.

4. Title, abstract and conclusions appear too wide-ranging. As partly commented in the points above, the present study has limitations from the method design with respect to tracer setup and case selection, and the tuning of Lagrangian methods can lead to
inconsistencies in the method. The discussion throughout the manuscript should be more nuanced and balanced by taking up these limitations. In particular the abstract is now formulated in a very definite, concluding language, which does not seem justified in the light of the limitations mentioned above. The title also suggests to a superficial reader that studies should generally apply the proposed tunings, but their general
validity is questionable, or is at least not generally established. In particular the aspect of AR case selection could be included in the title. The study design with coarse tagging regions does in my view not 'reconcile' different approaches, but is rather a tuning using a particular choice of parameters. Maybe the title could be rephrased in terms of sensitivity, and mention the importance of long-range transport for the
examined cases?

We have already mentioned how abstract, title and conclusions will be reformulated to reflect that our study is focused on AR-related precipitation events. Regarding the selection of source regions, we believe that the additional experiments we have conducted, together with the new figure in the Supplement and its corresponding
discussion in Sect. 3.2.2 give validity to our results. Nevertheless, in the same discussion we will explicitly acknowledge that a different choice of source regions could be better for comparison purposes.

5. Use of literature. There are some citations of previous tagging studies that are missing or could be valuable to add. There are also some wrong citations (Lagrangian
method cited in Eulerian context). These publications are listed in the detailed comments below.

We have updated the bibliography and citations with your comments, and this will be reflected in the revised version of the manuscript. Thank you very much for that.

**Detailed comments**

L. 21: What unit does the RMSE have, is this in percent, or a fraction?

The RMSE has the same units as the precipitation fractions, we are expressing them as percentages. We will correct the manuscript indicating these units where appropriate.

L. 22: "narrowly superior performance": How significant are the differences of less than 1 (%?) between both methods considering all sources of uncertainty?

We agree with the reviewer that this difference is not significant, but this statement is no longer in the new version of the manuscript.

L. 23: Maybe clarify that this is a relative improvement, since the RMSE appears to
have the same units. The 50 % relative improvement could be misleading, both because the untis are the same as for the RMSE, and given the overall quite small RMSE difference. Can the overall result be presented more balanced and objective here?

In order to avoid misunderstandings, we will use the mean absolute error skill score
(MAESS) to present the main results of our study in the abstract.

L. 24: I think this conclusion statement is going too far. The selection of cases and limitations in the setup does not allow this conclusion. Expressed more neutrally, the sensitivity test and tuning performed here increase the amount of long-range transport detected from the Lagrangian methods. There is not sufficient evidence presented
supporting that the tuning is valid generally in all cases. Maybe instead it could be emphasized that the overall approach of using a Eulerian tagging setup to validate Lagrangian methods is promising, but needs further refinement for generally valid modifications.

We agree with the reviewer that we should soften the language, especially given our
focus on AR events. Because of this we will modify the abstract and mention that the changes we are proposing may need to be adapted for other types of precipitation events. Moreover, we will emphasize that Lagrangian moisture tracking techniques are a real alternative to Eulerian water vapor tracers. All these changes were already highlighted in our response to the major comment 2.

L. 35 and elsewhere: It is customary to sort references by year of publication. Consider adding Yoshimura et al., 2004 to this list. Sodemann and Stohl (2009) is a Lagrangian study, did you mean to cite here Sodemann et al. (2009)?

We thank the reviewer for its clarification, we meant to cite here Sodemann et al., (2009). We will also add Yoshimura et al., (2004), and sort references by year of
publication.

L. 36: "Lagrangian transport models": Lagrangian transport models are the general category of models that simulate airmass transport. To be more specific to the case here, consider rephrasing as "Lagrangian moisture source diagnostics".

We will rewrite "Lagrangian transport models" as "Lagrangian moisture source
diagnostics" in the revised version of the manuscript.

L. 39: I do not know of an existing online implementation of a Lagrangian moisture source diagnostics. The offline/online distinction can however be made regarding the tagging and Lagrangian methods.

We only state that there exists another possibility to classify moisture source diagnostics, and this allows us to introduce the difference between offline/online methods, which is used later.

L. 40: "most academics often use": this point is debatable, there exist a range of studies that do make such comparison efforts.

We agree with the reviewer that there are different studies that do make such efforts.
However, it is also true that the majority of studies use a single model.

L. 41: "results can be highly discrepant": Winschall et al., 2014 does not provide highly discrepant results, at least that is not what is said in this paper. Please rephrase to do justice to the actual state of the literature, and to better clarify the intent and actual novelty of this study. In this context, please also consider the book chapter of
Sodemann and Joos (2021).

We will rephrase "results can be highly discrepant" to "results may not be in agreement", as we understand that although Winschall et al., (2014) does not provide highly discrepant results, they do show that tropical contributions calculated in one way or another, for example, may be clearly different.

L. 46: Consider adding references to the original AR studies in this context.

We are going to add a reference to Zhu and Newell, (1998) following the reviewer's suggestion.

L. 47: This statement does not seem to do justice to the existing literature. See Sodemann and Stohl (2013) for a tagging study focused on AR events, as well as Stohl
et al. (2008) for a study with Lagrangian methods. There are also a range of studies from other regions and locations (e.g. Terpstra et al., 2021, Bonne et al., 2015). Please update this statement in light of existing studies, and clarify what this study adds to the existing literature. Please also take notice of the book chapter about AR moisture budgets (Sodemann et al. 2020). What is meant by "go beyond the identification of
moisture sources to quantify them?"

We will reformulate this paragraph to include the references that the reviewer provides, in order to improve the presentation of the state of the art on moisture sources in atmospheric rivers. Since the main objective of our study is to compare two Lagrangian moisture tracking methodologies with WRF-WVTs, we will delete "go beyond the
identification of moisture sources to quantify them", as our study is not focused on providing additional insights on moisture sources for precipitation in ARs.

L. 55: There are two aspect here that are a little bit mixed together. One is that the tagging simulation is also only a model representation of the actual water cycle in nature. At the grid resolution of the model (here 20 km horizontally), a large spectrum
of the processes affecting the water cycle are parameterized. I assume that also a deep and shallow convection parameterisation (which one?) has been used in the Eulerian model simulation. Obviously, the model will thus not be identical with nature. However, the approach and argument of the present study is, as I understand it, that the tagging water cycle and the Lagrangian methods are internally consistent, even if
the tagging results differ from nature. This is important, as the authors write, since the source information that is being sought after is not directly available from observations.

This statement, although debatable, is based on the fact that Eulerian water vapor tracers do not suffer from some of the pontentially more conflicting approximations that other methods suffer from. To give a couple of examples, WAM2Layers suffers from the "well mixed assumption", and UTrack lacks a convection parameterization. To give another, in WaterSip, at least in the version we use, phase changes along the parcel trajectories are not taken into account. All these processes are considered in Eulerian water vapor tracers, so it is reasonable to assume that, if implemented correctly in the corresponding model, they should provide more realistic results. As for the parameterizations in the WRF simulations, we will explicitly state them in the methodology of the revised version of the manuscript, and they will also be mentioned in the response to a later comment.

L. 57: Another important limitation of the tagging approach, which also becomes apparent in this study, is the requirement to predefine moisture sources in this forward calculation approach. If more spatial detail is required, the computational overhead multiplies and can become prohibitive. In contrast, the Lagrangian backward approaches provide spatially detailed information, that can be more easily interpreted, for example in terms of the physical processes related to weather systems. This discrepancy between both approaches is important to mention here.

We thank the reviewer for his suggestion. In the revised manuscript we will explicitly mention this drawback of the Eulerian water vapor tracers, by adding the following sentence: "Additionally, the amount of information they provide is limited, as they need to predefine the moisture source to be tagged". In addition, we will also mention that Lagrangian methods provide gridded information, which is another advantage, along with their higher computational efficiency.

L. 59: Maybe mention here that the Lagrangian methods, being offline diagnostics, require a range of assumptions and parameter choices to which these methods are sensitive. Your comparison framework allows to assess what biases exist with the different diagnostics, and how those are related to parameters and assumptions in the Lagrangian methods.

Once more we thank the reviewer for this suggestion. We will explicitly state here that Lagrangian methods "are sensitive to a range of hypotheses and parameter choices, which significantly increases their uncertainty". These hypotheses and parameters are discussed further in the methodology, and the uncertainty they cause becomes apparent when introducing our physics-based modifications.

L. 61: "fully validated": I assume this relates to the internal consistency of the tagging approach. Validation can be misunderstood as a comparison to observable quantities. Please clarify/rephrase.

We are going to rephrase "fully validated" to "internally consistent".

L. 70: This is not correct, Sodemann et al. (2008) used trajectories from the LAGRANTO model (Sprenger and Wernli, 2015).

We thank the reviewer for his clarification. We will include this corrected information in the revised version of the manuscript.

L. 73: "limited to highlighting ... large discrepancies": Please rephrase to do more justice to what is presented in the cited studies. For example, Winschall et al. (2014) specifically investigated the basis of the boundary layer vs. free troposphere distinction in the WaterSip method.

What we mean here is that in these studies the authors did not propose any modifications to reconcile the results of the methods used. In any case, we agree that this sentence is misleading, as it implies that these studies only compared the tools, when in fact they looked at more, as in the case of Winschall et al., (2014). This sentence will therefore be rephrased in the manuscript.

L. 77: "two of the most widely used" -> "two widely used"

We will correct this typo in the revised version of the manuscript.

L. 84: "vast majority ... force": there is no evidence supporting this statement. I don't think it is necessary to make this statement, adding reanalysis data is useful because unlike forecast data, it includes analysis increments from data assimilation, see Fremme et al., 2023.

We will delete this sentence because we have not tested it, but from our experience we believe that in most studies on moisture sources with FLEXPART, the model is forced with reanalysis data. A more thorough literature review would serve to verify this.

L. 93: It is certainly positive with different AR cases, but these cases are all long-range transport events. Can you add some clear justification for this focus in the introduction? Some of the writing makes the impression that you seek general validity, while the focus on AR events only seems in contradiction to this.

On the one hand, we focus on AR cases because we plan to compute moisture sources from a climatological perspective using one of the Lagrangian moisture tracking methodologies assessed here in the future, which is mentioned in the introduction. On the other hand, we agree that considering other types of precipitation events would give more validation to our study. However, we are aware of the existence of a model intercomparison effort in the moisture tracking community, where different types of cases are analyzed, so we decided to focus our study exclusively on ARs in order to interfere as little as possible.

Figure 1: Please add panel labels, and mention all figure panels in the caption. It would be a large advantage to have common color bars for the left and right column panels each. For precipitation amount, it is quite common to use a categorized color bar to that end. This would avoid the saturation of the color scale that now seems to occur.

We will add panel labels and a reference to the geographic region in which precipitation is tracked, and use a categorized color bar as suggested.

Table 1: Could this table include information about the total rainfall amount of these events in the model and maybe observations? Is it correct that the two last events have the same date and time, but different regions?

We thank the reviewer for his suggestion. We will include two more columns in Table 1 with the precipitation in mm from the WRF simulations and the reanalysis ERA5. Regarding the last question, there was a typo, the correct initial date and time for the Greenland case is 2012-07-09 12.

L. 119: What has been used in terms of deep and shallow convection, microphysics schemes in the WRF simulation?

In the WRF simulations, the main parameterizations used were the Yonsei University (YSU) for the boundary layer, the WRF single-moment-6-class (WSM6) for microphysics, and the Kain-Fritsch for convection. In the revised version, we will move this information from the supplement to the main manuscript, specifically at the end of the first paragraph of Sect. 2.2.

L. 121: Which fields have been nudged, only winds or also specific humidity? How does the nudging affect the tagging? The authors emphasize the importance of the nudging, but actually I think for the study objective it does not make a difference if the results resemble the actual events closely or not.

Apart from the winds, temperature and geopotential height are nudged, and this will be mentioned in the revised version of the manuscript. As to whether nudging is important or not, it is crucial when comparing with FLEXPART forced with ERA5 data.

L. 126: What is meant by this statement, and how does it relate to this citation? Consider maybe citing Gimeno et al., 2021 here.

We meant to cite van der Ent and Tuinenburg, (2017), instead of van der Ent and Tuienburg, (2013). In the first case they explicitly show the positively skewed probability density functions for the residence time of atmospheric water vapor. This justifies the long duration of our simulations, 30 days. This typo will be corrected in the revised version of the manuscript.

L. 133: If QFX is assimilated from ERA5, this can introduce an inconsistency into WRF due to differences in resolution. What is the reason for this choice? How different are the results when using the WRF-internal evaporation flux?

Before assimilating QFX from ERA5 we have interpolated the reanalysis moisture flux to the WRF domain. This was something that came up when we started to compare the results of WRF-WVTs with those of both Lagrangian methodologies directly using simulations from FLEXPART forced with ERA5, but in the end we have found that it does not make a big difference.

L. 138: Is a time interpolation used here?

No, in our case we are not using a time interpolation, as we use hourly evaporation data from the ERA5 reanalysis.

L. 149: It may be useful to have some basic information in the main manuscript, such as the chosen parameterisation schemes, and the fact that simulations are hemispheric (?)

We agree with the reviewer that moving the information about the chosen parameterizations from the supplement could be useful, and we will do it in the revised version of the manuscript. Specifically, the chosen parameterizations will be mentioned at the end of the first paragraph of Sect. 2.2. Regarding the domains, simulations are not hemispheric, as Fig. 2 shows. The domain in the Northern Hemisphere, for example, covers latitudes from 0º to 65ºN.

Figure 2: The source regions are very large in comparison to the scale of the moisture sources revealed by the Lagrangian diagnostics. A separation into e.g. 10 degree latitude bands or latitude-longitude boxes could allow for a much more detailed comparison and evaluation of Lagrangian models in the Eulerian framework.

This question has already been answered in response to the reviewer's major comment 1.

L. 166: "FLEXPART assimilates hourly data": FLEXPART does not perform data
assimilation, please rephrase. It is not clear what is said from this sentence, the
previous section described WRF, not FLEXPART. How exactly is FLEXPART run with
WRF? Maybe some of the details from the supplement could be moved to the main
text. In particular, it is important to describe how particles were initiated and released,
and if any convection parameterisation was active in FLEXPART.

We are going to change "FLEXPART assimilates hourly data" by "FLEXPART ingests
hourly data". FLEXPART is run with WRF by using FLEXPART-WRF, which is able to
ingest input data from the output of the WRF model. Finally, we agree with the reviewer
that moving some details from the supplements could be useful. Specifically, in the
revised version of the manuscript we will explicitly state how parcels are released in the
FLEXPART simulations: "parcels are released using the domain filling option over the
black boxes in Fig. 1 such that they are vertically distributed following the density
profile".

L. 174: "it starts by assuming": This sentence and the following sound a bit strange.
What you describe seems to be the basic idea of Lagrangian analysis, which is not
particular to WaterSip. It would be useful to cite Stohl and James (2004) in this context,
or shorten the section altogether, because all of this has already been said elsewhere.

We are going to rewrite this paragraph to retain only the most essential information,
i.e., the following sentence: "the atmospheric column over the region where
precipitation occurs is filled with air parcels, and that their trajectories contain
information about their location and specific humidity at 6-hourly intervals for the
previous days".

L. 181 to 207: This section repeats a lot of information that is found in the original
publication, and is not necessarily more easy to follow. I recommend limiting this to the
most essential parts of the method which are modified here.

These paragraphs will be rewritten in the revised version of the manuscript.
Specifically, we will briefly explain how WaterSip works, focusing on how the spatial
distribution of moisture sources is obtained, and the remainder of that paragraph will be
shortened. For example, the explanation of the discounting algorithm is omitted, as it
can be found both in the original study of Sodemann et al., (2008) and in the
Supplement. Once the methodology is introduced, we will present some modifications
of WaterSip previously used in the literature, and which correspond with the different
configurations that we test in our study. Here we acknowledge Dütsch et al., (2018) and
Cheng and Lu, (2023) for introducing the modification that we later demonstrate reduce
biases in WaterSip, "A less common  modification is to filter the specific humidity
decreases, such that previous contributions are only discounted if a specific humidity
decrease occurs and the relative humidity of the parcel is higher than 80 % (Dütsch et
al., 2018; Cheng and Lu, 2023)". Finally, we also point out here the dependence of
WaterSip on the specific humidity threshold and time step.

L. 188: The threshold value has been repeatedly shown to be a key sensitivity
parameter (e.g., Sodemann and Stohl, 2009; Fremme and Sodemann, 2019). In
addition, this value is on the very low end, that has been previously recommended for
Arctic studies. How sensitive are your baseline results to this choice? To be in line with
literature, I recommend a delta q of 0.2 for a 6h time interval.

As we mentioned in our response to the major comment 3, in the revised version of the
manuscript Fig. 5 will include a sensitivity experiment in which we change both the threshold value and the time step. This demonstrates that our results are not very sensitive to the threshold value, but they are to the time step. Additionally, for the basic results we will use the recommended setup of 0.2 g/kg for a 6 h interval, instead of 0.05 g/kg for a 3 h interval as before.

Regarding section S3.1 referenced here, I wonder about what the role of this mathematical description is for the manuscript. There seem to be some arguments about correspondences between the two Lagrangian methods mathematically, but conceptually the two are quite different (e.g., well-mixed properties of the atmosphere). Section S3.1 could benefit from a closer connection to published literature to clarify its 450 purpose. Does this section describe what has been published before, but mathematically in a common framework?

The original idea of section S3.1 was to unify the mathematical framework of the methodologies to show that both use a linear discounting, so that they are equivalent from a mathematical (and computational) perspective. However, it is true that they are 455 conceptually different. Still, since we will present less details on the WaterSip methodology in the revised version of the manuscript, we believe it makes sense to keep this section of the Supplement as is.

L. 209: The authors refer to the UTrack method as the Dirmeyer and Brubaker (1999) implementation they use. However, as noted in L. 218, UTrack computes its own 460 trajectories. Is it then not more correct to refer to the second model as the Dirmeyer and Brubaker (1999) method? What really distinguishes the approach used here from UTrack and Dirmeyer and Brubaker (1999), respectively?

We agree with the reviewer that it is more correct to refer to the second model as the Dirmeyer and Brubaker, (1999) method (we will use the abbreviation DB99). We are 465 going to update the manuscript with this modification. The approach used here is the same as UTrack or Dirmeyer and Brubaker, (1999), but using FLEXPART trajectories instead of computing them. To do that we also need to consider how parcels are vertically released, as in the case of the Dirmeyer and Brubaker, (1999) methodology or UTrack they follow the humidity profile. Since in our case parcels are vertically 470 distributed following the density profile, we had to weight the moisture sources of each parcel using their specific humidity. These important remarks will be now included in the methodology:

"However, in our case we use the FLEXPART and FLEXPART-WRF trajectories at hourly resolution and implement only the diagnostic tool to compute the moisture 475 sources for precipitation. Thus, since in our simulations parcels are vertically released following the density profile, we weight the contribution of each parcel by its specific humidity to match the DB99 methodology."

L. 252: It has been common to initialize the domain at model time zero with all water vapour currently in the domain to achieve 100% accounting. Has this been tested 480 here?

No, we have not tested that. This is evaluated in Insua-Costa and Míguez-Macho, (2018), where they show the internal consistency of WRF-WVTs.

L. 255: This is not correct. The Dirmeyer and Brubaker (1999) method stops accounting evaporation when 100% have been reached. The WaterSip method does not generally 485 reach 100% (see Sodemann et al., 2008).

We agree with the reviewer that the WaterSip method does not generally reach 100 %. We meant that WaterSip typically reaches 100 % when the simulation time is 30 days and all uptakes are considered, but this is something that we have not shown. Because of this, we are going to delete that sentence from the manuscript.

L. 256: What is meant by "the bias will also be calculated after adjusting for these precipitation fractions"? This scaling should be explained in the methods section. Why is a scaling necessary at all? Is it not more correct to compare the actual identified fractions? What about comparing amounts rather than fractions?

We agree that scaling is not strictly necessary. In the revised version of the manuscript,
the results will correspond to the comparison of the actual precipitation fractions without scaling, with the exception of the "ABL" and "RH" configuration, as in these cases the attributed precipitation is typically much lower or much higher than 100 % if we do not scale the precipitation fractions. On the other hand, we discarded comparing absolute quantities instead of fractions, as we can then be unconcerned whether in
WaterSip the diagnosed precipitation, which comes from the specific humidity decreases at the time and location of the rainfall event, matches the WRF precipitation.

L. 266: I think the reference to Winschall et al. 2014 is not justified in such a general statement as done here. Winschall et al. 2014 did a sensitivity test of different tagging approaches, and their conclusion was: "The results of the Lagrangian diagnostics are
similar to the Eulerian results, with the fraction of remote versus local moisture sources lying in between the two realisations of the tagging technique."

We agree that Winschall et al., (2014) did a sensitivity test of different tagging approaches, instead of assessing the Lagrangian method. Because of this, we will omit that reference in this part of the manuscript.

L. 268: Is the RMSE expressed as a fraction as in Eq. (6) or in percent?

We have corrected the manuscript expressing the RMSE as percentage, as the RMSE should have the same units as the precipitation fractions.

L. 274: It is interesting to note that the biases of the UTrack method are different. Why is that the case? The US West Coast case for example, UTrack has a lower
performance.

This is because in the US West Coast case the moisture sources are highly dependent on altitude. This can be seen from Fig. 7, where we can check that the RMSE for the Dirmeyer and Brubaker, (1999) methodology changes a lot with the threshold for release height of parcels.

Figure 4: It is not possible the read the numbers printed in white on a light colour background.

We have updated this figure (and also Figure 9), so that all numbers can be easily read.

L. 279: I do not see a value of 29.6 in Fig. 3, nor of 14.88 for the Tropical Atlantic in Fig.
4. Is this example part of the supplement information? How does the scaling impact the results here?

We agree with the reviewer that the 29.6 value could not be easily deduced from Fig. 4., as the biases shown in Fig. 4 were computed scaling both the precipitation fractions in Fig. 3 and the precipitation fractions calculated with the Lagrangian methodologies.

In the revised version of the manuscript we will not scale the results (with the exception of the ABL and RH configurations, where the attributed precipitation is far from 100 %) so the precipitation fractions in the Lagrangian diagnostic tools can be easily deduced from Fig. 3 and 4.

L. 291: This statement applies to both Lagrangian methods. Before proceeding to tune
the methods, it would be useful to quantify the overall bias of the Lagrangian vs. Eulerian methods, maybe at the end of Sec. 3.1, potentially as a function of distance from the arrival location. It may also be worthwhile to comment on the overall consistency of the results from the 3 approaches here. It would also be interesting to know more about the sensitivity here already regarding the specific setup you chose.
How different are the errors/biases for a time interval of 6h, and when increasing the specific humidity threshold to 0.2 g kg-1 6h (or more)?

We agree with the reviewer that the statement applies to both Lagrangian methods, so we will mention WaterSip and the Dirmeyer and Brubaker, (1999) methodology in that sentence. Regarding the overall bias as function of distance from the arrival location,
we understand that this information is somehow implicit in the estimated moisture sources, which generally gives more weight to closer regions. Finally, the dependence on the time interval and specific humidity threshold will be discussed in the revised version of the manuscript and included in an updated version of Fig. 5.

L. 295: What is presented here is exactly the argument for introducing a specific
humidity threshold in WaterSip. So this needs not be formulated as a (new) hypothesis, it is part of the known uncertainty of the WaterSip diagnostic.

We will rewrite this sentence to explicitly mention that our hypothesis is that fluctuations penalize remote contributions. We know that the introduction of a specific humidity threshold tries to cope with noise in this diagnostic tool, but to our knowledge this
specific implication of fluctuations has not yet been formulated. Thus, we will change "we conjecture that non-physical humidity fluctuations" to "we explore the hypothesis that non-physical humidity fluctuations" at the beginning of Sect. 3.2.1, and explain better why non-physical negative changes in specific humidity penalize earlier contributions at the end of the same paragraph, by adding "as the error caused by a
single fluctuation affects all previous contributions, so the early moisture uptakes will be affected by many more non-physical changes".

L. 315: This distinction and modification have already been proposed by Dütsch et al., 2018 (Their Sec. 3.2). However, it is important to note that mixing with dry air can also lead to a specific humidity decrease. By only allowing for precipitation events to
decrease specific humidity, a bias is intoduced into the method. This can also result in an over-accounting of sources (more than 100% of moisture accounted for).

This has already been answered above (see reviewer's major comment 3).

Figure 8: Comparing the UTrack results with the corresponding results from WaterSip in Fig. 7, it is very interesting to note how different the spatial maps are from the two
methods. UTrack basically shows almost no sources at all in the vicinity of Greenland. While we don't know which one of the results is more correct, this difference is not picked up by the comparison to water vapour tagging in the present setup. This fact points to the current tracer setup being not sufficiently sharp (or detailed enough) to resolve and quantify such differences.

This has already been answered above (see reviewer's major comment 1).

**References**

Bonne, J.-L., Steen-Larsen, H. C., Risi, C., Werner, M., Sodemann, H., Lacour, J.-L., Fettweis, X., Cesana, G., Delmotte, M., Cattani, O., Vallelonga, P., Kjær, H. A., Clerbaux, C., Sveinbjörnsdóttir, A. E., and Masson-Delmotte, V., 2015: The summer 2012 Greenland heat wave: In situ and remote sensing observations of water vapor isotopic composition during an atmospheric river event, J. Geophys. Res., 120, 2970-2989, doi:10.1002/2014JD022602.

Cheng, T. F. and Lu, M.: Global Lagrangian Tracking of Continental Precipitation Recycling, Footprints, and Cascades, J. Clim., 36, 1923–1941, https://doi.org/10.1175/JCLI-D-22-0185.1, 2023.

van der Ent, R. J., Tuinenburg, O. A., Knoche, H.-R., Kunstmann, H., and Savenije, H. H. G.: Should we use a simple or complex model for moisture recycling and atmospheric moisture tracking?, Hydrol. Earth Syst. Sci., 17, 4869–4884, https://doi.org/10.5194/hess-17-4869-2013, 2013.

van Der Ent, R. J. and Tuinenburg, O. A.: The residence time of water in the atmosphere revisited, Hydrol. Earth Syst. Sci., 21, 779–790, https://doi.org/10.5194/hess-21-779-2017, 2017.

Dirmeyer, P. A. and Brubaker, K. L.: Contrasting evaporative moisture sources during the drought of 1988 and the flood of 1993, J. Geophys. Res. Atmospheres, 104, 19383–19397, https://doi.org/10.1029/1999JD900222, 1999.

Dütsch, M., Pfahl, S., Meyer, M., and Wernli, H.: Lagrangian process attribution of isotopic variations in near-surface water vapour in a 30-year regional climate simulation over Europe, Atmos. Chem. Phys., 18, 1653–1669, https://doi.org/10.5194/acp-18-1653-2018, 2018

Fremme, A. and Sodemann, H., 2019: The role of land and ocean evaporation on the variability of precipitation in the Yangtze River valley, HESS, 23, 2525–2540.

Fremme, A., Hezel, P. J., Seland, Ø., and Sodemann, H.: Model-simulated hydroclimate in the East Asian summer monsoon region during past and future climate: a pilot study with a moisture source perspective, Weather Clim. Dynam., 4, 449–470, https://doi.org/10.5194/wcd-4-449-2023, 2023.

Gimeno, L., Eiras-Barca, J., Durán-Quesada, A.M., Dominguez. F., van der Ent, R., Sodemann, H., Sánchez-Murillo, R., Nieto, R. and Kirchner, J. W.: The residence time of water vapour in the atmosphere. Nat. Rev. Earth Environ., https://doi.org/10.1038/s43017-021-00181-9, 2021.

Sodemann, H., Schwierz, C., and Wernli, H.: Interannual variability of Greenland winter precipitation sources: Lagrangian moisture diagnostic and North Atlantic Oscillation influence, J. Geophys. Res. Atmospheres, 113, D03107, https://doi.org/10.1029/2007JD008503, 2008.

Sodemann, H. and Stohl, A.: Asymmetries in the moisture origin of Antarctic precipitation, Geophys. Res. Lett., 36, https://doi.org/10.1029/2009GL040242, 2009.

Sodemann, H. and Stohl, A.: Moisture Origin and Meridional Transport in Atmospheric Rivers and Their Association with Multiple Cyclones, Mon. Weather Rev., 141, 2850–2868, https://doi.org/10.1175/MWR-D-12-00256.1, 2013.

Sodemann, H. and Joos, H., 2021: Numerical methods to identify model uncertainty in: Ólafsson, H. and Bao, J.-W. (Eds), Uncertainties in Numerical Weather Prediction, Elsevier, 309-329, doi: 10.1016/B978-0-12-815491-5.00012-4, 2020.

Sprenger, M. and Wernli, H.: The LAGRANTO Lagrangian analysis tool – version 2.0, Geosci. Model Dev., 8, 2569–2586, https://doi.org/10.5194/gmd-8-2569-2015, 2015.

Stohl, A., Forster, C. and Sodemann, H., 2008: Remote sources of water vapor forming precipitation on the Norwegian west coast at 60°N - a tale of hurricanes and an atmospheric river, J. Geophys. Res., 113, D05102, doi:10.1029/2007JD009006.

Terpstra, A., Gorodetskaya, I. V., Sodemann, H.: Linking sub-tropical evaporation and extreme precipitation over East Antarctica:an atmospheric river case study, J.
Geophys. Res., 126, https://agupubs.onlinelibrary.wiley.com/doi/10.1029/2020JD033617, 2021

Yoshimura, K., Oki, T., Ohte, N., and Kanae, S.: Colored Moisture Analysis Estimates of Variations in 1998 Asian Monsoon Water Sources, 気象集誌 第2輯, 82, 1315–1329, https://doi.org/10.2151/jmsj.2004.1315, 2004.

Winschall, A., Pfahl, S., Sodemann, H., and Wernli, H.: Comparison of Eulerian and Lagrangian moisture source diagnostics – the flood event in eastern Europe in May 2010, Atmospheric Chem. Phys., 14, 6605–6619, https://doi.org/10.5194/acp-14-6605-2014, 2014.

Zhu, Y. and Newell, R. E.: A Proposed Algorithm for Moisture Fluxes from Atmospheric
Rivers, Mon. Weather Rev., 126, 725–735, https://doi.org/10.1175/1520-0493(1998)126<0725:APAFMF>2.0.CO;2, 1998.

---

## Author Response (AR1)

General comments

This study investigates the uncertainty in precipitation source regions estimated by three different modeling approaches. Precipitation sources estimated by the online Eulerian-based WRF-WVT method are taken as the reference, against which estimates from two offline Lagrangian-based methods are compared: the WaterSip and UTrack methods. Both methods are found to exhibit biases in the estimated precipitation sources compared to the reference data set, in particular showing sources to be geographically closer to the precipitation than the more remote sources estimated by the reference. The study then tests a structural modification to each of the WaterSip and UTrack methods and finds bias is reduced and precipitation sources are made geographically closer to those of the WRF-WVT reference. A key conclusion of the study is that the Lagrangian methods can serve as viable alternatives to the more computationally-expensive WRF-WVT method. The study is well-defined, well-written and the conclusions logically follow the results. In particular, the authors are to be commended for detailing the structural differences between the models. The main area of improvement needed is the clarification of the proposed modifications to the Lagrangian models, and their resulting evaluation against the reference dataset.

*Thank you very much for your comments, which we think have substantially improved the article. Please, find below our detailed responses to them.*

Specifically, the modification of the UTrack model appears to contain two changes: (1)

only parcels released from above 2km may be used for tracking, and (2) of those parcels, only those with relative humidity above 90% are subsequently tracked. It is unclear which modification dominates the reported changes to precipitation sources relative to the WRF-WVT sources. Of more minor importance, it is unclear why a higher relative humidity threshold is applied to the UTrack model compared to the WaterSip model; this choice of model modification needs to be clarified.

*It is true that the proposed modification of UTrack contains two changes (releasing parcels from above 2 km and retaining only those with relative humidity higher than 80 %) and we did not test each one of them separately. We have now made it clear in the revised version of the manuscript that there are two changes, and we tested them in*

*two steps. First, we applied the relative humidity filter and evaluated the improvement. After that, we repeated the same experiment, but changing the threshold for the release height. We also modified Fig. 7 by adding another red dot resulting from applying only the relative humidity filter. The figure demonstrates that most of the impact in the results came from the second step, changing the threshold for the release*

*height. Regarding the choice of the relative humidity threshold, there was a typo in the manuscript, as it should always be 80 %.*

The modification of the WaterSip model, requiring parcels to have a minimum relative humidity of 80% immediately before a decrease in specific humidity, needs to be explained more clearly. It needs to be made clearer what the exact problem is with the way WaterSip reduces parcel specific humidity en route, and how applying an 80% threshold of relative humidity helps..*The explanation of the WaterSip model in the methodology has been rewritten. Specifically, we have omitted some information that*

can be found in the literature, particularly the explanation of the basic configuration of WaterSip, well documented in Sodemann et al., (2008). We have focused more on different modifications of this diagnostic tool that have been used in previous studies (Fremme and Sodemann, 2019; Dütsch et al., 2018), as they relate to the problem of specific humidity fluctuations that we investigate later. Moreover, in Sect. 3.2.1 we have clarified why these fluctuations may penalize remote contributions (see our response to your specific comment for a more detailed explanation). Under these assumptions, the application of an 80 % threshold of relative humidity helps to address this problem, as it is a simple approach to filter out non-physical specific humidity decreases (not associated with precipitation), and this has also been included in the second paragraph of that section.

Specific comments:

L47: Which problem is being referred to here?

We refer here to the problem of the origin of moisture in ARs. In the revised manuscript we have made it explicit for clarity, by replacing this sentence by "However, those studies in which they quantify the relative importance of different moisture sources focus on individual cases, so the debate on the origin of moisture in ARs is not yet completely closed. This is reflected in the definition of AR given in the Glossary of Meteorology, where it is indicated that the sources of moisture can be tropical and/or extratropical (Ralph et al., 2018)".

L55/60: Here it is asserted that Eulerian approaches are more accurate than Lagrangian approaches. I do not think it is true that, in general, Eulerian tracing approaches are considered to be more reliable than Lagrangian approaches in accurately estimating precipitation sources. Perhaps you mean *online* Eulerian water vapor tracers are considered more accurate? If this is the case, I suggest rephrasing to clarify. Furthermore, if Lagrangian approaches are asserted to contain "more uncertainty", than these uncertainties need to be outlined. Relatedly, I think it is important to be careful about asserting that WRF-WVTs can be "considered as synthetic observations". There needs to be some evidence that WRF-WVTs can in fact accurately represent real observations, for example through comparison with satellite observations of atmospheric moisture. If this or a similar type of evaluation has been done, please refer to it here. Otherwise, I would tone down the language by changing the words "considered as synthetic observations" in L63 (also in L436) to "used as a reference".

We agree with the reviewer that Eulerian approaches are not more accurate than Lagrangian approaches in general. Because of this, we have rewritten this sentence and clarified that it is the online water vapor tracers that we consider to be more accurate, provided that the model simulation where they run is realistic. Regarding the uncertainties in Lagrangian models, we agree that we did not elaborate enough about this issue. In the revised version, we have emphasized that the uncertainty comes from a number of hypotheses and parameters, which are precisely those explored in this study.

Finally, the reviewer's comment makes us realize that perhaps we haven't sufficiently clarified what we mean by "synthetic observations", something that is fundamental to this work. In our study, we use WRF-WVTs to create a "synthetic atmosphere", which is simply the atmosphere resulting from the model run. The word "synthetic" in this context is used to indicate that it is artificially generated trying to copy the real thing as closely as possible. In this artificial or "synthetic" atmosphere, the moisture from arbitrarily selected regions is accurately tracked in 3-dimensions until it precipitates. Since we cover all possible source regions, we know the moisture origin of precipitation anywhere, broken down into percentages from the different source regions that we defined earlier. These are our "synthetic observations" of the moisture origin of precipitation, within our "synthetic atmosphere", all generated by WRF-WVTs.

In this paper, our approach to test Lagrangian models is to run them not on the real (reanalysis) but on the synthetic atmosphere, for which we accurately know moisture pathways from the different source regions until it precipitates, i.e. the "observations". This allows us to make adjustments and assess results, since we have a truth to compare with and guide us in the process. That these "synthetic observations" accurately represent real observations depends on whether the "synthetic atmosphere" is realistic or not. But even if the WRF model run were to diverge substantially from the real atmosphere, the result from the online Eulerian tool would still be a "synthetic observation", because within the "model world", i.e. the "synthetic atmosphere", it is the truth. This approach and wording using "synthetic" experiments or observations is used in other disciplines to test different retrieval algorithms, etc. We have now reworded the text to ensure that there is no confusion and that we describe our strategy more clearly:

"In this context, the goal of this paper is to compare and adjust two Lagrangian methodologies for the computation of moisture sources for precipitation (or precipitation sources) focusing on AR-related rainfall events. The strategy we adopt is to run the Lagrangian models on atmospheric data from simulations of the Weather Research and Forecasting (WRF) model with Water Vapor Tracers (WRF-WVTs; Insua-Costa and Miguez-Macho, 2018) and introduce physically based modifications so that the results are aligned with those provided by the latter tool. The rationale for this approach is simple. Online Eulerian water vapor tracers, being coupled to a meteorological model, account for all the physical processes affecting atmospheric moisture that are resolved or parameterized by the model. In the case of WRF-WVTs, they are internally consistent, showing an almost exact performance within the "model world" (Insua-Costa and Miguez-Macho, 2018), i.e. they constitute synthetic observations generated from the model simulation. Furthermore, in the absence of direct observations, results provided by WRF-WVTs are particularly suitable to be considered as reference when comparing with other methods, as long as the simulated atmosphere behaves like the real one and follows it closely. Their disadvantage, however, is that they are computationally expensive, and therefore their application over long time periods or in many case studies is often unfeasible. Additionally, the amount of information they offer is limited, as the moisture source to be tagged needs to be predefined. In contrast, Lagrangian methods are much more computationally efficient and provide gridded information, but they are sensitive to a range of hypotheses and parameter choices, which significantly increases their uncertainty. Achieving a Lagrangian moisture source diagnostic that mimics the WRF-WVTs results would therefore imply having a very accurate and at the same time flexible tool that can be applied to a large number of ARs, our goal for the future, but probably also to other types of weather or climate phenomena."

L145: Is the specific humidity assimilated from ERA5 like the evaporation field? Does the WRF model close the water balance if ERA5 evaporation is assimilated?

No, the specific humidity is not assimilated from ERA5 like the evaporation field. The WRF model closes the water balance in this case, as we are only changing the surface moisture flux simulated by WRF by the surface moisture flux in ERA5, and the other
moisture fluxes are updated accordingly by the model itself.

L155: While the manuscript makes it clear that parcel trajectories are calculated using
WRF data in the first case, and ERA5 data in the second case, it is a little unclear
which dataset was used to calculate the moisture contribution for each Lagrangian
model. From reading section 2.3, I interpret that in the first case, "FLEXPART-WRF",
WaterSip reads the specific humidity field from WRF, and UTrack reads the precipitable
water field from WRF but the evaporation field is ERA5 data assimilated into WRF. In
the second case, "FLEXPART-ERA5", I interpret that both WaterSip and UTrack read
all fields from ERA5. If this is not the correct interpretation, please clarify.

This is exactly the correct interpretation. To make this clearer, we explain it better in the
last sentences of the first paragraph of Sect. 3: "Finally, in Sect. 3.3 we test the
introduced modifications when the trajectories are generated by FLEXPART-ERA5,
with input data from the ERA5 reanalysis, coming also the other fields that the
diagnostic tools need from the same reanalysis, instead of WRF simulations."

L172 & L210: The Dirmeyer and Brubaker approach is also used by other studies,
whose moisture tracking method is very similar to UTrack, e.g. Holgate, C. M., J. P.
Evans, A. I. J. M. van Dijk, A. J. Pitman, and G. D. Virgilio, 2020: Australian
Precipitation Recycling and Evaporative Source Regions. Journal of Climate, 33, 8721–
8735, https://doi.org/10.1175/JCLI-D-19-0926.1. Similarly, the WaterSip approach is
also used by other studies, e.g. Cheng, T. F., and M. Lu, 2023: Global Lagrangian
Tracking of Continental Precipitation Recycling, Footprints, and Cascades. Journal of
Climate, https://doi.org/10.1175/JCLI-D-22-0185.1. Though these specific methods are
not formerly evaluated here, it would be pertinent to acknowledge them.

We were aware of the problem with the nomenclature for the Dirmeyer and Brubaker,
(1999) methodology, but did not know how to solve it. In the revised manuscript, we
acknowledge these other studies and refer to the diagnostic tool as "the Dirmeyer and
Brubaker, (1999) methodology", abbreviated as DB99, instead of "UTrack".

Figures 3 and 4: it would be helpful to the reader if these figures could be placed side
by side for easier comparison. Is it possible to combine the two figures into one?

As both figures refer to different subsections, we do not think it is possible to combine
them into one. However, we will ask for them to be placed one after the other.

L230: To make it easier for the reader to interpret the error scores, it would be helpful to
add a sentence linking each score with a physical meaning, e.g. a higher value of
MAESS refers to a more accurate comparison with the reference dataset.

We agree with the reviewer that we did not make clear the interpretation of the Mean
Absolute Error Skill Score (MAESS). In the revised version of the manuscript, we have
explicitly mentioned that a closer value of MAESS to 1 means a more accurate aligning
with the reference dataset, by adding "as usual with a skill score, the closer to 1 means
that the results of the Lagrangian diagnostic are closer to those of WRF-WVTs" once
the MAESS is introduced.

L303: To make it clearer to the reader, it would be helpful for the accumulation over
time to be shown with a simple example. As the manuscript currently reads, it is
unclear what the problem with the WaterSip method is.

To better explain how the error accumulates over time, we have included another iteration of our simple example (a couple of non-physical increases and decreases): "If another non-physical decrease occurs, this value is updated to 1.95(1-0.05/2.0)=1,90 g kg$^{-1}$". Moreover, we have made it explicit that early contributions are more penalized, as they are affected by many more potential non-physical fluctuations, by adding "as the error caused by a single fluctuation affects all previous contributions, so the early moisture uptakes will be affected by many more non-physical changes".

L378: The original configuration of UTrack appears to release parcels from a random, humidity-weighted vertical level, indicating the starting parcel levels will be in the lower part of the troposphere. Yet here, and in Figure 7, it is indicated that the starting parcel level is 0km. Was the starting parcel height set at 0km in this study, or was a random, humidity-weighted vertical level used as in the original model? Further, did this study use a random, humidity-weighted vertical release height and simply ignore those parcels starting below 2km, or was the release height set at a constant 2km level in the modified case?

We agree with the reviewer that it is not clear how parcels are released vertically, and this should be clarified, as it is a key point of the modification we are proposing. In our case, parcels are vertically released following the density profile, using the domain-filling option of FLEXPART. In the case of the Dirmeyer and Brubaker, (1999) methodology, parcels are released following the humidity profile of the atmosphere. Thus, to match our approach with the original one, we need to weight the moisture origins for each parcel using their humidity. This additional (and important) information has been included in the methodology in the revised version of the manuscript. Specifically, we have added a sentence at the end of the first paragraph of Sect. 2.3 to explain how parcels are vertically released in FLEXPART: "In both cases parcels are released using the domain filling option over the black boxes in Fig. 1, such that they are vertically distributed following the density profile". When explaining the Dirmeyer and Brubaker, (1999) methodology, we have also added another sentence for clarification: "since in our simulations parcels are vertically released following the density profile, we weight the contribution of each parcel by its specific humidity to match the DB99 methodology".

Regarding the modification we propose, we simply ignore parcels starting below 2 km, being the rest released as usual. This has also been clarified in Sect. 3.2.2 of the revised manuscript, as we have added "particles are released as usual at the time and location of the precipitation event, but those below $z_b$ are excluded from the analysis" when introducing the modification.

L416: Can you provide some reasoning as to why WaterSip is superior to UTrack when using ERA5 data?

This is probably related to the different number of vertical levels (38 for WRF versus 70 for ERA5) and, in particular, to the extent to which the different methods are sensitive to having more or fewer levels, but this is something we do not know for sure and would require further analysis.

L475: The statement that the Lagrangian methods can serve as viable alternatives for WRF-WVTs is a key conclusion of the study. I would suggest including this conclusion in the abstract.

We thank the reviewer for this suggestion. In the revised version of the manuscript, this will be one of the main points of the abstract. Specifically, the last sentence of the revised abstract is now "Although these modifications may need to be adjusted for other types of precipitation events, our results demonstrate that Lagrangian techniques are a viable and compatible alternative to Eulerian water vapor tracers, and that the main discrepancies between the different methodologies can be derived from the obviation of basic physical considerations that may be easily straightened out."

Technical corrections

Figure 1: it would be helpful if the subplots each had a title describing their geographic location, e.g. "South Africa". These location labels can then be added to Table 1 to make it easier for the reader to associate the numerical description with a real-world
location.

This figure includes now the corresponding geographic labels in the revised version.

Figure 2: "Tropical Indic" should perhaps be "Tropical Indian" (same issue applies to later figures). Also some parts of the world are classed as "Tropical land" when they are in fact desert regions (e.g. northern and southern Africa, central Australia, Arabian
peninsula). To avoid re-running the model with different regions, I suggest touching on the implications of this classification in your results.

We have corrected all figures changing "Indic" by "Indian" where it corresponds. Regarding the classification of desert regions as "Tropical land", we understand that it could lead to confusion if we were to conclude that a certain amount of precipitation
comes from these areas, but this is not the case, as we are referring to the source as a whole. In any case, we do not consider it incorrect, since deserts in the tropics are still "tropical lands", so we decided to keep it as it was. Also, note that removing deserts from the sources classified as "tropical lands" would not affect our comparison at all.

L165: Should "Except for the position and the…" be "Except for the position of the
parcel and the …"?

The typo has been corrected.

**Answer to Harald Sodemann (R2) in the Interactive comment on "Simple physics-based adjustments reconcile the results of Eulerian and Lagrangian techniques for moisture tracking" by Alfredo Crespo-Otero, Damián Insua-Costa, Emilio Hernández-García, Cristóbal López and Gonzalo Míguez-Macho**

The authors present a study focused on the comparison between Eulerian and Lagrangian approaches to trace moisture and to identify the evaporation sources of precipitation. Using a regional model simulation with water tagging as a reference, they then evaluate two Lagrangian offline approaches in that framework for a set of Atmospheric River events from different regions. Two tunings are proposed to reduce a general bias towards shorter transport distances in Lagrangian methods. The study is overall interesting, presented clearly, and well written.

Thank you very much for your detailed revision. We believe that the modifications you suggest have improved the manuscript and the robustness of our analysis. Please, find below the responses to your comments.

However, the fairly coarse choice of tagging regions, as well as the exclusive selection of AR cases introduces limitations that are currently not well addressed. A more careful and balanced discussion of the results and implications from this study are thus advised. I also see further issues with the proposed tuning and with regards to some parts of the literature detailed below that the authors should address when preparing a revised manuscript.

We are aware of some limitations of our study, particularly regarding the exclusive selection of AR cases. Because of this, we have softened the language in the abstract and conclusions and recognized them in the discussion. Moreover, we conducted additional experiments to address some of the issues you encountered, which will appear in the response to your major and specific comments.

Major comments:

1. Coarse definition of tagging regions. The authors subdivide the hemispheric land and ocean into 9 sectors, separated along 30 N and S. This allows only for a very coarse distinction between ocean basins and continental areals and the boundaries. As the Lagrangian diagnostics are showing, the majority of sources are located at different regions within the same ocean basin. As a consequence, the RMSE computed here only picks up the outermost differences. An example for this is seen for the Greenland AR, where the structures in the North Atlantic region widely differ between the two Lagrangian approaches. The current tagging setup misses these differences entirely, and exclusively focusses on the fringe of the moisture sources. There are probably two ways to approach this deficiency: One is to increase the number of tracer subdivisions depending on every case, adding complexity to the study, but providing more sharpness in the tagging simulation (e.g. using a setup similar to Sodemann and Stohl, 2013). The other way is to openly address this deficiency in the study design, and adjust the discussion to be more nuanced, and formulate the conclusions more carefully.

We agree with the reviewer that our selection of source regions overlooks some differences that may exist between both Lagrangian approaches, this being evident in the Greenland case. This selection is based on the fact that for many applications in the field of moisture sources we are only interested in the basin as a whole and not so much in the contribution of the different sub-areas of the basin. An example is the calculation of continental recycling; in that case we are interested in the contribution of the continents as a whole. Another recurring example with atmospheric rivers is the separation between tropical and non-tropical moisture inputs, for which we only need two sources.

However, for comparison purposes, we acknowledge that this division of sources may substantially affect the results, so we have conducted an additional experiment.
Specifically, for each AR case we divided the tropical and extratropical ocean in which it is located into two additional sections (thus four sources for each basin) and repeated the comparison with WRF-WVTs using only six source regions for each case: the four sources for the corresponding ocean, the next most important source region (continental in all cases except for the Andes) and the rest of the world. This is a
slightly coarser configuration than that of Sodemann and Stohl, (2013), but it overlooks much fewer differences than the original one. For example, in the Greenland case this new setup reveals that the correct distribution of moisture sources - in relation to WRF-WVTs - is that of UTrack, as shown below (UTrack is now referred to as DB99). In particular, WaterSip overestimates the contribution from the northernmost part of the
basin, i.e. the region closest to Greenland.

[Figure]

Nevertheless, we found that the main conclusions of our study do not change with this new configuration; when the errors are averaged to yield a single deviation per event (e.g. Fig. 5), the values obtained are very similar to the old ones. Thus, this experiment
validates our results, and is now included in the Supplement, where we have added the following figure with the results of Fig. 5 and 7 for this new selection of source regions. Additionally, we briefly refer to this experiment in the main manuscript just after discussing the appropriateness of the selection of source regions, in the last paragraph of Sect. 3.2.2.

[Figure]

a) New selected source regions b) WaterSip c) Dirmeyer and Brubaker, (1999) methodology

2. Biased selection of cases. The study includes five AR cases from different parts of the world. All cases are thus potentially related to a large amount of long-range transport. While this selection in itself is no matter of concern, proposing a general tuning of the Lagrangian methods based on a selection of long-range transport cases only is problematic, as it may introduce biases during cases of more local precipitation sources (e.g. convective summertime precipitation, weaker precipitation cases). The focus on AR cases only, and the limitations following along with that, should be more clearly highlighted in the title, abstract, and conclusions.

We agree with the reviewer and therefore propose the following changes:

- New title: "Simple physics-based adjustments reconcile the results of Eulerian and Lagrangian techniques for moisture tracking in atmospheric rivers".
- Abstract. We have rewritten the last sentence of the abstract: "Although these modifications may need to be adjusted for other types of precipitation events,
our results demonstrate that Lagrangian techniques are a viable and compatible alternative to Eulerian water vapor tracers, and that the main discrepancies between the different methodologies can be derived from the obviation of basic physical considerations that may be easily straightened out".
- Conclusions. We have recognized again that the proposed modification
(specifically, the threshold for release height) to the DB99 methodology (UTrack) may depend on the type of precipitation event when discussing the achieved improvements in RMSE.

3. Proposed tuning to the WaterSip method. The authors propose to introduce a relative humidity threshold in the WaterSip method during the identification of
precipitation/moisture loss events en route. While such a proposal seems physically plausible at first, there are some downsides as well. Importantly, a moisture loss can be due to one of two reasons, either removal of water vapour from the atmosphere due to condensation and precipitation, or due to the mixing with drier air masses. The second case will be necessarily ignored in unsaturated situations if a relative humidity
threshold is introduced as proposed here. Ignoring the lowering of specific humidity due to mixing can then lead to an over-accounting of the moisture sources, i.e. a larger amount of uptakes are assigned to the specific humidity of the air parcel that are contained within. Duetsch et al. 2018 proposed a distinction between mixing events and rainout events. However, both types of situations still need to be part of the
accounting method to be physically plausible.

We were not aware that the modification we are proposing had already been introduced in a previous study. Therefore, in the revised version we acknowledge that it is an existing and used modification to the WaterSip method. This is recognized both in the methodology and in the conclusions. To be more explicit, in the latter we now state
at the beginning of the third paragraph: "In the case of WaterSip, we assessed a modification already applied in Dütsch et al., (2018) and Cheng and Lu, (2023)". Regarding the decrease in specific humidity due to the mixing of dry air, in our opinion, what the relative humidity filter does is precisely to prevent these decreases from being attributed to precipitation, which would be incorrect.

A more conventional tuning of the WaterSip method is to change the specific humidity thresholds and the time step. While the authors have tested different time steps, the specific humidity threshold has been set to a quite low value compared to literature (a common value is 0.2 g kg-1 6h-1). The specific humidity threshold will have a similar effect as the RH threshold, and is justified by interpolation errors in the offline
approach. Can the authors report how sensitive the moisture sources are, and thus the RMSE values to a variety of changes in the specific humidity threshold?

Following the reviewer's suggestion, we have performed several experiments in which we modify the specific humidity threshold and found almost no changes. Regarding the time step, there is a clear dependence on that parameter, as explicitly shown in the
Supplement. In the revised version, we have changed the standard setup to be that of the literature ($0.2$ g kg$^{-1}$ in a 6 h time interval) and included in the main manuscript the dependence of the average RMSE on both the specific humidity threshold (setting dq to 0.01, 0.05, 0.1, 0.2 and 0.3 g/kg) and time step (1 h, 3 h and 6 h). This shows that the optimal choice is that of the literature, although changing dq is not very relevant. All this additional information has now been included in a revised version of Figure 5, by adding three more panels reflecting the dependence of the average RMSE on both parameters (one for each of the No ABL, ABL and RH configurations).

Ultimately, I think one also has to acknowledge that offline trajectory methods do have their inherent limitations, both from the computation of trajectories, and the specifics of the moisture source diagnostic, which are sort of the price for the lower computational expense, and the more detailed spatial information on the source location. Knowing different methods' limitations may be in the end more valuable than tuning methods towards an expected or desired outcome for a specific type of cases. Maybe the authors could reflect on this perspective in their discussion and conclusions?

We agree that knowing the limitations of the tool being used is always essential in order to make a proper interpretation of the results. However, we believe that, in the long run, merely describing the limitations of tools does not make them any better, which is precisely what we are trying to do here. Finally, it is worth stressing again that the changes we are proposing have a physical basis, not a mere tuning of methods to match each other.

4. Title, abstract and conclusions appear too wide-ranging. As partly commented in the points above, the present study has limitations from the method design with respect to tracer setup and case selection, and the tuning of Lagrangian methods can lead to inconsistencies in the method. The discussion throughout the manuscript should be more nuanced and balanced by taking up these limitations. In particular the abstract is now formulated in a very definite, concluding language, which does not seem justified in the light of the limitations mentioned above. The title also suggests to a superficial reader that studies should generally apply the proposed tunings, but their general validity is questionable, or is at least not generally established. In particular the aspect of AR case selection could be included in the title. The study design with coarse tagging regions does in my view not 'reconcile' different approaches, but is rather a tuning using a particular choice of parameters. Maybe the title could be rephrased in terms of sensitivity, and mention the importance of long-range transport for the examined cases?

We have already mentioned how abstract, title and conclusions are reformulated to reflect that our study is focused on AR-related precipitation events. Regarding the selection of source regions, we believe that the additional experiments we have conducted, together with the new figure in the Supplement and its corresponding discussion in Sect. 3.2.2 give validity to our results. Nevertheless, in the same discussion we explicitly acknowledge that a different choice of source regions could be better for comparison purposes.

5. Use of literature. There are some citations of previous tagging studies that are missing or could be valuable to add. There are also some wrong citations (Lagrangian method cited in Eulerian context). These publications are listed in the detailed comments below.

We have updated the bibliography and citations with your comments, and this is reflected in the revised version of the manuscript. Thank you very much for that.

**Detailed comments**

L. 21: What unit does the RMSE have, is this in percent, or a fraction?

The RMSE has the same units as the precipitation fractions, we are expressing them as percentages. We have corrected the manuscript indicating these units where appropriate.

L. 22: "narrowly superior performance": How significant are the differences of less than
1 (%?) between both methods considering all sources of uncertainty?

We agree with the reviewer that this difference is not significant, but this statement is no longer in the new version of the manuscript.

L. 23: Maybe clarify that this is a relative improvement, since the RMSE appears to have the same units. The 50 % relative improvement could be misleading, both
because the untis are the same as for the RMSE, and given the overall quite small RMSE difference. Can the overall result be presented more balanced and objective here?

In order to avoid misunderstandings, we use now the mean absolute error skill score (MAESS) to present the main results of our study in the abstract.

L. 24: I think this conclusion statement is going too far. The selection of cases and limitations in the setup does not allow this conclusion. Expressed more neutrally, the sensitivity test and tuning performed here increase the amount of long-range transport detected from the Lagrangian methods. There is not sufficient evidence presented supporting that the tuning is valid generally in all cases. Maybe instead it could be
emphasized that the overall approach of using a Eulerian tagging setup to validate Lagrangian methods is promising, but needs further refinement for generally valid modifications.

We agree with the reviewer in that we should soften the language, especially given our focus on AR events. Because of this, we have modified the abstract and mentioned
that the changes we are proposing may need to be adapted for other types of precipitation events. Moreover, we have emphasized that Lagrangian moisture tracking techniques are a real alternative to Eulerian water vapor tracers. All these changes were already highlighted in our response to the major comment 2.

L. 35 and elsewhere: It is customary to sort references by year of publication. Consider
adding Yoshimura et al., 2004 to this list. Sodemann and Stohl (2009) is a Lagrangian study, did you mean to cite here Sodemann et al. (2009)?

We thank the reviewer for its clarification, we meant to cite here Sodemann et al., (2009). We have also added Yoshimura et al., (2004), and sorted references by year of publication.

L. 36: "Lagrangian transport models": Lagrangian transport models are the general category of models that simulate airmass transport. To be more specific to the case here, consider rephrasing as "Lagrangian moisture source diagnostics".

We have reformulated "Lagrangian transport models" as "Lagrangian moisture source diagnostics" in the revised version of the manuscript.

L. 39: I do not know of an existing online implementation of a Lagrangian moisture source diagnostics. The offline/online distinction can however be made regarding the tagging and Lagrangian methods.

We only state that there exists another possibility to classify moisture source diagnostics, and this allows us to introduce the difference between offline/online
methods, which is used later.

L. 40: "most academics often use": this point is debatable, there exist a range of studies that do make such comparison efforts.

We agree with the reviewer in that there are different studies that do make such efforts. However, it is also true that the majority of studies use a single model.

L. 41: "results can be highly discrepant": Winschall et al., 2014 does not provide highly discrepant results, at least that is not what is said in this paper. Please rephrase to do justice to the actual state of the literature, and to better clarify the intent and actual novelty of this study. In this context, please also consider the book chapter of Sodemann and Joos (2021).

We have reformulated "results can be highly discrepant" to "results may not be in agreement", as we understand that although Winschall et al., (2014) do not provide highly discrepant results, they do show that tropical contributions calculated in one way or another, for example, may be clearly different.

L. 46: Consider adding references to the original AR studies in this context.

We have added a reference to Zhu and Newell, (1998) following the reviewer's suggestion.

L. 47: This statement does not seem to do justice to the existing literature. See Sodemann and Stohl (2013) for a tagging study focused on AR events, as well as Stohl et al. (2008) for a study with Lagrangian methods. There are also a range of studies
from other regions and locations (e.g. Terpstra et al., 2021, Bonne et al., 2015). Please update this statement in light of existing studies, and clarify what this study adds to the existing literature. Please also take notice of the book chapter about AR moisture budgets (Sodemann et al. 2020). What is meant by "go beyond the identification of moisture sources to quantify them?"

We have reformulated this paragraph to include the references that the reviewer provides, in order to improve the presentation of the state of the art on moisture sources in atmospheric rivers. Regarding the sentence "go beyond the identification of moisture source to quantify them", we wanted to highlight that the studies that quantify the relative importance of different source regions focus on individual cases, and thus
the debate on the origin of moisture in ARs is not yet completely closed. This is now clarified in the revised version of the manuscript.

L. 55: There are two aspect here that are a little bit mixed together. One is that the tagging simulation is also only a model representation of the actual water cycle in nature. At the grid resolution of the model (here 20 km horizontally), a large spectrum
of the processes affecting the water cycle are parameterized. I assume that also a deep and shallow convection parameterisation (which one?) has been used in the Eulerian model simulation. Obviously, the model will thus not be identical with nature. However, the approach and argument of the present study is, as I understand it, that the tagging water cycle and the Lagrangian methods are internally consistent, even if the tagging results differ from nature. This is important, as the authors write, since the source information that is being sought after is not directly available from observations.

We agree with the reviewer and have now reworded the discussion from L55 on to better separate the idea that using Eulerian water vapor tracers to compare with and guide us in the process of assessing Lagrangian models is independent of the fact that
they reflect reality more or less accurately. When compared with the real world, the tracer results are as good as the WRF simulation is with respect to the real atmosphere, but in the model world they can be considered as observations. As for the parameterizations in the WRF simulations, we explicitly state them in the methodology of the revised version of the manuscript, and they will also be mentioned in the
response to another comment further down.

L. 57: Another important limitation of the tagging approach, which also becomes apparent in this study, is the requirement to predefine moisture sources in this forward calculation approach. If more spatial detail is required, the computational overhead multiplies and can become prohibitive. In contrast, the Lagrangian backward
approaches provide spatially detailed information, that can be more easily interpreted, for example in terms of the physical processes related to weather systems. This discrepancy between both approaches is important to mention here.

We thank the reviewer for his suggestion. In the revised manuscript we explicitly mention this drawback of the Eulerian water vapor tracers, by adding the following
sentence: "Additionally, the amount of information they provide is limited, as the moisture source to be tagged needs to be predefined". In addition, we also mention that Lagrangian methods provide gridded information, which is another advantage, along with their higher computational efficiency.

L. 59: Maybe mention here that the Lagrangian methods, being offline diagnostics,
require a range of assumptions and parameter choices to which these methods are sensitive. Your comparison framework allows to assess what biases exist with the different diagnostics, and how those are related to parameters and assumptions in the Lagrangian methods.

Once more we thank the reviewer for this suggestion. In the revised manuscript, we
have explicitly stated here that Lagrangian methods "are sensitive to a range of hypotheses and parameter choices, which significantly increases their uncertainty". These hypotheses and parameters are discussed further in the methodology, and the uncertainty they cause becomes apparent when introducing our physics-based modifications.

L. 61: "fully validated": I assume this relates to the internal consistency of the tagging approach. Validation can be misunderstood as a comparison to observable quantities. Please clarify/rephrase.

We have rephrased "fully validated" to "internally consistent".

L. 70: This is not correct, Sodemann et al. (2008) used trajectories from the
LAGRANTO model (Sprenger and Wernli, 2015).

We thank the reviewer for his clarification. We have included this corrected information in the revised version of the manuscript.

L. 73: "limited to highlighting ... large discrepancies": Please rephrase to do more justice to what is presented in the cited studies. For example, Winschall et al. (2014)

specifically investigated the basis of the boundary layer vs. free troposphere distinction in the WaterSip method.

What we mean here is that in these studies the authors did not propose any modifications to reconcile the results of the methods used. In any case, we agree that this sentence is misleading, as it implies that these studies only compared the tools,
when in fact they examined them more in depth, as in the case of Winschall et al., (2014). This sentence is reformulated in the revised manuscript to "While Winschall et al., (2014) show the complementarity of the results provided by the Eulerian and Lagrangian approaches, in Cloux et al., (2021) they specifically highlight the large discrepancies between the results provided by Lagrangian and Eulerian tools, although
they did not provide improvements to reconcile the different methodologies."

L. 77: "two of the most widely used" -> "two widely used"

We have corrected this sentence in the revised version of the manuscript.

L. 84: "vast majority ... force": there is no evidence supporting this statement. I don't think it is necessary to make this statement, adding reanalysis data is useful because
unlike forecast data, it includes analysis increments from data assimilation, see Fremme et al., 2023.

We have deleted this sentence because we have not checked it, but from our experience we believe that in most studies on moisture sources with FLEXPART, the model is forced with reanalysis data. A more thorough literature review would verify
this.

L. 93: It is certainly positive with different AR cases, but these cases are all long-range transport events. Can you add some clear justification for this focus in the introduction? Some of the writing makes the impression that you seek general validity, while the focus on AR events only seems in contradiction to this.

We focus on AR cases because in the future we plan to compute their moisture sources from a climatological perspective using one of the Lagrangian moisture tracking methodologies assessed here, which is mentioned in the introduction. Although we agree that considering other type of precipitation events would give more validation to our study, we are aware of the existence of a model intercomparison effort
in the moisture tracking community in which different kinds of cases are analyzed, so we decided to focus our study exclusively on ARs in order to interfere as little as possible with it.

Figure 1: Please add panel labels, and mention all figure panels in the caption. It would be a large advantage to have common color bars for the left and right column panels
each. For precipitation amount, it is quite common to use a categorized color bar to that end. This would avoid the saturation of the color scale that now seems to occur.

We have added panel labels and a reference to the geographic region in which precipitation is tracked and used a categorized color bar as suggested.

Table 1: Could this table include information about the total rainfall amount of these
events in the model and maybe observations? Is it correct that the two last events have the same date and time, but different regions?

We thank the reviewer for his suggestion. We have included two more columns in Table 1 with the precipitation in mm from the WRF simulations and ERA5 reanalysis.

Regarding the last question, there was a typo, the correct initial date and time for the Greenland case is 2012-07-09 12.

L. 119: What has been used in terms of deep and shallow convection, microphysics schemes in the WRF simulation?

In the WRF simulations, the main parameterizations used were the Yonsei University (YSU) for the boundary layer, the WRF single-moment-6-class (WSM6) for
microphysics, and the Kain-Fritsch for convection. In the revised version, we have moved this information from the supplement to the main manuscript, specifically at the end of the first paragraph of Sect. 2.2.

L. 121: Which fields have been nudged, only winds or also specific humidity? How does the nudging affect the tagging? The authors emphasize the importance of the
nudging, but actually I think for the study objective it does not make a difference if the results resemble the actual events closely or not.

Apart from the winds, temperature and geopotential height are also nudged, and this will be mentioned in the revised version of the manuscript. Humidity cannot be nudget because otherwise it would not be conserved. As to whether nudging is important or
not, it is crucial to keep the WRF model result close to reanalysis throughout the simulation, which is what later makes comparing with FLEXPART forced with ERA5 data meaningful.

L. 126: What is meant by this statement, and how does it relate to this citation? Consider maybe citing Gimeno et al., 2021 here.

We meant to cite van der Ent and Tuinenburg, (2017), instead of van der Ent and Tuienburg, (2013). In the first case they explicitly show the positively skewed probability density functions for the residence time of atmospheric water vapor. This justifies the long duration of our simulations, 30 days. This typo has been corrected in the revised version of the manuscript.

L. 133: If QFX is assimilated from ERA5, this can introduce an inconsistency into WRF due to differences in resolution. What is the reason for this choice? How different are the results when using the WRF-internal evaporation flux?

Before assimilating QFX from ERA5 we have interpolated the reanalysis moisture flux to the WRF domain. This was something that came up when we started to compare the
results of WRF-WVTs with those of both Lagrangian methodologies directly using simulations from FLEXPART forced with ERA5, but in the end we have found that it does not make a big difference.

L. 138: Is a time interpolation used here?

No, in our case we are not using a time interpolation, as we use hourly evaporation
data from ERA5 reanalysis.

L. 149: It may be useful to have some basic information in the main manuscript, such as the chosen parameterisation schemes, and the fact that simulations are hemispheric (?)

We agree with the reviewer that moving the information about the chosen
parameterizations from the supplement could be useful, and we have done it in the revised version of the manuscript. Specifically, the chosen parameterizations are now mentioned at the end of the first paragraph of Sect. 2.2. Regarding the domains, simulations are not entirely hemispheric since the polar caps are excluded, as Fig. 2 shows. The domain in the Northern Hemisphere, for example, covers latitudes from 0º to 65ºN.

Figure 2: The source regions are very large in comparison to the scale of the moisture sources revealed by the Lagrangian diagnostics. A separation into e.g. 10 degree latitude bands or latitude-longitude boxes could allow for a much more detailed comparison and evaluation of Lagrangian models in the Eulerian framework.

This question has already been answered in response to the reviewer's major comment 1.

L. 166: "FLEXPART assimilates hourly data": FLEXPART does not perform data assimilation, please rephrase. It is not clear what is said from this sentence, the previous section described WRF, not FLEXPART. How exactly is FLEXPART run with WRF? Maybe some of the details from the supplement could be moved to the main text. In particular, it is important to describe how particles were initiated and released, and if any convection parameterisation was active in FLEXPART.

We have changed "FLEXPART assimilates hourly data" by "FLEXPART-ERA5 reads hourly data". FLEXPART is run with WRF by using the FLEXPART-WRF model, which is able to ingest input data from the output of the WRF model. Finally, we agree with the reviewer that moving some details from the supplements could be useful. Specifically, in the revised version of the manuscript we explicitly state how parcels are released in the FLEXPART simulations: "parcels are released using the domain filling option over the black boxes in Fig. 1 such that they are vertically distributed following the density profile".

L. 174: "it starts by assuming": This sentence and the following sound a bit strange. What you describe seems to be the basic idea of Lagrangian analysis, which is not particular to WaterSip. It would be useful to cite Stohl and James (2004) in this context, or shorten the section altogether, because all of this has already been said elsewhere.

We have reformulated this paragraph to retain only the most essential information, i.e., the following sentence: "the atmospheric column over the region where precipitation occurs is filled with air parcels, and that their trajectories contain information about their location and specific humidity at 6-hourly intervals for the previous days".

L. 181 to 207: This section repeats a lot of information that is found in the original publication, and is not necessarily more easy to follow. I recommend limiting this to the most essential parts of the method which are modified here.

These paragraphs have been rewritten in the revised version of the manuscript. Specifically, we briefly explain how WaterSip works, focusing on how the spatial distribution of moisture sources is obtained, and the remainder of that paragraph will be shortened. For example, the explanation of the discounting algorithm is omitted, as it can be found both in the original study of Sodemann et al., (2008) and in the Supplement. Once the methodology is introduced, we present some modifications of WaterSip previously used in the literature corresponding with the different configurations that we test in our study. Here we acknowledge Dütsch et al., (2018) and Cheng and Lu, (2023) for introducing the modification that we later demonstrate reduce biases in WaterSip, "A less common  modification is to filter the specific humidity decreases, such that previous contributions are only discounted if a specific humidity decrease occurs and the relative humidity of the parcel is higher than 80 % (Dütsch et al., 2018; Cheng and Lu, 2023)". Finally, we also point out here the dependence of
WaterSip on the specific humidity threshold and time step.

L. 188: The threshold value has been repeatedly shown to be a key sensitivity
parameter (e.g., Sodemann and Stohl, 2009; Fremme and Sodemann, 2019). In
addition, this value is on the very low end, that has been previously recommended for
Arctic studies. How sensitive are your baseline results to this choice? To be in line with
literature, I recommend a delta q of 0.2 for a 6h time interval.

As we mentioned in our response to the major comment 3, in the revised version of the
manuscript Fig. 5 includes a sensitivity experiment in which we change both the
threshold value and the time step. This demonstrates that our results are not very
sensitive to the threshold value, but they are to the time step. Additionally, for the basic
results we now use the recommended setup of 0.2 g/kg for a 6 h interval, instead of
0.05 g/kg for a 3 h interval as before.

Regarding section S3.1 referenced here, I wonder about what the role of this
mathematical description is for the manuscript. There seem to be some arguments
about correspondences between the two Lagrangian methods mathematically, but
conceptually the two are quite different (e.g., well-mixed properties of the atmosphere).
Section S3.1 could benefit from a closer connection to published literature to clarify its
purpose. Does this section describe what has been published before, but
mathematically in a common framework?

The original idea of section S3.1 was to unify the mathematical framework of the
methodologies to show that both use a linear discounting, so that they are equivalent
from a mathematical (and computational) perspective. However, it is true that they are
conceptually different. Still, since we present less details on the WaterSip methodology
in the revised version of the manuscript; we believe it makes sense to keep this section
of the Supplement as is.

L. 209: The authors refer to the UTrack method as the Dirmeyer and Brubaker (1999)
implementation they use. However, as noted in L. 218, UTrack computes its own
trajectories. Is it then not more correct to refer to the second model as the Dirmeyer
and Brubaker (1999) method? What really distinguishes the approach used here from
UTrack and Dirmeyer and Brubaker (1999), respectively?

We agree with the reviewer that it is more correct to refer to the second model as the
Dirmeyer and Brubaker, (1999) method (we will use the abbreviation DB99). We have
updated the manuscript with this modification. The approach used here is the same as
UTrack or DB99, but using FLEXPART trajectories instead of computing them. To do
that we also need to consider how parcels are vertically released, as in the case of the
DB99 methodology or UTrack they follow the humidity profile. Since in our case parcels
are vertically distributed following the density profile, we had to weight the moisture
sources of each parcel using their specific humidity. These important remarks are now
included in the methodology, at the end of the paragraph in which DB99 is explained:

"However, in our case we use FLEXPART-ERA5 and FLEXPART-WRF trajectories at
hourly resolution and implement only the diagnostic tool to compute the moisture
sources for precipitation. Thus, since in our simulations parcels are vertically released
following the density profile, we weight the contribution of each parcel by its specific
humidity to match the DB99 methodology."

L. 252: It has been common to initialize the domain at model time zero with all water vapour currently in the domain to achieve 100% accounting. Has this been tested here?

No, we have not tested that. This is evaluated in Insua-Costa and Miguez-Macho, (2018), where they show the internal consistency of WRF-WVTs.

L. 255: This is not correct. The Dirmeyer and Brubaker (1999) method stops accounting evaporation when 100% have been reached. The WaterSip method does not generally reach 100% (see Sodemann et al., 2008).

We agree with the reviewer that the WaterSip method does not generally reach 100 %. We meant that WaterSip typically reaches 100 % when the simulation time is 30 days and all uptakes are considered, but this is something that we have not shown. Because of this, we have deleted that sentence from the manuscript.

L. 256: What is meant by "the bias will also be calculated after adjusting for these precipitation fractions"? This scaling should be explained in the methods section. Why is a scaling necessary at all? Is it not more correct to compare the actual identified fractions? What about comparing amounts rather than fractions?

We agree that scaling is not strictly necessary. In the revised version of the manuscript, the results correspond to the comparison of the actual precipitation fractions without scaling, with the exception of the "ABL" and "RH" configuration, as in these cases the attributed precipitation is typically much lower or much higher than 100 % if we do not scale the precipitation fractions. On the other hand, we discarded comparing absolute quantities instead of fractions, as we can then be unconcerned whether in WaterSip the diagnosed precipitation, which comes from the specific humidity decreases at the time and location of the rainfall event, matches the WRF precipitation.

L. 266: I think the reference to Winschall et al. 2014 is not justified in such a general statement as done here. Winschall et al. 2014 did a sensitivity test of different tagging approaches, and their conclusion was: "The results of the Lagrangian diagnostics are similar to the Eulerian results, with the fraction of remote versus local moisture sources lying in between the two realisations of the tagging technique."

We agree that Winschall et al., (2014) did a sensitivity test of different tagging approaches, instead of assessing the Lagrangian method. Because of this, we now omit this reference in that part of the manuscript.

L. 268: Is the RMSE expressed as a fraction as in Eq. (6) or in percent?

We have corrected the manuscript expressing the RMSE as percentage, as the RMSE should have the same units as the precipitation fractions.

L. 274: It is interesting to note that the biases of the UTrack method are different. Why is that the case? The US West Coast case for example, UTrack has a lower performance.

This is because in the US West Coast case the moisture sources are highly dependent on altitude. This can be inferred from Fig. 7, where we can see that the RMSE for the Dirmeyer and Brubaker, (1999) methodology changes a lot with the threshold for the release height of parcels.

Figure 4: It is not possible the read the numbers printed in white on a light colour background.

We have updated this figure (and also Figure 9), so that all numbers can be easily read.

L. 279: I do not see a value of 29.6 in Fig. 3, nor of 14.88 for the Tropical Atlantic in Fig. 4. Is this example part of the supplement information? How does the scaling impact the results here?

We agree with the reviewer that the 29.6 value could not be easily deduced from Fig. 4., as the biases shown in Fig. 4 were computed scaling both the precipitation fractions in Fig. 3 and the precipitation fractions calculated with the Lagrangian methodologies. In the revised version of the manuscript, we do not scale the results (with the exception of the ABL and RH configurations, where the attributed precipitation is far from 100 %) so the precipitation fractions in the Lagrangian diagnostic tools can be easily deduced from Fig. 3 and 4.

L. 291: This statement applies to both Lagrangian methods. Before proceeding to tune the methods, it would be useful to quantify the overall bias of the Lagrangian vs. Eulerian methods, maybe at the end of Sec. 3.1, potentially as a function of distance from the arrival location. It may also be worthwhile to comment on the overall consistency of the results from the 3 approaches here. It would also be interesting to 810 know more about the sensitivity here already regarding the specific setup you chose. How different are the errors/biases for a time interval of 6h, and when increasing the specific humidity threshold to 0.2 g kg-1 6h (or more)?

We agree with the reviewer that the statement applies to both Lagrangian methods, so we mention now WaterSip and the DB99 methodology in that sentence. Regarding the overall bias as a function of distance from the arrival location, we understand that this 815 information is somehow implicit in the estimation of moisture sources, which generally gives more weight to closer regions. Finally, the dependence on the time interval and specific humidity threshold are discussed in the revised version of the manuscript and included in an updated version of Fig. 5.

L. 295: What is presented here is exactly the argument for introducing a specific humidity threshold in WaterSip. So this needs not be formulated as a (new) hypothesis, it is part of the known uncertainty of the WaterSip diagnostic.

We have rewritten this sentence to explicitly mention that our hypothesis is that fluctuations penalize remote contributions. We know that the introduction of a specific 825 humidity threshold tries to cope with noise in this diagnostic tool, but to our knowledge this specific implication of fluctuations has not yet been formulated. Thus, we change "we conjecture that non-physical humidity fluctuations" to "we explore the hypothesis that non-physical humidity fluctuations" at the beginning of Sect. 3.2.1, and explain better why non-physical negative changes in specific humidity penalize earlier 830 contributions at the end of the same paragraph, by adding "as the error caused by a single fluctuation affects all previous contributions, so that the early moisture uptakes will be affected by many more non-physical changes".

L. 315: This distinction and modification have already been proposed by Dütsch et al., 2018 (Their Sec. 3.2). However, it is important to note that mixing with dry air can also 835 lead to a specific humidity decrease. By only allowing for precipitation events to decrease specific humidity, a bias is intoduced into the method. This can also result in an over-accounting of sources (more than 100% of moisture accounted for).

This has already been answered above (see reviewer's major comment 3).

Figure 8: Comparing the UTrack results with the corresponding results from WaterSip in Fig. 7, it is very interesting to note how different the spatial maps are from the two methods. UTrack basically shows almost no sources at all in the vicinity of Greenland. While we don't know which one of the results is more correct, this difference is not picked up by the comparison to water vapour tagging in the present setup. This fact points to the current tracer setup being not sufficiently sharp (or detailed enough) to resolve and quantify such differences.

This has already been answered above (see reviewer's major comment 1).

**References**

Bonne, J.-L., Steen-Larsen, H. C., Risi, C., Werner, M., Sodemann, H., Lacour, J.-L., Fettweis, X., Cesana, G., Delmotte, M., Cattani, O., Vallelonga, P., Kjær, H. A., Clerbaux, C., Sveinbjörnsdóttir, A. E., and Masson-Delmotte, V., 2015: The summer 2012 Greenland heat wave: In situ and remote sensing observations of water vapor isotopic composition during an atmospheric river event, J. Geophys. Res., 120, 2970-2989, doi:10.1002/2014JD022602.

Cheng, T. F. and Lu, M.: Global Lagrangian Tracking of Continental Precipitation Recycling, Footprints, and Cascades, J. Clim., 36, 1923–1941, https://doi.org/10.1175/JCLI-D-22-0185.1, 2023.

van der Ent, R. J., Tuinenburg, O. A., Knoche, H.-R., Kunstmann, H., and Savenije, H. H. G.: Should we use a simple or complex model for moisture recycling and atmospheric moisture tracking?, Hydrol. Earth Syst. Sci., 17, 4869–4884, https://doi.org/10.5194/hess-17-4869-2013, 2013.

van Der Ent, R. J. and Tuinenburg, O. A.: The residence time of water in the atmosphere revisited, Hydrol. Earth Syst. Sci., 21, 779–790, https://doi.org/10.5194/hess-21-779-2017, 2017.

Dirmeyer, P. A. and Brubaker, K. L.: Contrasting evaporative moisture sources during the drought of 1988 and the flood of 1993, J. Geophys. Res. Atmospheres, 104, 19383–19397, https://doi.org/10.1029/1999JD900222, 1999.

Dütsch, M., Pfahl, S., Meyer, M., and Wernli, H.: Lagrangian process attribution of isotopic variations in near-surface water vapour in a 30-year regional climate simulation over Europe, Atmos. Chem. Phys., 18, 1653–1669, https://doi.org/10.5194/acp-18-1653-2018, 2018

Fremme, A. and Sodemann, H., 2019: The role of land and ocean evaporation on the variability of precipitation in the Yangtze River valley, HESS, 23, 2525–2540.

Fremme, A., Hezel, P. J., Seland, Ø., and Sodemann, H.: Model-simulated hydroclimate in the East Asian summer monsoon region during past and future climate: a pilot study with a moisture source perspective, Weather Clim. Dynam., 4, 449–470, https://doi.org/10.5194/wcd-4-449-2023, 2023.

Gimeno, L., Eiras-Barca, J., Durán-Quesada, A.M., Dominguez. F., van der Ent, R., Sodemann, H., Sánchez-Murillo, R., Nieto, R. and Kirchner, J. W.: The residence time of water vapour in the atmosphere. Nat. Rev. Earth Environ., https://doi.org/10.1038/s43017-021-00181-9, 2021.

Sodemann, H., Schwierz, C., and Wernli, H.: Interannual variability of Greenland winter precipitation sources: Lagrangian moisture diagnostic and North Atlantic Oscillation influence, J. Geophys. Res. Atmospheres, 113, D03107,
https://doi.org/10.1029/2007JD008503, 2008.

Sodemann, H. and Stohl, A.: Asymmetries in the moisture origin of Antarctic precipitation, Geophys. Res. Lett., 36, https://doi.org/10.1029/2009GL040242, 2009.

Sodemann, H. and Stohl, A.: Moisture Origin and Meridional Transport in Atmospheric Rivers and Their Association with Multiple Cyclones, Mon. Weather Rev., 141, 2850–
2868, https://doi.org/10.1175/MWR-D-12-00256.1, 2013.

Sodemann, H. and Joos, H., 2021: Numerical methods to identify model uncertainty in: Ólafsson, H. and Bao, J.-W. (Eds), Uncertainties in Numerical Weather Prediction, Elsevier, 309-329, doi: 10.1016/B978-0-12-815491-5.00012-4, 2020.

Sprenger, M. and Wernli, H.: The LAGRANTO Lagrangian analysis tool – version 2.0,
Geosci. Model Dev., 8, 2569–2586, https://doi.org/10.5194/gmd-8-2569-2015, 2015.

Stohl, A., Forster, C. and Sodemann, H., 2008: Remote sources of water vapor forming precipitation on the Norwegian west coast at 60°N - a tale of hurricanes and an atmospheric river, J. Geophys. Res., 113, D05102, doi:10.1029/2007JD009006.

Terpstra, A., Gorodetskaya, I. V., Sodemann, H.: Linking sub-tropical evaporation and
extreme precipitation over East Antarctica:an atmospheric river case study, J. Geophys. Res., 126, https://agupubs.onlinelibrary.wiley.com/doi/10.1029/2020JD033617, 2021

Yoshimura, K., Oki, T., Ohte, N., and Kanae, S.: Colored Moisture Analysis Estimates of Variations in 1998 Asian Monsoon Water Sources, 気象集誌 第2輯, 82, 1315–1329,
https://doi.org/10.2151/jmsj.2004.1315, 2004.

Winschall, A., Pfahl, S., Sodemann, H., and Wernli, H.: Comparison of Eulerian and Lagrangian moisture source diagnostics – the flood event in eastern Europe in May 2010, Atmospheric Chem. Phys., 14, 6605–6619, https://doi.org/10.5194/acp-14-6605-2014, 2014.

Zhu, Y. and Newell, R. E.: A Proposed Algorithm for Moisture Fluxes from Atmospheric Rivers, Mon. Weather Rev., 126, 725–735, https://doi.org/10.1175/1520-0493(1998)126<0725:APAFMF>2.0.CO;2, 1998.

---

## Referee Report (RR1)

**Review of "Simple physics-based adjustments reconcile the results of Eulerian and Lagrangian techniques for moisture tracking in atmospheric rivers" by Crespo-Otero et al.**

This is my second review of this manuscript. The authors addressed most of my comments to the previous submission in a very constructive way and adjusted the manuscript accordingly. Still, I was very surprised that the authors state they find no sensitivity to the specific humidity threshold Δq. The authors claim throughout the text that they use the method "WaterSip", which in my understanding refers to a specific computer code implementation of the Sodemann et al. (2008) method. In an attempt to understand where these differences come from, I have made a comparison of two cases with the code that was used in previous studies for cases investigated here. It turns out that the results I get from running WaterSip V3.2, based on a global run with 5 million particles using ERA-Interim (Läderach and Sodemann, 2016) look entirely different for the Greenland case (Fig. 1), and only somewhat similar for the South Africa case (Fig. 2). The differences are so large that I do not think this could be due to the WRF simulation setup, and I rather suspect that the way the Sodemann et al. (2008) algorithm has been implemented differs in some way from the original. In fact, the results in Fig. 1b suggest that only a small contribution comes from the North Atlantic domain as defined by the authors, and a larger part from the North American continent. As the findings from the Greenland case are a cornerstone in the argumentation throughout the manuscript, this is a critical situation that needs to be resolved before publication.

Please note that WaterSip refers to a specific computer code that implements the Sodemann et al. (2008) method for FLEXPART and LAGRANTO with additional diagnostics. This software is now in public discussion (Sodemann 2025). In order to avoid confusing readers with regard to what software has been used, I would like the authors to not claim that they use WaterSip in their analysis, but rather their own implementation of an algorithm that is maybe inspired or resembles the Sodemann et al. (2008) method. The major differences that I see for the Greenland case imply that something else is done in the authors' code that leads to moisture uptakes identified close to Greenland in that case, and it would be misleading to assign these differences to either WaterSip or the Sodemann et al. (2008) method in itself. The lacking sensitivity to Δq is another hint at substantial differences in the author's code to what is done in Sodemann et al. (2008).

I still find the setup of the WRF moisture source tags overly coarse to make strong conclusions. In their reply, the authors state that they made a simulation with more finely resolved source boxes, but I do not see this discussed or shown in the manuscript, except for a figure in the supplementary material that is not mentioned anywhere (maybe I missed it). If the setup stays like this in the manuscript, it is in my opinion of critical importance to openly discuss the limitations of their approach in this study. For example, the boundary for the South Africa case at 30 deg N has the source footprint just south of that boundary (Fig. 2). This makes the verification statistics very sensitive so small meridional shifts in the source footprint, while really the AR case is one dominated by zonal advection. Thus, if the aim is to verify differences in the remoteness of sources, a setup with boxes over different longitudes would be needed to make the results more robust and help to support the conclusions, rather than a setup with different latitudes as is now shown in the supplementary material. All this may be still ok for showing the principle of how such a verification can be done, but such limitations need to be stated and discussed clearly to come to a well-justified overall assessment.

I also have a number of minor and technical comments below:

L. 14 and elsewhere: Instead of "WaterSip", which denotes a particular implementation of the Sodemann et al. (2008) method, refer to your own implementation of the Sodemann et al. (2008) algorithm here and elsewhere.

L. 37: This reference is missing from the reference section:

Sodemann, H., Wernli, H. and Schwierz, C., 2009: Sources of water vapour contributing to the Elbe flood in August 2002: A tagging study in a mesoscale model, Quart. J. Royal Meteorol. Soc., 135, 205-223, doi:10.1002/qj.374.

L. 43 and elsewhere: There is occasionally an extra comma before the bracket with references.

L. 60: remove extra '.'

L. 150: What are the units of QFX?

L. 152: The term "assimilate" is used here in a non-standard way, and can be confused with data assimilation (which I don't have the impression is meant here). Please rephrase to e.g., "ingest directly from ERA5" or the like.

L. 177: This should be cited as Sodemann and Stohl (2009) as in the references.

L. 183: Here it says hourly, in the sentence in L. 187 it says 3-hourly. Which one is correct?

L. 188: Give a typical pressure height of the topmost retained level.

L. 208: add "at arrival" after relative humidity

L. 222: 0.05 g kg 6 h$^{-1}$ is only ¼ of the typically used value for mid-latitudes of 0.2 g kg 6 h$^{-1}$

L. 262: "coming also": check sentence

L. 264: "basic results" -> "baseline"?

Figure 3: I don't really see a great value in using a figure instead of a table, which I would think would be clearer/more easy to read with bigger font and better contrast. Values could by highlighted by making them bold or italtic.

L. 285 and onward: The terms "good" and "bad" or "better" and "worse" could be more neutrally be rephased by using expressions such as "more consistend with reference" or "less consistent", or "larger deviation", "smaller deviation". There is not really a way of telling objectively here at what value a difference can be considered good or bad.

L. 320: Such fluctuations are always a remaining part of the uncertainties from offline trajectory calculations. This is why the Δq threshold has been introduced. Given that there is virtually no impact from the threshold on the results, as in Fig. 5 indicates that there could be an error in the way the method has been implemented.

Figure 5: The low contrast in some of the tables makes the numbers hard to read.

L. 342: It was not easy to follow the narrative here, please rephrase.

L. 353: important for what?

L. 358 and L. 362: significantly: rephrase if no statistical test has been performed

L. 369: Move to figure caption

L. 374: What means "the moisture field is less intense"?

Harald Sodemann

[Figure]

**Figure 1:** Moisture sources from WaterSip V3.2 extracted from the dataset of Läderach and Sodemann (2016) for the AR arriving in Greenland during a 48 h period before . (a) Combined boundary-layer and free-troposphere moisture uptakes (mm day$^{-1}$). (b) Boundary-layer moisture uptakes (mm day$^{-1}$). The WaterSip extraction was set up with a $\Delta q_c = 0.2$ g kg$^{-1}$ 6 h$^{-1}$ and a trajectory length of 20 days.

[Figure]

**Figure 2:** Moisture sources from WaterSip V3.2 extracted from the dataset of Läderach and Sodemann (2016) for the AR arriving in South Africa during . (a) Combined boundary-layer and free-troposphere moisture uptakes (mm day$^{-1}$). (b) Boundary-layer moisture uptakes (mm day$^{-1}$). The WaterSip extraction was set up with a $\Delta q_c = 0.2$ g kg$^{-1}$ 6 h$^{-1}$ and a trajectory length of 20 days.

**References**

Sodemann, H.: The Lagrangian moisture source and transport diagnostic WaterSip V3.2, EGUsphere [preprint], https://doi.org/10.5194/egusphere-2025-574, 2025.

Läderach, A. and Sodemann, H., 2016: A revised picture of the atmospheric moisture residence time, Geophys. Res. Lett., 43, doi:10.1002/2015GL067449.

---

## Author Response (AR2)

Dear authors,

Thank you for constructively and comprehensively addressing the concerns of the two reviewers. Your revised manuscript has been reviewed again by Harald Sodemann, but not the first reviewer, who unfortunately declined to review the manuscript again. In the referee report of Harald Sodemann, you will see that he is surprised by the lack of sensitivity to the specific humidity threshold. Could you please address this major concern of his, as well as the other minor comments and suggestions he provided? In addition, relevant data and code should be made accessible in accordance with ESD policy: "Data and model modifications used in the manuscript should be available in open-access repositories. See also the data policy web page on the ESD site for further information and options of repositories." The transparent sharing of code and data could potentially also help clarify questions regarding model code discrepancies, as raised by Prof. Sodemann.

In addition, as the first reviewer declined to review the revised manuscript, I took a detailed look at your revisions in his/her place. I think most of his/her comments are addressed really well. Further down, I would just like to follow up on a few of his/her concerns regarding clarity and provide some additional suggestions. Hope you find them useful.

Best wishes,
Lan Wang-Erlandsson

Thank you very much for continuing the work of the first reviewer and handling the interactive discussion. Following your suggestions and ESD policies, we will make our code accessible in an open-access repository. Please, find below the responses to your comments.

Introduction paragraph 1: Please consider explaining a bit more why understanding the moisture sources of atmospheric rivers (e.g., in the paragraph starting at L45) is important. This would help contextualise the paper and motivate its publication in an interdisciplinary Earth system science journal like ESD. The reason provided in L29-30 "how water is transported between different regions of our planet" is only a general and climatological rationale, and does not explain the focus on atmospheric rivers.

Thank you for the suggestion, we agree that the focus on atmospheric rivers (ARs) has not been completely justified so far. On the one hand, despite occupying only a small portion of the Earth's surface, ARs are responsible for ~90 % of the poleward water vapor flux across mid-latitudes (Gimeno et al., 2014; Zhu and Newell, 1998). In some regions of the world (particularly, extratropical and western coastal areas), they account for more than 30 % of the total precipitation (Ralph et al., 2020), and it has also been shown the close connection between ARs and extreme precipitation events (Ralph et al., 2006; Lavers and Villarini, 2013). On the other hand, although there are previous studies where they compute the precipitation sources for ARs, the problem is not completely closed. This is mainly because of the focus on specific regions and events, but also due to the use of different moisture tracking methodologies, which makes the

comparison difficult. Thus, by focusing on ARs we are addressing an important and unresolved problem relevant to the climate of a substantial portion of the planet. To better explain this in the manuscript we will change the first part of the second paragraph:

"The aforementioned techniques have been particularly used to identify moisture sources in precipitation events associated with atmospheric rivers (ARs). ARs are structures of enhanced moisture and intense water vapor transport in the atmosphere, typically located in the pre-cold frontal region of an extratropical cyclone (Ralph et al., 2005), and are responsible for the majority of the poleward water vapor flux across mid-latitudes (Zhu and Newell, 1998). Their large-scale nature, together with their connection to extreme precipitation events (Ralph et al., 2006; Lavers and Villarini, 2013), has made them the focus of several studies aimed at determining the origin of the moisture they carry —specifically, whether it is primarily transported from remote regions or local sources. This issue has been addressed using Eulerian water vapor tracers (Sodemann and Stohl, 2013; Eiras-Barca et al., 2017; Hu and Dominguez, 2019), Lagrangian techniques (Liberato et al., 2013; Ramos et al., 2016) or both (Bonne et al., 2015)."

L55: The overarching objective statement formulation could be sharpened to better explain what the study is trying to achieve and what insights it is hoping to gain. 'Comparisons' and 'adjustments' are more methods than end-goals in themselves. At L88 there's a good example of a clearer explanation of 'why' comparisons are made and 'what' adjustments are meant to achieve, but currently not tied to the main objective statement.

We agree with the reviewer that the main objective of our study is not well stated here. We will reformulate "compare and adjust two Lagrangian methodologies for the computation of moisture sources for precipitation" to "assess the differences and reduce the discrepancies between two Lagrangian methodologies for the computation of moisture sources for precipitation".

L60: typo

We will delete the extra punctuation mark in the revised version of the manuscript.

L56-L94: These paragraphs currently both describe and motivate the aims and methodology. For clarity and flow, you could consider adding a paragraph or subsection immediately after the Section 2 heading to provide the details of the methodological flow. This would allow the final paragraph of the introduction to focus on explaining the study aims and the overall structure.

Thank you very much for the suggestion. We agree that the three paragraphs between L56 and L94 both describe the objectives and briefly introduce the methodology used in this study. However, these paragraphs are closely connected: the second paragraph (L75-L85) discusses previous studies that employed a similar approach (specifically, the use of water vapor tracers as reference to evaluate a Lagrangian methodology), and the third builds directly on this by detailing which Lagrangian tools we assess and what is our ground truth. Given that the main focus of this paper is to understand the origin of discrepancies between moisture tracking techniques and rectify them (rather than identifying moisture sources), we believe that this contextual and methodological framing should be in the introduction.

L97 'a series of analyses': consider elaborating

We will reformulate this sentence for clarity. Specifically, to "Section 3 includes the main results of our study, where we first focus on the comparison of the results produced by the Lagrangian methodologies with those provided by the WRF-WVTs model, and then assess how the Lagrangian and Eulerian approaches can be brought into closer agreement."

L312-324: Please consider explaining what the term 'non-physical humidity fluctuations' refers to, especially as the term is not used by Sodemann et al. 2008. Sodemann et al. (2008) discuss: "other physical or numerical processes caused the moisture increase in the traced air parcel, such as convection, evaporation of precipitating hydrometeors, subgrid-scale turbulent fluxes, numerical diffusion, numerical errors associated with the trajectory calculation, or physical inconsistencies between two ECMWF analysis time steps". If the authors refer to the same processes as Sodemann et al. (2008), which seems to be the case, the term 'non-physical' would be confusing.

We agree with the reviewer that the term "non-physical" may be misleading. As in Sodemann et al. (2008), we refer here to processes other than surface evaporation or precipitation that may change the humidity of the parcel computed by FLEXPART. We consider that such processes should be excluded from the calculation of moisture sources for precipitation. In the revised version of the manuscript, we will replace the term "non-physical humidity fluctuations" to "humidity fluctuations" and explicitly clarify that we are referring to changes in the parcel moisture not caused by surface evaporation or precipitation.

L51-54 "This is reflected in the definition of AR given in the Glossary of Meteorology, where it is indicated that the sources of moisture can be tropical and/or extratropical (Ralph et al., 2018)." Please consider reformulating and/or elaborating. It is unclear how this definition constitutes a reflection of tracking model uncertainties. Also, please check the citations (the reference Ralph et al. 2018 does not lead to the Glossary of Meteorology, and the Glossary of Meteorology does not cite Ralph et al., 2018).

In this paragraph we are trying to explain why the problem of the moisture sources in ARs remains unresolved, despite the existence of several studies addressing individual cases or computing climatologies for selected regions. In our opinion, the statement "the sources of moisture can be tropical and/or extratropical" reflects this lack of information, as it encompasses nearly all possible combinations of tropical and extratropical contributions. Furthermore, by putting together all the information extracted from these studies, it can also be deduced that the issue is not completely closed from a global and climatological perspective.

Regarding the citations, Ralph et al. (2018) does not lead to the Glossary of Meteorology, but to the article where they explain how the AR definition was created. We will include a citation to the Glossary of Meteorology also in the revised version of the manuscript.

L262: typo?

We agree that the last sentence is not correctly written right now. We will change it to "Finally, in Sect. 3.3 we test the proposed modifications when the trajectories are generated by FLEXPART-ERA5, with input data from the ERA5 reanalysis. In this case, the additional fields required by the diagnostic tools (e.g., evaporation and precipitable

water in the case of DB99) are also taken from ERA5, rather than from WRF simulations."

L463: Consider reformulating the heading and/or first sentence to make it clear at a glance that the section addresses the validation of the adjustments.

We thank the reviewer for this suggestion. We will change the heading to "Extension of the proposed modifications to ERA5 forced simulations" in the revised version of the manuscript.

L578 "in global or climatological applications": Please check the formulation. The analyses focused on specific regions and events, rather than global or climatological moisture tracking.

Although our analysis focused on specific regions and events, it was not limited to a single geographical area or time period. In addition, the results consistently pointed to the same conclusions, which suggests that our findings are broadly applicable to precipitation events (at least those associated with ARs) anywhere on the globe. In this scenario, the modifications we propose may be applied in global or climatological studies. We will incorporate this clarification into the manuscript for greater clarity.

**References**

Gimeno, L., Nieto, R., Vázquez, M., and Lavers, D. A.: Atmospheric rivers: a mini-review, Front. Earth Sci., 2, https://doi.org/10.3389/feart.2014.00002, 2014.

Lavers, D. A. and Villarini, G. (2013). The nexus between atmospheric rivers and extreme precipitation across Europe. Geophysical Research Letters, 40(12), 3259-3264.

Ralph, F. M., Neiman, P. J., Wick, G. A., Gutman, S. I., Dettinger, M. D., Cayan, D. R., and White, A. B.: Flooding on California's Russian River: Role of atmospheric rivers, Geophys. Res. Lett., 33, https://doi.org/10.1029/2006GL026689, 2006.

Ralph, F. M., Dettinger, M. D., Schick, L. J., and Anderson, M. L. (2020). Introduction to atmospheric rivers (pp. 1-13). Springer International Publishing.

Zhu, Y. and Newell, R. E.: A Proposed Algorithm for Moisture Fluxes from Atmospheric Rivers, Mon. Weather Rev., 126, 725–735, https://doi.org/10.1175/1520-0493(1998)126<0725:APAFMF>2.0.CO;2, 1998.

**Answer to Harald Sodemann (R2) in the Interactive comment on "Simple physics-based adjustments reconcile the results of Eulerian and Lagrangian techniques for moisture tracking in atmospheric rivers" by Alfredo Crespo-Otero, Damián Insua-Costa, Emilio Hernández-García, Cristóbal López and Gonzalo Míguez-Macho**

This is my second review of this manuscript. The authors addressed most of my comments to the previous submission in a very constructive way and adjusted the manuscript accordingly. Still, I was very surprised that the authors state they find no sensitivity to the specific humidity threshold $\Delta q$. The authors claim throughout the text that they use the method "WaterSip", which in my understanding refers to a specific computer code implementation of the Sodemann et al. (2008) method. In an attempt to understand where these differences come from, I have made a comparison of two cases with the code that was used in previous studies for cases investigated here. It turns out that the results I get from running WaterSip V3.2, based on a global run with 5 million particles using ERA-Interim (Läderach and Sodemann, 2016) look entirely different for the Greenland case (Fig. 1), and only somewhat similar for the South Africa case (Fig. 2). The differences are so large that I do not think this could be due to the WRF simulation setup, and I rather suspect that the way the Sodemann et al. (2008) algorithm has been implemented differs in some way from the original. In fact, the results in Fig. 1b suggest that only a small contribution comes from the North Atlantic domain as defined by the authors, and a larger part from the North American continent. As the findings from the Greenland case are a cornerstone in the argumentation throughout the manuscript, this is a critical situation that needs to be resolved before publication.

Please note that WaterSip refers to a specific computer code that implements the Sodemann et al. (2008) method for FLEXPART and LAGRANTO with additional diagnostics. This software is now in public discussion (Sodemann 2025). In order to avoid confusing readers with regard to what software has been used, I would like the authors to not claim that they use WaterSip in their analysis, but rather their own implementation of an algorithm that is maybe inspired or resembles the Sodemann et al. (2008) method. The major differences that I see for the Greenland case imply that something else is done in the authors' code that leads to moisture uptakes identified close to Greenland in that case, and it would be misleading to assign these differences to either WaterSip or the Sodemann et al. (2008) method in itself. The lacking sensitivity to $\Delta q$ is another hint at substantial differences in the author's code to what is done in Sodemann et al. (2008).

Thank you very much for continuing the review of our article. As you mention, the results presented were obtained with our own implementation of the Sodemann et al. (2008) method. For clarity and rigor, as you suggest, we will explicitly state this in the revised version of the manuscript, claiming that we cannot assure that the results are identical to those obtained with WaterSip. We will also change everywhere "WaterSip" to "SOD08". In addition, as requested by the editor, we will make our code available in an open-access repository, hoping this helps to clarify some of the doubts you have regarding our results.

In an attempt to explain the differences you observed between your results and ours, we generated a similar figure to the one included in your review by changing the scale accordingly (Fig. 1 in this reply). As you mention, the discrepancies for the Greenland case in the No ABL configuration are noticeable, as we obtain a very important contribution from a large area just to the South of Greenland, and this may suggest that

we made a mistake when implementing the Sodemann et al. (2008) method. However, the discrepancies can better be explained by the differences between the WRF simulation and the reanalysis, as the pattern obtained in the case of FLEXPART-ERA5 trajectories is much more similar to yours (see Fig. 2).

To better illustrate that the results shown here correspond to those in our manuscript, at the end of this document you can find Fig. 4, which is exactly Fig. S8 in the supplement, and Fig. 5  (same figure with the new scale). Figure 6 and Fig. 7 present the same results for FLEXPART-ERA5 trajectories. We remark that the precipitation sources in this last case were not included in the manuscript or the supplement so far. As you may observe, for the other cases analyzed the differences between the results obtained with FLEXPART-WRF and FLEXPART-ERA5 are much less important than in the Greenland case.

[Figure]

Figure 1: Precipitation sources for the Greenland and South Africa events, computed with our implementation of the Sodemann et al. (2008) method, for trajectories generated with FLEXPART-WRF. A threshold of 0.2 g kg-1 in a 6 h interval is used in all cases. In panels a) and c) all moisture increments larger than the previous threshold are considered, while in panels b) and d) increments above the ABL are also neglected.

[Figure]

Figure 2: Precipitation sources for the Greenland and South Africa events, computed with our implementation of the Sodemann et al. (2008) method, for trajectories generated with FLEXPART-ERA5. A threshold of 0.2 g kg-1 in a 6 h interval is used in all cases. In panels a) and c) all moisture increments larger than the previous threshold are considered, while in panels b) and d) increments above the ABL are also neglected.

We believe that the similarities between the FLEXPART-ERA5 results and those in your review —especially in the No ABL setup— support the correctness of our implementation of the Sodemann et al. (2008 method. While some differences remain, there are several plausible explanations for them:

- First, and most importantly, the differences between the reanalysis driving the Lagrangian models. In our case we use ERA5, while the results you show correspond to simulations done with ERA-Interim. Apart from the improved resolution, ERA5 provides much more reliable precipitation fields. In Fig. 3 we compare the total precipitation in the Greenland event derived from ERA5 (left) and ERA-Interim (right), where we can check the differences between both patterns. While the Sodemann et al. (2008) method does not use directly the precipitation, it uses specific humidity, and there need to be also differences between ERA5 and ERA-Interim specific humidities to close the water balance. Thus, the different reanalysis used can affect the calculated moisture source field.
- Second, the configuration of the Lagrangian model. Apart from a different version of FLEXPART, as we use FLEXPART v10.4, Pisso et al. (2019), in Läderach and Sodemann (2016) it is stated that, on average, 70 air parcels reside at each column over a 1º x 1º grid cell and instant. Thus, an average of 6 x 4 x 70 = 1680 parcels are being used for the calculation of the moisture sources at each instant. In our case we are not using global simulations, but releasing 500000 parcels hourly over the black box shown in Fig. 3. This different configuration may also contribute to the discrepancies observed — particularly in the ABL setup, where the largest differences occur. In this case, aside from our use of a larger number of parcels, which could help identify more moisture uptakes within the ABL, it is also important to note that the boundary layer height computed by different versions of FLEXPART (each using a different reanalysis) may not align, potentially leading to inconsistencies.

[Figure]

Figure 3: Precipitation during the Greenland event derived from the ERA5 (left panel) and ERA-Interim (right panel). The black box shows the region where precipitation is being tracked.

I still find the setup of the WRF moisture source tags overly coarse to make strong conclusions. In their reply, the authors state that they made a simulation with more

finely resolved source boxes, but I do not see this discussed or shown in the manuscript, except for a figure in the supplementary material that is not mentioned anywhere (maybe I missed it). If the setup stays like this in the manuscript, it is in my opinion of critical importance to openly discuss the limitations of their approach in this study. For example, the boundary for the South Africa case at 30 deg N has the source footprint just south of that boundary (Fig. 2). This makes the verification statistics very sensitive so small meridional shifts in the source footprint, while really the AR case is one dominated by zonal advection. Thus, if the aim is to verify differences in the remoteness of sources, a setup with boxes over different longitudes would be needed to make the results more robust and help to support the conclusions, rather than a setup with different latitudes as is now shown in the supplementary material. All this may be still ok for showing the principle of how such a verification can be done, but such limitations need to be stated and discussed clearly to come to a well-justified overall assessment.

Regarding the finer configuration of sources that we used for a better comparison, the results are included in Fig. S12 of the supplement, as you point out. This is indeed commented in the last paragraph of Sect. 3.2.2 (L453 to L462), after the strongest differences between both approaches appear by comparing Fig. 6 and 8:

"Finally, an important difference can be observed by comparing the results for the Greenland case in Figures 6 and 8. In the case of WaterSip (even in the "RH" configuration) there is an important contribution from the northernmost part of the North Atlantic source (above 45º N), whereas this contribution is much less important in the case of the DB99 methodology. Our selection of source regions when comparing with WRF-WVTs overlooks this difference, and this could make our results not valid. However, by looking at the precipitation sources fields for all cases in Figures S8 and S9 in the Supplement we observe that only for the Greenland case there are important differences between the fields computed with the two different approaches. Moreover, we recomputed the RMSEs in Figures 5 and 7 with a finer (and more complex) selection of source regions, such that the ocean where the AR is located for each case is divided in four regions, instead of two. The results, shown in Figure S12 in the Supplement, demonstrate that the modifications we analyze and propose here provide also the best configuration with this new selection of source regions. "

In order to better reflect the limitations of our work we will move this paragraph to the conclusions. We will also mention that a moisture source setup with finer and different regions would be better to obtain more robust conclusions about the performance of the Lagangian methodologies. However, we understand that the changes we propose are validated by our experiments for the objective we posed: obtaining a Lagrangian methodology which can be an alternative to WRF-WVTs, where the tagged regions are typically large (oceanic basins or continents).

I also have a number of minor and technical comments below:

L. 14 and elsewhere: Instead of "WaterSip", which denotes a particular implementation of the Sodemann et al. (2008) method, refer to your own implementation of the Sodemann et al. (2008) algorithm here and elsewhere.

We will explicitly state in the introduction that we are assessing our implementation of both methodologies, by changing the first sentence of the paragraph starting at L86 to "In our case, the FLEXPART-WRF model is employed to generate back trajectories of air parcels contributing to precipitation in five AR events, and our own implementation

of two widely used Lagrangian diagnostic tools for estimating moisture sources are assessed: the Sodemann et al. (2008), and the Dirmeyer and Brubaker, (1999) methodologies.". Moreover, we will change WaterSip to Sodemann et al. (2008), abrreviated to SOD08, everywhere, and in Sect. 2.3, L192, we will explicitly claim that we cannot assure our implementation of this method is identical to WaterSip.

340

L. 37: This reference is missing from the reference section:

Sodemann, H., Wernli, H. and Schwierz, C., 2009: Sources of water vapour contributing to the Elbe flood in August 2002: A tagging study in a mesoscale model, Quart. J. Royal Meteorol. Soc., 135, 205-223, doi:10.1002/qj.374.

345

We will include this article in the reference section and correct the citation at L72.

L. 43 and elsewhere: There is occasionally an extra comma before the bracket with references.

350

Thank you for bringing this to our attention. We will review the entire manuscript and remove the extra commas before the brackets with references.

L. 60: remove extra '.'

We will remove the extra punctuation mark in the revised version of the manuscript.

L. 150: What are the units of QFX?

355

QFX is the surface moisture flux, so its units are kg m$^{-2}$ s$^{-1}$.

L. 152: The term "assimilate" is used here in a non-standard way, and can be confused with data assimilation (which I don't have the impression is meant here). Please rephrase to e.g., "ingest directly from ERA5" or the like.

Thank you for this suggestion. We will change "assimilate" to "ingest" in the revised manuscript.

360

L. 177: This should be cited as Sodemann and Stohl (2009) as in the references.

The citation will be corrected in the revised version of the manuscript.

L. 183: Here it says hourly, in the sentence in L. 187 it says 3-hourly. Which one is correct?

365

Both are correct. In L183 we state that we are storing hourly trajectories for both FLEXPART-ERA5 and FLEXPART-WRF, while in L187 we refer to input data, which is 3-hourly in the case of FLEXPART-WRF and hourly in the case of FLEXPART-ERA5.

L. 188: Give a typical pressure height of the topmost retained level.

Thank you for this suggestion. In the revised version of the manuscript we will indicate that a typical pressure for the highest ERA5 level we use is 140 hPa.

370

L. 208: add "at arrival" after relative humidity.

We will add "at arrival" after relative humidity in the revised version of the manuscript.

L. 222: 0.05 g kg 6 h-1 is only ¼ of the typically used value for mid-latitudes of 0.2 g kg 6 h-1

375

This is a typo. The results included here correspond to the recommended configuration, 0.2 g kg$^{-1}$ for a 6 h interval.

L. 262: "coming also": check sentence

We agree that the last sentence is not correctly written right now. We will change it to "Finally, in Sect. 3.3 we test the proposed modifications when the trajectories are generated by FLEXPART-ERA5, with input data from the ERA5 reanalysis. In this case, the additional fields required by the diagnostic tools (e.g., evaporation and precipitable water in the case of DB99) are also taken from ERA5, rather than from WRF simulations."

380

L. 264: "basic results" -> "baseline"?

385    Thank you for the suggestion. We will rephrase the heading of Sect 3.1 from "Basic results for WRF-WVTs vs WaterSip and DB99 (Dirmeyer and Brubaker, 1999)" to "Raw comparison of WRF-WVTs vs SOD08 and DB99".

Figure 3: I don't really see a great value in using a figure instead of a table, which I would think would be clearer/more easy to read with bigger font and better contrast.
390    Values could by highlighted by making them bold or italtic.

We appreciate your suggestion, but we consider that a figure with a continuos color scale is more appropriate to highlight the different contributions of the moisture sources.

L. 285 and onward: The terms "good" and "bad" or "better" and "worse" could be more
395    neutrally be rephased by using expressions such as "more consistend with reference" or "less consistent", or "larger deviation", "smaller deviation". There is not really a way of telling objectively here at what value a difference can be considered good or bad.

Thank you for your comment. We will replace these terms with others more precise in the revised version of the manuscript.

400    L. 320: Such fluctuations are always a remaining part of the uncertainties from offline trajectory calculations. This is why the Δq threshold has been introduced. Given that there is virtually no impact from the threshold on the results, as in Fig. 5 indicates that there could be an error in the way the method has been implemented.

There are differences when changing Δq, but the impact on the RMSE is small
405    compared to that of the time step. For example, the precipitation fractions for the source "Tropical Atlantic" in the Iberian Peninsula case are 19.87, 22.80, 22.74, 22.68 and 22.59 for Δq equal to 0.01, 0.05, 0.1, 0.2 and 0.3 g kg-1 for a 6 h interval. The change is relevant when going from 0.01 to 0.05, and then there appears to be a plateau. If we continued to increase Δq we would probably obtained more different
410    results, although the overall RMSE would probably decrease.

As mentioned before, we believe our implementation of the Sodemann et al. (2008) method to be correct, and we will make our code available in an open access repository, in accordance with ESD policies.

Figure 5: The low contrast in some of the tables makes the numbers hard to read.

415    Thank you for the suggestion. In the revised version of the manuscript we will change the color of the first rows in panels c) and d) to make them more readable.

L. 342: It was not easy to follow the narrative here, please rephrase.

Thank you for your comment. We will reformulate this sentence to "Figure 5a shows the RMSE of the precipitation fractions computed using the SOD08 diagnostic, using

420    WRF-WVTs results as the true values (Fig. 3), while Fig. 5b shows the average RMSE across the five cases.".

L. 353: important for what?

In L355 we explain that in the South Africa, Iberian Peninsula and Greenland cases there is a large bias for the contribution of the most important source (South Atlantic in
425    the first case, North Atlantic in the other cases), which is almost halved when introducing the proposed modification. Thus, it makes sense to consider the improvement an important one. In any case, in the revised version we will change "important" to "noticeable" for clarity.

L. 358 and L. 362: significantly: rephrase if no statistical test has been performed

430    Thank you for this suggestion. In the revised version of the manuscript, we will change "significantly higher" to "noticeably higher" and "significant effect" to "considerable effect".

L. 369: Move to figure caption

In the revised version of the manuscript, we will clarify that "the most basic
435    configuration" refers to the "No ABL setup" in the caption of Fig. 6.

L. 374: What means "the moisture field is less intense"?

It is a typo, it should be "the moisture source field is less intense". We are trying to explain that the values of the spatial distribution of precipitation sources are smaller. For clarity we will rephrase this sentence as "the moisture source field takes lower
440    values over the North Atlantic"

[Figure]

Figure 4: Precipitation sources for the different AR-related rainfall events, computed with WaterSip, for trajectories generated with FLEXPART-WRF. Panels show the results for the most basic configuration, while panels on the right present the results of the "RH" configuration. The fraction of precipitation coming from the tropics and the extratropics is shown in black for each case, and the red box shows these same contributions from WRF-WVTs.

[Figure]

Figure 5: Same as Fig. 4, different scale.

[Figure]

Figure 6: Precipitation sources for the different AR-related rainfall events, computed with WaterSip, for trajectories generated with FLEXPART-ECMWF. Panels show the results for the most basic configuration, while panels on the right present the results of the "RH" configuration. The fraction of precipitation coming from the tropics and the extratropics is shown in black for each case, and the red box shows these same contributions from WRF-WVTs.

455

[Figure]

Figure 7: Same as Fig. 6, different scale.